# Sliced Distributional Reinforcement Learning

## Abstract

Distributional reinforcement learning (DRL) models full return distributions rather than expectations, but extending to multivariate settings can be challenging. Univariate tractability is lost, and multivariate approaches are either computationally expensive or lack contraction guarantees. We propose Sliced Distributional Reinforcement Learning (SDRL) which lifts the tractable one-dimensional divergences to the multivariate case through random projections and aggregation. We prove Bellman contraction under uniform slicing for shared scalar discounts and under max slicing for general anisotropic matrix-discount updates, providing the first contraction result in this setting. SDRL accommodates a broad class of base divergences, instantiated here with Wasserstein, Cramér and Maximum Mean Discrepancy (MMD). In experiments, SDRL achieves competitive results on multivariate control tasks in MO-Gymnasium. As an application of matrix discounting, we extend multi-horizon RL with hyperbolic scalarization to the distributional regime. Taken together, these findings position slicing as a principled and scalable foundation for multivariate distributional reinforcement learning.[1]

## 1 Introduction

Distributional reinforcement learning (DRL) models return full distributions rather than expectations, with strong empirical (Dabney et al., 2018b;a; Barth-Maron et al., 2018; Hessel et al., 2017) and theoretical support (Lyle et al., 2019; Rowland et al., 2018; 2019a), building on the foundational perspective of Bellemare et al. (2017a; 2023b). In practice, DRL hinges on two choices, the distributional discrepancy and the critic's parameterization (Rowland et al., 2019a). In the univariate case, many tractable solutions exist. Discrepancies such as Wasserstein or KL admit efficient estimators, and parameterizations like quantiles or categorical grids are straightforward. This tractability is largely lost in the multivariate setting, categorical grids explode combinatorially, quantile parameterizations do not scale, and Wasserstein estimation becomes costly, typically $\mathcal{O}(n^3 \log n)$ for optimal transport solvers (Genevay et al., 2018).

A classical approach to high dimensional comparison is *slicing*, which represents multivariate distributions by their one dimensional projections and aggregates discrepancies across directions. This idea underlies *Sliced Probability Divergences* (SPDs) (Rabin et al., 2011; Bonneel et al., 2015; Nadjahi et al., 2020), where distributions are projected onto random directions, one dimensional discrepancies are computed, and the results are aggregated. This projection–aggregation mechanism reduces multivariate comparison to a series of tractable univariate computations, enabling the use of base divergences with efficient one dimensional estimators. With Wasserstein as the base metric, this yields the Sliced Wasserstein Distance (SWD), widely adopted in generative modeling for its simplicity, stability, and $\mathcal{O}(n \log n)$ per slice cost (Kolouri et al., 2019b; Wu et al., 2019; Deshpande et al., 2018; 2019; Liutkus et al., 2019), while avoiding adversarial games (Arjovsky et al., 2017).

We introduce a DRL framework built on SPDs, leveraging tractable one-dimensional projections to compare multivariate return distributions efficiently. Our approach lifts base divergences with efficient one-dimensional estimators, such as Wasserstein or Cramér, to the multivariate setting. Following the sample-based critic paradigm (Nguyen-Tang et al., 2021), our critics generate samples from the value distribution and are optimized with sliced objectives. Concretely, we adopt a

---

[1]We will make the code publicly available upon acceptance of the paper.

reparameterized generative model that maps noise to *true* samples (Singh et al., 2022), providing a flexible parameterization that scales to multivariate settings while preserving the computational advantages of sliced methods.

Beyond random slicing, we rely on the max slicing framework (Deshpande et al., 2019) to lift these divergences in a stronger form. Max slicing replaces the aggregation over random directions with an optimization, yielding divergences that remain contractive in settings more general than scalar-discounted multivariate returns. This extension opens the door to a wide range of future applications where contractivity beyond the standard RL setup is essential.

**Contributions**

- We introduce **Sliced Distributional RL (SDRL)**, the first framework for multivariate returns with sliced divergences, and prove contraction of the usual distributional Bellman operator under scalar discount.

- We extend to a **Max-Sliced (MSDRL)** variant, establishing contraction guarantees for the general case of *matrix-discounted* multivariate Bellman updates.

## 2 BACKGROUND AND RELATED WORKS

In the *expected* reinforcement learning framework, an agent interacts with an environment modeled as a Markov decision process (MDP) $(\mathcal{S}, \mathcal{A}, P, R, \{\Gamma_t\}_{t \geq 0})$, where rewards may be *d-dimensional* $(R_t \in \mathbb{R}^d, d \geq 1)$. Here $\Gamma_t \in \mathbb{R}^{d \times d}$ denotes a (possibly dense) time-varying discount–mixing matrix; any implicit dependence on the transition is suppressed in the subscript $t$. Given a policy $\pi(a|s)$, the agent seeks to maximize the expected discounted return

$$Q^\pi(s, a) = \mathbb{E}\left[\sum_{t=0}^{\infty} \Big( \prod_{k=1}^{t} \Gamma_k \Big) R_t \,\Big|\, S_0 = s, A_0 = a \right], \tag{1}$$

with the convention $\prod_{k=1}^{0} \Gamma_k = I_d$. Classical RL methods focus on estimating $Q^\pi(s, a)$, the *expectation* of the return distribution (componentwise when $d > 1$).

The *distributional* perspective (Bellemare et al., 2017a), originally developed for scalar rewards with scalar discounting, can be applied here as well: it models the full return random variable

$$Z^\pi(s, a) = \sum_{t=0}^{\infty} \Big( \prod_{k=1}^{t} \Gamma_k \Big) R_t, \tag{2}$$

whose expectation recovers $Q^\pi(s, a) = \mathbb{E}[Z^\pi(s, a)]$ (componentwise when $d > 1$). This viewpoint leads to the *distributional Bellman operator* with time-dependent matrix discount:

$$(\mathcal{T}^\pi Z)(s, a) \stackrel{D}{=} R(s, a) + \Gamma_1 Z(S', A'), \quad A' \sim \pi(\cdot|S'), \ S' \sim P(\cdot|s, a), \tag{3}$$

where $\stackrel{D}{=}$ denotes equality in distribution.

**Special cases.**

1. **Classical distributional RL:** $d = 1, \Gamma_t = \gamma \in [0, 1)$.

2. **Multivariate with shared scalar discount:** $d > 1, \Gamma_t = \gamma I_d$ (Zhang et al., 2021).

3. **Time-invariant general matrix:** $\Gamma_t \equiv \Gamma$, e.g., multi-horizon design with $\Gamma = \text{diag}(\gamma_1, \ldots, \gamma_d)$ assigning distinct horizons to objectives (Fedus et al., 2019).

4. **Time-varying dense matrix:** $\Gamma_t$ evolves over time and may couple objectives (the multivariate analogue of Generalized Value Functions (Sutton et al., 2011)).

More details and examples of this matrix-discounted perspective are given in Appendix A.

**Related work.** Several alternative divergences have been investigated in the multivariate case. We briefly review the approaches most relevant to our setting.

**Adversarial $W_1$.** Freirich et al. (2019) reinterpret the distributional Bellman equation as a GAN problem, optimized with WGAN-style training where the discriminator approximates $W_1$ (the Wasserstein-1 distance) (Villani et al., 2008). While motivated by contraction properties of Wasserstein metrics $W_p$ (Bellemare et al., 2017a), practical discriminators can suffer from Lipschitz violations, finite-sample bias, and optimization error (Mallasto et al., 2019), yielding objectives that may deviate substantially from true optimal-transport distances (Mallasto et al., 2019; Stanczuk et al., 2021), thus weakening contraction claims that presume exact $W_1$.

**MMD.** Moment matching with MMD was explored in the univariate case (Nguyen-Tang et al., 2021) and later extended to multivariate returns (Zhang et al., 2021). In the multivariate setting, contractivity results are available only for a narrow class of kernels (Wiltzer et al., 2024a), and identifying a kernel that is both empirically strong and contractive remains challenging (Killingberg & Langseth, 2023a). Consequently, practitioners often resort to Gaussian mixture despite their limited contractivity guarantees in the multivariate setting.

*Synthesis.* Taken together, existing approaches suffer from at least one of three limitations: performant variants are non-contractive, theoretical guarantees do not extend to the general anisotropic discount setting we target, or the estimation is too loose to support contraction claims (adversarial $W_1$). This gap motivates our sliced approach.

# 3 SDRL: Distributional RL via Sliced Probability Divergences

## 3.1 Sliced Probability Divergences

**Slicing a base divergence.** Let $\Delta : \mathcal{P}(\mathbb{R}) \times \mathcal{P}(\mathbb{R}) \to \mathbb{R}_+ \cup \{\infty\}$ be a divergence on one–dimensional probability laws. For a direction $\theta \in \mathbb{S}^{d-1}$, let $P_\theta : \mathbb{R}^d \to \mathbb{R}$ denote the linear projection $P_\theta(x) = \langle \theta, x \rangle$, and write $(P_\theta)_{\#}\mu$ for the pushforward of $\mu \in \mathcal{P}(\mathbb{R}^d)$ by $P_\theta$. With $\sigma$ the uniform measure on $\mathbb{S}^{d-1}$ and $p \geq 1$, the associated sliced probability divergence (SPD) is

$$\mathbf{S}\Delta_p^p(\mu, \nu) = \int_{\mathbb{S}^{d-1}} \Delta^p\big((P_\theta)_{\#}\mu, (P_\theta)_{\#}\nu\big) \, d\sigma(\theta), \qquad \mu, \nu \in \mathcal{P}(\mathbb{R}^d). \tag{4}$$

This averages a 1D discrepancy across random linear views, lifting $\Delta$ to multivariate laws (Nadjahi et al., 2020).

**Monte Carlo approximation.** In practice, this integral is estimated via Monte Carlo sampling by drawing $N$ i.i.d. directions $\{\theta_i\}_{i=1}^N \sim \sigma$ and computing

$$\widehat{\mathbf{S}\Delta}_p^p(\mu, \nu) = \frac{1}{N} \sum_{i=1}^N \Delta^p\big((P_{\theta_i})_{\#}\mu, (P_{\theta_i})_{\#}\nu\big). \tag{5}$$

Each projected subproblem is independent, so the $N$ evaluations can be carried out in parallel.

**Sliced Wasserstein distance.** Among sliced probability divergences, the most widely used instance is the *sliced Wasserstein distance* (SWD) (Rabin et al., 2011; Bonneel et al., 2015), where the base divergence is chosen as $\Delta = \mathbf{W}_p$. For $\mu, \nu \in \mathcal{P}(\mathbb{R}^d)$ and $p \geq 1$,

$$\mathbf{SW}_p^p(\mu, \nu) = \int_{\mathbb{S}^{d-1}} \mathbf{W}_p^p\big((P_\theta)_{\#}\mu, (P_\theta)_{\#}\nu\big) \, d\sigma(\theta), \tag{6}$$

which reduces the high-dimensional Wasserstein problem to an average of one-dimensional Wasserstein distances between the projected pushforwards $(P_\theta)_{\#}\mu$ and $(P_\theta)_{\#}\nu$. For estimators from samples, the overall cost is $\mathcal{O}(L\,n \log n)$, as it involves $L$ sorts of the projected samples (each $\mathcal{O}(n \log n)$). This contrasts with solving a $d$-dimensional optimal transport problem, which typically costs $\mathcal{O}(n^3 \log n)$ (Genevay et al., 2019). Further details on properties and the estimator are provided in Appendix B.1.

**Sliced Cramér distance.** A natural family of discrepancies is the $\ell_p$ distances between cumulative distribution functions:

$$\ell_p^p(\alpha, \beta) := \int_{\mathbb{R}} \left| F_\alpha(u) - F_\beta(u) \right|^p du, \tag{7}$$

where $F_\alpha$ and $F_\beta$ are the univariate CDFs of $\alpha, \beta$. The special case $p = 2$ is the *Cramér distance* (Bellemare et al., 2017b). The *sliced Cramér* distance lifts this metric to $\mathbb{R}^d$ via random projections:

$$\mathbf{SC}_2^2(\mu, \nu) = \int_{\mathbb{S}^{d-1}} \ell_2^2\big((P_\theta)_{\#}\mu, (P_\theta)_{\#}\nu\big) \, d\sigma(\theta). \tag{8}$$

This distance is also known as the *Cramér–Wold distance* and has already been investigated in the context of machine learning (Knop et al., 2020; Kolouri et al., 2020). Its estimator has the same complexity as sliced Wasserstein, as the Cramér distance can be estimated in $\mathcal{O}(n \log n)$. The use of the Cramér distance in distributional RL has been explored in prior work (Rowland et al., 2018; Lhéritier & Bondoux, 2021; Théate et al., 2023). Further properties and the estimator we use are detailed in Appendix B.2.

**Sliced MMD.** Another tractable choice is the *Maximum Mean Discrepancy* (MMD) (Gretton et al., 2012), which has already been explored in distributional RL (Nguyen et al., 2020b; Killingberg & Langseth, 2023b; Wiltzer et al., 2024b). For laws $P, Q \subset \mathbb{R}^{d'}$ and a kernel $k$, the squared **MMD** is

$$\mathbf{MMD}^2(P, Q) = \mathbb{E}_{x, x' \sim P}[k(x, x')] + \mathbb{E}_{y, y' \sim Q}[k(y, y')] - 2\,\mathbb{E}_{x \sim P, y \sim Q}[k(x, y)]. \tag{9}$$

Lifting this discrepancy through random projections yields the *sliced MMD*: for $\mu, \nu \in \mathcal{P}(\mathbb{R}^d)$,

$$\mathbf{SMMD}_k^2(\mu, \nu) = \int_{\mathbb{S}^{d-1}} \mathbf{MMD}_k^2\big((P_\theta)_{\#}\mu, (P_\theta)_{\#}\nu\big) \, d\sigma(\theta). \tag{10}$$

Sliced MMD was first introduced in Nadjahi et al. (2020). Its sliced estimator from samples scales as $\mathcal{O}(L\,n^2)$, as the base MMD estimator is quadratic in $n$. More details on the properties of MMD and the estimator we use are provided in Appendix B.3.

### 3.2 MAX SLICED PROBABILITY DIVERGENCES.

Uniform random slicing may be inefficient as many directions could be needed to get an accurate picture of the discrepancy between two distributions. Moreover, as discussed in Section 4, uniform sliced divergences are not sufficient to establish contraction under the most general class of Bellman updates we target, namely those with general discount matrices. One solution proposed in Deshpande et al. (2019) involves learning the most discriminative projection direction, along which the 1D marginal divergence is the largest, in an adversarial way (Goodfellow et al., 2014).

$$\mathbf{MS}\Delta(\mu, \nu) = \sup_{\theta \in \mathbb{S}^{d-1}} \Delta\big((P_\theta)_{\#}\mu, (P_\theta)_{\#}\nu\big), \qquad P_\theta(x) = \langle \theta, x \rangle. \tag{11}$$

This framework was originally proposed for $\Delta = \mathbf{W}_p$, yielding the *max–sliced Wasserstein distance* $\mathbf{MSW}_p$ (Deshpande et al., 2019). By analogy, we denote by $\mathbf{MSC}_2$ and $\mathbf{MSMMD}_k$ the max–sliced Cramér distance and max–sliced MMD, respectively.

**Estimation.** Since the supremum in the definition of max–sliced divergences cannot be computed exactly, it is typically approximated by iterative optimization of the projection direction on the unit sphere. At each step a gradient ascent update on the divergence is followed by renormalization onto the unit sphere, and the final direction defines the empirical estimate. The full procedure is outlined in Algorithm 2.

### 3.3 PROBLEM SETTING AND ALGORITHMIC APPROACH

We wish to model the *joint vector of multivariate returns* in order to capture their correlations and higher-order structure, rather than only marginal statistics. Let $d > 1$ and $\mathcal{X} = \mathbb{R}^d$. For any policy $\pi(\cdot \mid s)$ (discrete or continuous actions), let $\mu^\pi(s, a) \in \mathcal{P}(\mathcal{X})$ denote the law of the multivariate return $Z^\pi(s, a)$. The distributional Bellman operator $\mathcal{T}^\pi$ relates return laws across state–action pairs via

$$(\mathcal{T}^\pi \mu)(s, a) := \int_{\mathcal{S}} \int_{\mathcal{A}} \int_{\mathcal{X}} (f_{\Gamma, r})_{\#} \mu(s', a') \, R(dr \mid s, a) \, \pi(da' \mid s') \, P(ds' \mid s, a), \tag{12}$$

---

**Algorithm 1:** Distributional policy evaluation with sliced divergence

---

**Input:** Number of samples $N$; base divergence $\Delta$ and order $p$; discount matrix $\Gamma$

**Input:** Either projection count $L$ *or* a projection direction $\theta$

**Input:** Sample transition $(s, a, r, s')$; policy $\pi$; model parameters $\phi$ (and target $\phi^-$)

$a' \sim \pi(\cdot|s')$

**for** $i = 1, \ldots, N$ **do**

$\quad \varepsilon_i \sim p(\varepsilon)$                                        `// noise for predicted sample`

$\quad \tilde{\varepsilon}_i \sim p(\varepsilon)$                                `// independent noise for target`

$\quad z_i \leftarrow Z_\phi(s, a, \varepsilon_i)$                               `// predicted sample`

$\quad \hat{z}_i \leftarrow r + \Gamma\, Z_{\phi^-}(s', a', \tilde{\varepsilon}_i)$                  `// target sample`

**Choose projection set** $\Theta$**: if** *a direction $\theta$ is provided (max setting)* **then** $\Theta \leftarrow \{\theta\}$

**else** draw $\{\theta_\ell\}_{\ell=1}^L \sim \mathrm{Unif}(\mathbb{S}^{d-1})$ and set $\Theta \leftarrow \{\theta_\ell\}_{\ell=1}^L$

**Monte Carlo estimator over projections (operate directly on samples):**

$S \leftarrow \frac{1}{|\Theta|} \sum_{\theta' \in \Theta} \left[ \Delta\big(\{\langle \theta', z_i \rangle\}_{i=1}^N, \; \{\langle \theta', \hat{z}_i \rangle\}_{i=1}^N \big) \right]^p$        `// ` $P_{\theta'}(x) = \langle \theta', x \rangle$

**Output:** $S\Delta_p^p\big(\{z_i\}_{i=1}^N, \{\hat{z}_i\}_{i=1}^N\big) \leftarrow S$

---

where $f_{\Gamma,r}(z) = r + \Gamma z$ for $z \in \mathbb{R}^d$ and $\Gamma \in \mathbb{R}^{d \times d}$ is a (possibly dense) discount–mixing matrix. The target in policy evaluation is the fixed point $\mu^\pi$ of $\mathcal{T}^\pi$, i.e., $\mathcal{T}^\pi \mu^\pi = \mu^\pi$.

**Control via scalarization.** The multivariate distributional policy evaluation above can be plugged into any control learning method once a fixed scalarization rule is chosen, e.g. a linear functional $\alpha^\top \mathbb{E}[Z^\pi(s, a)]$ or a rule induced by $\Gamma$. This scalarized value recovers a standard RL control signal, enabling the use of off-the-shelf algorithms such as DQN for discrete action spaces (Mnih et al., 2013) or DDPG for continuous ones (Lillicrap et al., 2015), while retaining the multivariate distributional critic for stability and richer statistical modeling.

**Algorithmic approach** We approximate $\mu^\pi(s, a)$ with a reparameterized generator $Z_\phi(s, a, \varepsilon)$, where the noise variable $\varepsilon$ is typically drawn from $p(\varepsilon) = \mathcal{N}(0, I)$. Given a transition $(s, a, r, s')$ and next action $a' \sim \pi(\cdot|s')$, we draw $N$ samples of $\varepsilon$ to produce predicted samples $z_i = Z_\phi(s, a, \varepsilon_i)$ and target samples $\hat{z}_i = r + \Gamma Z_\phi(s', a', \tilde{\varepsilon}_i)$, which represent the current law and its $\Gamma$-discounted Bellman target. Their discrepancy is measured by a sliced probability divergence with base $\Delta$, using either $L$ random projections or a single optimized direction (max–sliced). The loss is the Monte Carlo average of projected divergences, and minimizing it w.r.t. $\phi$ yields a distributional TD update toward the matrix-discounted target (Algorithm 1).

## 4 THEORETICAL RESULTS

In this section, we provide the theoretical foundations of multivariate distributional RL with sliced divergences. We use the notion of a *supremum divergence* and establish sufficient conditions under which these divergences yield contraction of the distributional Bellman operator in the multivariate setting.

**Definition 1** (Supremum divergence)**.** *Let $\mathcal{D}$ be a divergence on probability laws and let $\mu, \nu : \mathcal{S} \times \mathcal{A} \to \mathcal{P}(\mathbb{R}^d)$. The* supremum divergence *is defined as*

$$\overline{\mathcal{D}}(\mu, \nu) := \sup_{(s,a) \in \mathcal{S} \times \mathcal{A}} \mathcal{D}\big(\mu(s, a), \nu(s, a)\big). \tag{13}$$

We focus on the following questions:

1. **Metric property:** If the base divergence $\Delta$ is a metric on $\mathcal{P}(\mathbb{R})$, when do $\overline{\mathbf{S\Delta}}^p$ and $\overline{\mathbf{MS\Delta}}^p$ induce metrics on $\mathcal{P}(\mathbb{R}^d)^{\mathcal{S} \times \mathcal{A}}$?

2. **Contraction property:** Under what conditions on $\Delta$ and on the discount structure $\Gamma$ does the Bellman operator $\mathcal{T}^\pi$ contract in $\overline{\mathbf{S\Delta}}^p$ or $\overline{\mathbf{MS\Delta}}^p$?

3. **Sample complexity:** How does the estimation error of the sliced and max–sliced divergences scale with the number of samples, and do they avoid the curse of dimensionality?

## 4.1 METRIC PROPERTY

It is known that uniform slicing preserves the metric property of a base divergence (Nadjahi et al., 2020). Similarly, Deshpande et al. (2019) established that the max–sliced Wasserstein distance is a metric; we extend this in Lemma 2, showing that max–slicing preserves the metric property for any base divergence. Finally, taking the supremum over state–action pairs also preserves metricity. These results are summarized in Theorem 1, with full proofs provided in Appendix C.1.

**Theorem 1.** *Assume $\Delta$ is a metric on $\mathcal{P}(\mathbb{R})$ and fix $p \in [1, \infty)$. Then: (i) $\mathbf{S}\Delta_p$ is a metric on $\mathcal{P}(\mathbb{R}^d)$; (ii) $\mathbf{MS}\Delta$ is a metric on $\mathcal{P}(\mathbb{R}^d)$; and (iii) for return–distribution maps $\eta_i : \mathcal{S} \times \mathcal{A} \to \mathcal{P}(\mathbb{R}^d)$, the sup–lifts $\overline{\mathbf{S}\Delta}_p$ and $\overline{\mathbf{MS}\Delta}$ are metrics on $\mathcal{P}(\mathbb{R}^d)^{\mathcal{S} \times \mathcal{A}}$.*

## 4.2 CONTRACTION PROPERTY

**Setup** Let $\mathcal{D}$ be any divergence between probability laws on $\mathbb{R}$, or on $\mathbb{R}^d$ after a lift. An operator $\mathcal{T}$ on return models is a $\kappa$ contraction with respect to $\overline{\mathcal{D}}$ if there exists $\kappa \in [0, 1)$ such that

$$\overline{\mathcal{D}}(\mathcal{T}\eta_1, \mathcal{T}\eta_2) \leq \kappa \overline{\mathcal{D}}(\eta_1, \eta_2) \quad \text{for all } \eta_1, \eta_2.$$

**Univariate contraction** A set of sufficient conditions under which the univariate distributional Bellman operator $\mathcal{T}^\pi$ is a $c(\gamma)$ contraction for $\overline{\mathcal{D}}$ is recalled (in a slightly generalized form from Bellemare et al. (2023a)) in Theorem 2, with the proof provided in Appendix C.2.1.

**Theorem 2.** *Let $\Delta$ be a metric on $\mathcal{P}(\mathbb{R})$. For $t \in \mathbb{R}$, let $T_t(x) = x + t$ denote translation, and for $\gamma \in (0, 1)$ let $S_\gamma(x) = \gamma x$ denote scaling. Suppose $\Delta$ satisfies:*

- **(T)** *Translation nonexpansion: $\Delta\big((T_t)_\# \mu, (T_t)_\# \nu\big) \leq \Delta(\mu, \nu)$ for all $t \in \mathbb{R}$.*

- **(S)** *Scale–Lipschitz: there exists a nondecreasing function $c : \mathbb{R}_{>0} \to \mathbb{R}_{>0}$ such that for every $s \geq 0$,*

$$\Delta\big((S_s)_\# \mu, (S_s)_\# \nu\big) \leq c(s) \Delta(\mu, \nu).$$

- **($\mathbf{M}_p$)** *Mixture $p$–convexity: for some $p \in [1, \infty)$, any probability measure $\rho$ and measurable families $(\mu_c), (\nu_c) \subset \mathcal{P}(\mathbb{R})$,*

$$\Delta\Big( \int \mu_c \, d\rho, \int \nu_c \, d\rho \Big) \leq \Big( \int \Delta(\mu_c, \nu_c)^p \, d\rho \Big)^{1/p}.$$

*Then the Bellman operator $\mathcal{T}^\pi$ is a $c(\gamma)$–contraction:*

$$\overline{\Delta}(\mathcal{T}^\pi \eta_1, \mathcal{T}^\pi \eta_2) \leq c(\gamma) \overline{\Delta}(\eta_1, \eta_2).$$

**Shared scalar discount (slicing)** We are now ready to introduce the main contraction results of this paper. We begin with the canonical multivariate case *with vector-valued objects in $\mathbb{R}^d$ and $d > 1$*, where the Bellman update involves the shared scalar discount introduced in Section 2. This setting coincides with those studied in Freirich et al. (2019); Zhang et al. (2021); Sun et al. (2024). Our result, however, also covers the more general form $\gamma O$ with $O \in \mathsf{O}(d)$, where $\mathsf{O}(d)$ is the set of $d \times d$ orthogonal matrices. The corresponding distributional Bellman update is

$$(\mathcal{T}^\pi \eta)(s, a) = \text{Law}\big(R(s, a) + \gamma I_d X'\big), \qquad X' \sim \eta(S', A'), \ S' \sim P(\cdot | s, a), \ A' \sim \pi(\cdot | S'), \quad (14)$$

*where $R(s, a) \in \mathbb{R}^d$, $X' \in \mathbb{R}^d$, and $\eta : \mathcal{S} \times \mathcal{A} \to \mathcal{P}(\mathbb{R}^d)$ with $d > 1$, and $I_d$ denotes the $d \times d$ identity matrix.*

The key observation is that the sufficient conditions **(T)**, **(S)**, **($\mathbf{M}_p$)** of Theorem 2, which guarantee contraction of a base divergence $\Delta$ in the univariate setting, can be lifted directly to show that the sliced divergence $\mathbf{S}\Delta$ is contractive in the multivariate setting of Equation 14 *with the same contraction constant $c(\gamma)$ as in the univariate case (**no dimension-dependent penalty**).* This is summarized in Theorem 3 whose proof can be found in Appendix C.2.2.

**Theorem 3.** *If a base divergence $\Delta$ satisfies **(T)**, **(S)** at $\gamma \in (0, 1)$ with $c(\gamma) < 1$, and **($\mathbf{M}_p$)**, then the Bellman operator $\mathcal{T}^\pi$ in equation 14 with scaled isometry updates on $\mathbb{R}^d$ for $d > 1$ is a $c(\gamma)$–contraction w.r.t. the sup–sliced divergence:*

$$\overline{\mathbf{S}\Delta}_p(\mathcal{T}^\pi \eta_1, \mathcal{T}^\pi \eta_2) \leq c(\gamma) \overline{\mathbf{S}\Delta}_p(\eta_1, \eta_2),$$

where $\eta_i : \mathcal{S} \times \mathcal{A} \to \mathcal{P}(\mathbb{R}^d)$ *and slicing uses the fixed $\sigma$ on $\mathbb{S}^{d-1}$.*

**General anisotropic discount (max–slicing)**  We now discuss the contraction of the maximum sliced divergence $\overline{\text{MS}\Delta}$. We do so under a much more general family of Bellman updates that covers any type of fixed or time-varying discount matrix $\Gamma_t$ as well as state–action dependent $\Gamma(s, a)$.

$$(\mathcal{T}^\pi \eta)(s, a) = \text{Law}\big(R(s, a) + \Gamma_t(s, a)\, X'\big), \tag{15}$$

$$X' \sim \eta(S', A'), \quad S' \sim P(\cdot | s, a), \quad A' \sim \pi(\cdot | S').$$

We show in Theorem 4 that the sufficient conditions on $\Delta$ extend to the max–sliced divergence $\overline{\text{MS}\Delta}$ under the Bellman update from Equation 15. The contraction constant is $c$ of the worst-case operator norm of the discount matrices, and, as with uniform slicing, this introduces **no explicit dimension-dependent penalty**. This result generalizes multivariate distributional RL to a much wider class of problems for which some examples are discussed in Appendix A. The proof can be found in Appendix C.2.3.

**Theorem 4.** *If a base divergence $\Delta$ satisfies* (T)*,* (S) *with $c$ nondecreasing, and* (M$_p$)*, then the Bellman operator $\mathcal{T}^\pi$ in equation 15 with anisotropic linear updates on $\mathbb{R}^d$ for $d > 1$ is a $c(\bar{L})$–contraction w.r.t. the sup–max–sliced divergence:*

$$\overline{\text{MS}\Delta}\big(\mathcal{T}^\pi \eta_1, \mathcal{T}^\pi \eta_2\big) \leq c(\bar{L})\, \overline{\text{MS}\Delta}(\eta_1, \eta_2),$$

*where $\eta_i : \mathcal{S} \times \mathcal{A} \to \mathcal{P}(\mathbb{R}^d)$ and $\bar{L} = \sup_{(s,a)} \sup_C \|A_{s,a}(C)\|_{\text{op}}$, with $C$ accounting for the one-step randomness.*

### 4.3 SAMPLE COMPLEXITY

We now analyze the sample complexity of uniform and max slicing. For the uniform case, Theorem 6, following the result of Nadjahi et al. (2020), shows that the sliced divergence inherits the one–dimensional sample complexity of its base divergence, without any additional dependence on the ambient dimension. For the maximum case, Theorem 7 relies on a bounded-support assumption, which is natural in RL where returns are often assumed bounded, and shows that, depending on the base divergence, one can obtain upper bounds that avoid the curse of dimensionality.

**Theorem 6.** *Fix $p \in [1, \infty)$. Let $\Delta$ be a divergence on $\mathcal{P}(\mathbb{R})$ and assume there exists a function $\alpha(p, n) \geq 0$ such that for every $\mu \in \mathcal{P}(\mathbb{R})$ with empirical $\hat{\mu}_n$ we have $\mathbb{E}\big[\Delta(\hat{\mu}_n, \mu)^p\big] \leq \alpha(p, n)$. Then for any $\mu \in \mathcal{P}(\mathbb{R}^d)$ with empirical $\hat{\mu}_n$,*

$$\mathbb{E}\big|\mathbf{S}\Delta_p^p(\hat{\mu}_n, \mu)\big| \leq \alpha(p, n).$$

**Theorem 7.** *Assume $\text{diam}(\text{supp}\, P) \leq D$. Let $\Delta$ be a divergence on $\mathcal{P}(\mathbb{R})$. Suppose that for any one–dimensional laws $\mu, \nu$ supported on an interval of length $\leq D$, there exist $\alpha \in (0, 1]$, $\beta \geq 0$, and $L > 0$ such that the* CDF–dominance *inequality $\Delta(\mu, \nu) \leq L\, D^\beta\, \|F_\mu - F_\nu\|_\infty^\alpha$ holds. Then*

$$\mathbb{E}\, \mathbf{MS}\Delta(P_n, P) = O\Big(D^\beta \big(\tfrac{d \log n}{n}\big)^{\alpha/2}\Big).$$

### 4.4 INSTANTIATIONS

Now we apply the theorems presented above to specific base divergences of interest. We verify that conditions (T), (S), (M$_p$) hold, and summarize the resulting contraction factors under both the standard multivariate Bellman update and the general matrix–discounted case in Table 1. For $\mathbf{MMD}_k$, we focus on the multiquadric (MQ) kernel from Killingberg & Langseth (2023a), defined as $k(x, y) = -\sqrt{c^2 \|x - y\|^2 + 1}$, which is known to perform best among contraction–inducing kernels. The result can be naturally extended to other similar kernels, but we restrict our analysis to MQ for clarity. We do not establish an upper bound for $\mathbf{MSMMD}_k$, leaving this as future work. Full proofs are provided in Appendix C.4.

## 5 EXPERIMENTS

**Setup.** We evaluate uniform and max-sliced divergences on continuous control tasks using MuJoCo (Todorov et al., 2012). All environments are drawn from the Gymnasium library (Towers

| Divergence | 3 prop. | Contr. factor $\gamma$ | Contr. factor $\bar{L}$ | Sample complexity |
|---|---|---|---|---|
| $\mathbf{SW}_p$ | ✓ | $\gamma$ | / | $\mathcal{O}\big(n^{-1/(2p)}\big)$ |
| $\mathbf{MSW}_p$ | ✓ | $\gamma$ | $\bar{L}$ | $\mathcal{O}\Big(D\big(\frac{d\log n}{n}\big)^{1/(2\beta)}\Big)$ |
| $\mathbf{SC}_2$ | ✓ | $\gamma^{1/2}$ | / | $\mathcal{O}\big(n^{-1/2}\big)$ |
| $\mathbf{MSC}_2$ | ✓ | $\gamma^{1/2}$ | $\bar{L}^{1/2}$ | $\mathcal{O}\Big(\sqrt{D}\sqrt{\frac{d\log n}{n}}\Big)$ |
| $\mathbf{SMMD}_k$ | ✓ | $\gamma^{1/2}$ | / | $\mathcal{O}\big(n^{-1/2}\big)$ |
| $\mathbf{MSMMD}_k$ | ✓ | $\gamma^{1/2}$ | $\bar{L}^{1/2}$ | $\times$ |

Table 1: Summary of contraction factors and sample complexity results for sliced and max–sliced divergences. Here $\beta := \max\{p,1\}$. For $\mathbf{MSMMD}_k$, the contraction factor simplifies from $\max\{\bar{L}^{1/2}, \bar{L}\}$ to $\bar{L}^{1/2}$ under the assumption $\bar{L} < 1$.

et al., 2024), with reward decompositions provided by MO-Gymnasium (Felten et al., 2023). Neural network implementations are developed in JAX (Bradbury et al., 2018).

**Proposals and baseline.** We compare sliced and max-sliced divergences against a standard baseline for multivariate distributional RL. Specifically, we experiment with slicing and max-slicing using the Wasserstein distance ($p=1, 2$), the Cramér distance, and $\mathbf{MMD}_{\mathrm{MQ}}$ (MMD under the multiquadric kernel). As a baseline, we include plain $\mathbf{MMD}_{\mathrm{MQ}}$, the most widely used divergence in multivariate distributional RL (Zhang et al., 2021; Wiltzer et al., 2024a). Further details on architectures and hyperparameters are provided in Appendix E.3.

**Shared scalar discount.** We first consider the multivariate setting with a shared scalar discount, modeling the joint distribution of discounted returns. For control, we use the same scalarization rule as in the univariate benchmarks, following prior work (Zhang et al., 2021; Sun et al., 2024). Details on reward decompositions and scalarization are provided in Appendix E.1. Figure 1a reports results on five MuJoCo environments for all the variants we introduced, with MMD serving as the baseline. Most variants converge to value distributions that are useful for control, and MMD with the MQ kernel stands out as a strong baseline, with many variants performing on par. Importantly, we used the same hyperparameters (e.g., number of max-slicing steps and learning rate) across all configurations.

**Anisotropic case: multi-horizon RL.** To motivate our framework beyond the shared-scalar setting, we consider a simple instance of the anisotropic case, namely *multi-horizon reinforcement learning* (Fedus et al., 2019), which models a vector of returns using distinct discount factors. Unlike prior work, we *jointly* model all discounted values in a single distributional Bellman update (Equation 15), with $\Gamma = \mathrm{diag}(\gamma_1, \ldots, \gamma_d)$ a diagonal discount matrix. As summarized in Table 1, this setting is contractive for max–sliced Wasserstein, max–sliced Cramér, and max–$\mathbf{MMD}_{\mathrm{MQ}}$. For control, we scalarize the vector of multi-horizon returns using the hyperbolic discount rule of Fedus et al. (2019). Concretely, if $w \in \mathbb{R}^d$ denotes the hyperbolic mixture weights over the geometric discounts $\{\gamma_i\}_{i=1}^d$, the scalarized value is $\langle w, \ \mathbb{E}[Z^\pi(s,a)]\rangle$, thereby extending hyperbolic discounting to the *distributional* setting. More information on this setting is provided in Appendix E.2. Figure 1b presents results on four MuJoCo environments. Once again, many variants prove effective on at least three tasks. Notably, the max-sliced variants, although contractive in this setting, do not exhibit superior performance.

## 6 CONCLUSION

In this work, we introduced the framework of Sliced Distributional RL (SDRL) and proposed several divergences that are provably contractive in the most common multivariate setting. We further extended these results with Maximum Sliced Distributional RL (MSDRL), which handles a broader class of Bellman updates involving general matrix discounts. We evaluated our approach on canonical multi-objective distributional RL tasks in several MuJoCo environments and showed that most of the variants we introduced are effective. As a practical application of general matrix discounting, we also experimented with multihorizon distributional RL, where the new divergences successfully learned multivariate value distributions useful for control.

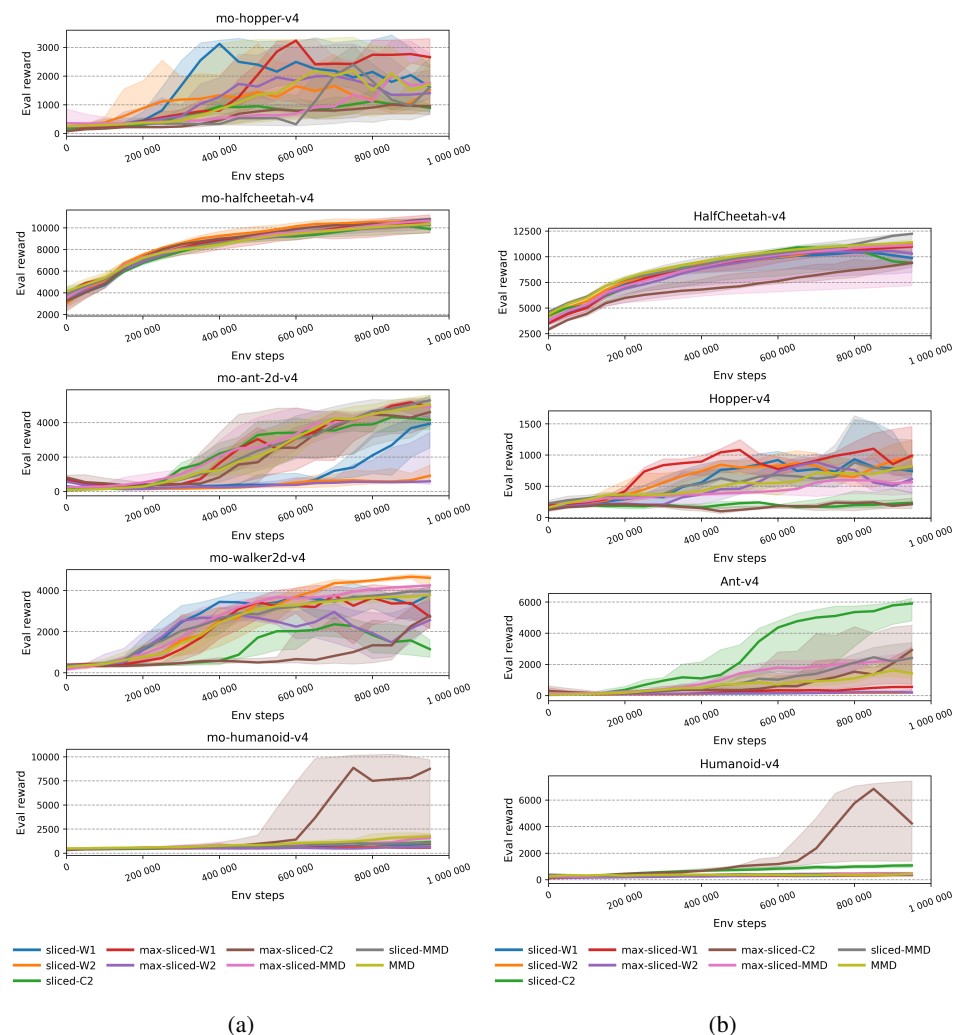

Figure 1: Evaluation of SDRL and MSDRL on multi-objective MuJoco environments and a multi-horizon setting from (Fedus et al., 2019). Results are reported over 5 random seeds with median and 95% bootstrap confidence intervals. (a) Experiments on multi-objective distributional RL with usual fixed scalarization rule (b) Distributional multi-horizon experiments using hyperpolic discounting (Fedus et al., 2019) as scalarization rule. Most variants seem capable to reach or sometimes beat the baseline which is MMD.

We believe the theoretical results can be extended to other base divergences. Moreover, although we specialized our discussion of MMD to a single kernel, this choice could be generalized. We are far from having explored the full potential of slicing, which has seen many improvements and suggestions over the years (Kolouri et al., 2019a; Rowland et al., 2019b). Some of these, such as amortization techniques for max-slicing optimization (Nguyen et al., 2020a), might further benefit the methods we proposed while preserving contraction guarantees. Finally, true multi-objective control has been outside the scope of this work, but a natural application would be to learn control policies across several scalarization rules.

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

APPENDIX CONTENTS

# A  SPECIFIC PROBLEM EXAMPLES

## A.1  MULTIHORIZON RL AND DISTRIBUTIONAL GENERALIZATION

The work of Fedus et al. (2019) instantiates the idea of *multihorizon reinforcement learning*: instead of a single discount factor $\gamma$, the agent simultaneously learns value functions over a family of discounts $\{\gamma_i\}_{i=1}^d$. This multihead architecture provides auxiliary benefits and can approximate non-exponential discounting schemes such as hyperbolic discounting.

**Multihorizon temporal difference.**  We generalize this approach by introducing a vector of discounted returns. Concretely, let $\Gamma = \mathrm{diag}(\gamma_1, \ldots, \gamma_d)$ be a diagonal matrix of discount factors. The expected multihorizon value is

$$Q^\pi(s, a) \;=\; \mathbb{E}\left[\sum_{t=0}^{\infty} \Big(\prod_{k=1}^{t} \Gamma\Big) R_t \;\Bigg|\; S_0 = s, A_0 = a\right],$$

where the product of diagonal matrices $\prod_{k=1}^{t} \Gamma = \Gamma^t$ simply raises each $\gamma_j$ to the $t$-th power. The corresponding Bellman operator is

$$(\mathcal{T}^\pi Q)(s, a) \;=\; R(s, a) + \Gamma\, \mathbb{E}_{S', A'}[\, Q(S', A') \,],$$

with $\Gamma$ a diagonal matrix. This formulation makes explicit that each component corresponds to a distinct effective horizon, while being learned jointly.

**Distributional multihorizon returns.**  We further lift this idea to the distributional setting. Let $Z^\pi(s, a)$ denote the full random return vector,

$$Z^\pi(s, a) \;\overset{D}{=}\; R(s, a) + \Gamma\, Z^\pi(S', A'), \quad A' \sim \pi(\cdot|S'),\; S' \sim P(\cdot|s, a).$$

Here $\Gamma$ remains diagonal, and the recursion models the entire vector distribution rather than only its expectation. This connects multi-horizon temporal-difference learning with distributional RL.

**Scalarization rule.**  We scalarize the multihorizon estimates via a hyperbolic weighting over exponentially discounted heads. For $k > 0$, let $w(\gamma) = \frac{1}{k}\, \gamma^{1/k - 1}$ on $\gamma \in (0, 1]$. Define the hyperbolic scalar value

$$Q_{\mathrm{hyp}}^\pi(s, a) \;=\; \int_0^1 w(\gamma)\, Q_\gamma^\pi(s, a)\, d\gamma,$$

and its practical Riemann approximation over a grid $\mathcal{G} = \{\gamma_0 < \cdots < \gamma_n\}$:

$$\widehat{Q}_{\mathrm{hyp}}^\pi(s, a) \;=\; \sum_{i=0}^{n-1} (\gamma_{i+1} - \gamma_i)\, w(\gamma_i)\, Q_{\gamma_i}^\pi(s, a).$$

*Implementation with an N-head critic.* We fix a grid $\mathcal{G}$ of size $N$ and train a critic with $N$ outputs, where head $i$ uses the *exponential Bellman discount* $(\gamma_i)^k$ and estimates $Q_{(\gamma_i)^k}^\pi(s, a)$. The scalarized value is then the left Riemann sum over the *integration variable* $\gamma$:

$$\widehat{Q}_{\mathrm{hyp}}^\pi(s, a) \;=\; \sum_{i=0}^{n-1} (\gamma_{i+1} - \gamma_i)\, Q_{(\gamma_i)^k}^\pi(s, a),$$

At the distributional level, define

$$Z_{\mathrm{hyp}}^\pi(s, a) \;\overset{D}{=}\; \int_0^1 w(\gamma)\, Z_\gamma^\pi(s, a)\, d\gamma, \qquad \widehat{Z}_{\mathrm{hyp}}^\pi(s, a) \;\overset{D}{=}\; \sum_{i=0}^{n-1} (\gamma_{i+1} - \gamma_i)\, w(\gamma_i)\, Z_{\gamma_i}^\pi(s, a).$$

By linearity of expectation, $\mathbb{E}[Z_{\mathrm{hyp}}^\pi(s, a)] = Q_{\mathrm{hyp}}^\pi(s, a)$ and $\mathbb{E}[\widehat{Z}_{\mathrm{hyp}}^\pi(s, a)] = \widehat{Q}_{\mathrm{hyp}}^\pi(s, a)$. In practice, we use $\widehat{Q}_{\mathrm{hyp}}^\pi$ as the scalar critic in the policy-gradient update; for deterministic policies:

$$\nabla_\theta J(\theta) \;\approx\; \mathbb{E}_s\Big[\nabla_a \widehat{Q}_{\mathrm{hyp}}^\pi(s, a)\big|_{a=\pi_\theta(s)} \nabla_\theta \pi_\theta(s)\Big].$$

## A.2 GENERALIZED VALUE FUNCTIONS (GVFs)

GVFs extend value prediction beyond reward by replacing the reward with a generic *cumulant* and allowing a state/action/transition–dependent *continuation* (discount) (Sutton et al., 2011). In our notation, this is exactly the matrix–discounted setting.

**Definition.** Let $c : \mathcal{S} \times \mathcal{A} \to \mathbb{R}^d$ be a (vector) cumulant and let the continuation be a (possibly dense, time–varying) matrix $\Gamma_t \in \mathbb{R}^{d \times d}$. The GVF action–value is

$$Q_c^\pi(s,a) \;=\; \mathbb{E}\left[ \sum_{t=0}^{\infty} \Big( \prod_{k=1}^{t} \Gamma_k \Big) c(S_t, A_t) \,\Big|\, S_0{=}s,\, A_0{=}a \right], \quad \prod_{k=1}^{0} \Gamma_k := I_d.$$

**Bellman form.** The corresponding (expected) Bellman operator is
$$(\mathcal{T}_c^\pi Q)(s,a) \;=\; c(s,a) + \Gamma_1 \, \mathbb{E}_{S',A'}\big[\, Q(S', A') \,\big],$$
and in the distributional case
$$(\mathcal{T}_c^\pi Z)(s,a) \;\overset{D}{=}\; c(s,a) + \Gamma_1 \, Z(S', A'), \quad A' \sim \pi(\cdot|S'),\; S' \sim P(\cdot|s,a).$$

**Cumulants.** The cumulant $c$ can represent not only the reward but any signal of interest, such as state features, event indicators, or sensor readings.

## B BASE PROBABILITY DIVERGENCES

### B.1 WASSERSTEIN DISTANCE

The Wasserstein distance, arising from optimal transport theory (Villani et al., 2008), provides a principled way of comparing probability measures by quantifying the minimal cost of transporting mass from one distribution to another. Let $(\mathbb{R}^d, d)$ be a metric space and denote by $\mathcal{P}_p(\mathbb{R}^d)$ the set of Borel probability measures with finite $p$-th moment. For $\mu, \nu \in \mathcal{P}_p(\mathbb{R}^d)$, the $p$-Wasserstein distance is defined as

$$W_p(\mu, \nu) = \left( \inf_{\pi \in \Pi(\mu,\nu)} \int_{\mathbb{R}^d \times \mathbb{R}^d} d(x,y)^p \, d\pi(x,y) \right)^{1/p}, \tag{16}$$

where $\Pi(\mu, \nu)$ denotes the set of couplings (or transport plans) $\pi$ whose marginals are $\mu$ and $\nu$. When the underlying measures admit densities $I_\mu$ and $I_\nu$, we may write $W_p(I_\mu, I_\nu)$ without ambiguity.

Computing $W_p$ directly is challenging in high dimensions, but there are settings where closed-form expressions exist. In the special case where $\mu$ and $\nu$ are one-dimensional distributions on a normed linear space, the Wasserstein distance simplifies to

$$W_p(\mu, \nu) = \left( \int_0^1 \left| F_\mu^{-1}(z) - F_\nu^{-1}(z) \right|^p dz \right)^{1/p}, \tag{17}$$

where $F_\mu^{-1}$ and $F_\nu^{-1}$ are the quantile functions (inverse CDFs) of $\mu$ and $\nu$, respectively.

#### B.1.1 ESTIMATOR

For empirical measures $\tilde{\mu} = \frac{1}{N} \sum_{n=1}^{N} \delta_{x_n}$ and $\tilde{\nu} = \frac{1}{N} \sum_{n=1}^{N} \delta_{y_n}$ in one dimension, $W_p$ can be computed by sorting the samples and comparing corresponding order statistics (Villani et al., 2008):

$$W_p(\tilde{\mu}, \tilde{\nu}) = \left( \frac{1}{N} \sum_{n=1}^{N} \left| x_{I_x[n]} - y_{I_y[n]} \right|^p \right)^{1/p}, \tag{18}$$

where $I_x[n]$ and $I_y[n]$ are the indices that sort $\{x_n\}$ and $\{y_n\}$ in ascending order.

#### B.1.2 PROPERTIES

**Metric** It is a classical result that the Wasserstein distances are genuine metrics. In particular, Proposition 2 in Givens & Shortt (1984) establishes that
$$W_p \quad \text{is a metric on } \mathcal{P}_p(\mathbb{R}) \qquad \text{for every } p \in [1, \infty],$$
where $\mathcal{P}_p(\mathbb{R}) = \{ \mu \in \mathcal{P}(\mathbb{R}) : \int |x|^p \, d\mu(x) < \infty \}$ for $p < \infty$, and $\mathcal{P}_\infty(\mathbb{R}) = \mathcal{P}(\mathbb{R})$.

**Translation invariant**  By definition

**Scale-Lipschitz**

**Proposition 1** (Exact scaling under deterministic multiplication for $W_p$, $p \in [1, \infty)$. ] *Let $(X, \|\cdot\|)$ be a normed vector space with metric $d(x, y) = \|x - y\|$, let $S_s : X \to X$ be the dilation $S_s(x) = s\,x$ with $s \geq 0$, and let $p \in [1, \infty]$. If $p < \infty$, assume $\mu, \nu \in \mathcal{P}_p(X)$; if $p = \infty$, assume $W_\infty(\mu, \nu) < \infty$ (e.g. $\mu, \nu$ have compact support). Then*

$$W_p\big((S_s)_\# \mu,\, (S_s)_\# \nu\big) = s\, W_p(\mu, \nu).$$

*Proof.* If $s = 0$ then $(S_0)_\# \mu = (S_0)_\# \nu = \delta_0$, so both sides are $0$ and the statement holds. In the remainder assume $s > 0$.

**Case** $1 \leq p < \infty$. Define $\Phi_s : \Pi(\mu, \nu) \to \Pi\big((S_s)_\# \mu, (S_s)_\# \nu\big)$ by

$$\Phi_s(\pi) := (S_s \times S_s)_\# \pi.$$

Then $\Phi_s$ is a bijection with inverse $\Phi_{1/s}$, since $(S_{1/s})_\#(S_s)_\# \mu = \mu$ and similarly for $\nu$. Therefore,

$$\begin{aligned}
W_p\big((S_s)_\# \mu, (S_s)_\# \nu\big)^p &= \inf_{\pi' \in \Pi((S_s)_\# \mu, (S_s)_\# \nu)} \int d(u, v)^p \, d\pi'(u, v) \\
&= \inf_{\pi \in \Pi(\mu, \nu)} \int d\big(S_s x, S_s y\big)^p \, d\pi(x, y) \\
&= \inf_{\pi \in \Pi(\mu, \nu)} s^p \int d(x, y)^p \, d\pi(x, y) \\
&= s^p \, W_p(\mu, \nu)^p.
\end{aligned}$$

Taking $p$th roots gives the claim for $p < \infty$.

**Case** $p = \infty$. By definition,

$$W_\infty(\mu, \nu) = \inf_{\pi \in \Pi(\mu, \nu)} \sup_{(x,y) \in \mathrm{supp}(\pi)} d(x, y).$$

As above, $\Phi_s$ is a bijection between $\Pi(\mu, \nu)$ and $\Pi\big((S_s)_\# \mu, (S_s)_\# \nu\big)$. Hence

$$\begin{aligned}
W_\infty\big((S_s)_\# \mu, (S_s)_\# \nu\big) &= \inf_{\pi' \in \Pi((S_s)_\# \mu, (S_s)_\# \nu)} \sup_{(u,v) \in \mathrm{supp}(\pi')} d(u, v) \\
&= \inf_{\pi \in \Pi(\mu, \nu)} \sup_{(x,y) \in \mathrm{supp}(\pi)} d\big(S_s x, S_s y\big) \\
&= \inf_{\pi \in \Pi(\mu, \nu)} \sup_{(x,y) \in \mathrm{supp}(\pi)} s\, d(x, y) \\
&= s\, W_\infty(\mu, \nu).
\end{aligned}$$

This proves the claim for $p = \infty$. $\qquad\square$

**p-convexity**

**Proposition 2** (Mixture $p$-convexity for $W_p$). *Let $(X, d)$ be a metric space, $p \in [1, \infty)$, and let $(\Omega, \mathcal{F}, \rho)$ be a probability space. Let $(\mu_c)_{c \in \Omega}, (\nu_c)_{c \in \Omega} \subset \mathcal{P}_p(X)$ be measurable families. Then*

$$W_p\Big( \int_\Omega \mu_c \, \rho(dc), \int_\Omega \nu_c \, \rho(dc) \Big) \leq \left( \int_\Omega W_p(\mu_c, \nu_c)^p \, \rho(dc) \right)^{1/p}.$$

*Proof.* **Step 1: $\varepsilon$-optimal couplings for each $c$.**
Fix $\varepsilon > 0$. For each $c \in \Omega$, pick an $\varepsilon$-optimal coupling $\pi_c^\varepsilon \in \Pi(\mu_c, \nu_c)$ such that

$$\int_{X \times X} d(x, y)^p \, \pi_c^\varepsilon(dx, dy) \leq W_p(\mu_c, \nu_c)^p + \varepsilon.$$

**Step 2: Measurable selection and mixed coupling.**

Assume the family $(\pi_c^\varepsilon)_{c \in \Omega}$ can be chosen measurably, so that $c \mapsto \pi_c^\varepsilon$ is a probability kernel. We then define the mixed coupling

$$\Pi^\varepsilon(U) := \int_\Omega \pi_c^\varepsilon(U)\,\rho(dc), \qquad U \subseteq X \times X \text{ Borel.}$$

For any measurable $A \subseteq X$,

$$\Pi^\varepsilon(A \times X) = \int_\Omega \pi_c^\varepsilon(A \times X)\,\rho(dc) = \int_\Omega \mu_c(A)\,\rho(dc) = \Big( \int_\Omega \mu_c\,\rho(dc) \Big)(A),$$

and similarly

$$\Pi^\varepsilon(X \times A) = \int_\Omega \pi_c^\varepsilon(X \times A)\,\rho(dc) = \int_\Omega \nu_c(A)\,\rho(dc) = \Big( \int_\Omega \nu_c\,\rho(dc) \Big)(A).$$

Hence $\Pi^\varepsilon$ has the mixed marginals $\int_\Omega \mu_c\,\rho(dc)$ and $\int_\Omega \nu_c\,\rho(dc)$, i.e.

$$\Pi^\varepsilon \in \Pi\Big( \int_\Omega \mu_c\,\rho(dc),\ \int_\Omega \nu_c\,\rho(dc) \Big).$$

**Step 3: Bound the transport cost of the mixed coupling.**

Since $(c, x, y) \mapsto d(x, y)^p$ is nonnegative and measurable and $c \mapsto \pi_c^\varepsilon$ is a probability kernel, Tonelli's theorem allows us to exchange the order of integration in $(c, x, y)$:

$$\begin{aligned}
\int_{X \times X} d(x, y)^p\,\Pi^\varepsilon(dx, dy) &= \int_{X \times X} d(x, y)^p \Big( \int_\Omega \pi_c^\varepsilon(dx, dy)\,\rho(dc) \Big) \\
&= \int_\Omega \Big( \int_{X \times X} d(x, y)^p\,\pi_c^\varepsilon(dx, dy) \Big)\,\rho(dc) \\
&\leq \int_\Omega \big( W_p(\mu_c, \nu_c)^p + \varepsilon \big)\,\rho(dc) \\
&= \int_\Omega W_p(\mu_c, \nu_c)^p\,\rho(dc)\ +\ \varepsilon.
\end{aligned}$$

**Step 4: Take the infimum over couplings and pass to the limit.**

By definition of $W_p$,

$$W_p\Big( \int \mu_c\,d\rho,\ \int \nu_c\,d\rho \Big)^p \leq \int_{X \times X} d(x, y)^p\,\Pi^\varepsilon(dx, dy) \leq \int_\Omega W_p(\mu_c, \nu_c)^p\,\rho(dc) + \varepsilon.$$

Taking $p$th roots and letting $\varepsilon \downarrow 0$ yields

$$W_p\Big( \int_\Omega \mu_c\,\rho(dc),\ \int_\Omega \nu_c\,\rho(dc) \Big) \leq \Big( \int_\Omega W_p(\mu_c, \nu_c)^p\,\rho(dc) \Big)^{1/p}.$$

$\square$

### B.2 THE $\ell_p$–FAMILY OF CDF DISTANCES ON $\mathbb{R}$

Let $\mu, \nu \in \mathcal{P}(\mathbb{R})$ be probability measures with cumulative distribution functions (CDFs) $F_\mu, F_\nu$. For $p \in [1, \infty)$, the $\ell_p$ distance between $\mu$ and $\nu$ is defined as

$$\ell_p(\mu, \nu) := \Big( \int_{-\infty}^\infty \big| F_\mu(t) - F_\nu(t) \big|^p\,dt \Big)^{1/p} = \| F_\mu - F_\nu \|_{L^p(\mathbb{R})},$$

that is, the $\ell_p$–family can be seen as the $L^p$ norm between the two CDFs (Bellemare et al., 2017b).

**Connections to other distances**

- **Wasserstein distance.** For $p = 1$, one recovers the 1–Wasserstein distance (Bellemare et al., 2017b):

$$\ell_1(\mu, \nu) = W_1(\mu, \nu) = \int_0^1 \left| F_\mu^{-1}(u) - F_\nu^{-1}(u) \right| du.$$

  Thus, $\ell_1$ coincides with the classical earth mover's distance on $\mathbb{R}$.

- **Cramér distance.** For $p = 2$, the squared $\ell_2$ distance coincides with the Cramér distance (Bellemare et al., 2017b):

$$\ell_2^2(\mu, \nu) = \int_{-\infty}^{\infty} \left( F_\mu(t) - F_\nu(t) \right)^2 dt,$$

  which also admits the energy distance form

$$\ell_2^2(\mu, \nu) = \mathbb{E}|X - Y| - \tfrac{1}{2}\mathbb{E}|X - X'| - \tfrac{1}{2}\mathbb{E}|Y - Y'|,$$

  for $X, X' \sim \mu$ and $Y, Y' \sim \nu$ i.i.d.

### B.2.1 ESTIMATOR

For empirical measures $\tilde{\mu} = \frac{1}{n} \sum_{i=1}^n \delta_{u_i}$ and $\tilde{\nu} = \frac{1}{m} \sum_{j=1}^m \delta_{v_j}$ in one dimension, the $\ell_p$ CDF distance (Cramér when $p = 2$) admits a closed form: after merging and sorting all samples, one tracks the cumulative difference of the two empirical CDFs, which is piecewise constant between successive breakpoints. The distance then reduces to a weighted sum of gap lengths multiplied by the corresponding powers of this difference.

$$\ell_p^p(\tilde{\mu}, \tilde{\nu}) = \sum_{k=1}^{K-1} (t_{k+1} - t_k) |\Delta_k|^p.$$

Algorithmically, the estimator amounts to sorting the combined samples once, tracking the cumulative difference of the two empirical CDFs, and summing the piecewise contributions. This requires $\mathcal{O}((n+m) \log(n+m))$ time for sorting and linear time for the scan.

### B.2.2 PROPERTIES

**Metric**

**Proposition 3** (Metric property of the $\ell_p$ CDF distance). *Let $\mu, \nu \in \mathcal{P}(\mathbb{R})$ have CDFs $F_\mu, F_\nu$. For $p \in [1, \infty)$ define*

$$\ell_p(\mu, \nu) := \left\| F_\mu - F_\nu \right\|_{L^p(\mathbb{R})} = \left( \int_{\mathbb{R}} \left| F_\mu(t) - F_\nu(t) \right|^p dt \right)^{1/p}.$$

*Let $\mathcal{P}_1(\mathbb{R}) := \{ \xi \in \mathcal{P}(\mathbb{R}) : \int_{\mathbb{R}} |x| \, d\xi(x) < \infty \}$. Then for every $p \in [1, \infty)$, $\ell_p$ is a metric on $\mathcal{P}_1(\mathbb{R})$.*

*Proof.* **Finiteness on $\mathcal{P}_1(\mathbb{R})$.**
In one dimension, $\ell_1(\mu, \nu) = \int_{\mathbb{R}} |F_\mu - F_\nu| \, dt = W_1(\mu, \nu)$, hence $\ell_1(\mu, \nu) < \infty$ for $\mu, \nu \in \mathcal{P}_1(\mathbb{R})$. For $p > 1$, since $0 \le |F_\mu - F_\nu| \le 1$,

$$\ell_p(\mu, \nu)^p = \int |F_\mu - F_\nu|^p \, dt \le \int |F_\mu - F_\nu| \, dt = \ell_1(\mu, \nu) < \infty.$$

**Nonnegativity and symmetry.**
By definition, $\ell_p(\mu, \nu) = \|F_\mu - F_\nu\|_{L^p} \ge 0$ and $\ell_p(\mu, \nu) = \|F_\mu - F_\nu\|_{L^p} = \|F_\nu - F_\mu\|_{L^p} = \ell_p(\nu, \mu)$.

**Identity of indiscernibles.**
If $\ell_p(\mu, \nu) = 0$, that means

$$\int |F_\mu(x) - F_\nu(x)|^p \, dx = 0.$$

An $L^p$ norm is zero iff the functions are equal almost everywhere. So $F_\mu = F_\nu$ except maybe on a measure zero set. Now, CDFs are monotone and right–continuous. Such functions cannot differ only on a measure zero set, if they are different at one point, they must differ on an interval of positive length. So "equal almost everywhere" forces them to be equal everywhere. If the CDFs are identical, then the distributions are the same.

**Triangle inequality.**
We use Minkowski's inequality in $L^p(\mathbb{R})$. Writing $F_\mu - F_\lambda = (F_\mu - F_\nu) + (F_\nu - F_\lambda)$, we obtain

$$
\begin{aligned}
\ell_p(\mu, \lambda) &= \|F_\mu - F_\lambda\|_{L^p} \\
&= \|(F_\mu - F_\nu) + (F_\nu - F_\lambda)\|_{L^p} \\
&\leq \|F_\mu - F_\nu\|_{L^p} + \|F_\nu - F_\lambda\|_{L^p}. \qquad \text{(Minkowski)}
\end{aligned}
$$

Therefore $\ell_p(\mu, \lambda) \leq \ell_p(\mu, \nu) + \ell_p(\nu, \lambda)$. $\qquad\qquad\square$

**Translation invariant**  By non-trivial arguments (see Theorem 2 in Bellemare et al. (2017b) and Proposition 3.2 in Odin & Charpentier (2020)), the $\ell_p$ distance is invariant under translations: for all $\mu, \nu \in \mathcal{P}_1(\mathbb{R})$ and every $t \in \mathbb{R}$,

$$
\ell_p\big((T_t)_\#\mu, (T_t)_\#\nu\big) = \ell_p(\mu, \nu), \qquad T_t(x) = x + t.
$$

**Scale-Lipschitz**

**Proposition 4** (Scale–Lipschitz property of the $\ell_p$ CDF distance). *Let $\mu, \nu \in \mathcal{P}_1(\mathbb{R})$ have CDFs $F_\mu, F_\nu$. For $p \in [1, \infty)$ and $S_s(x) = s\,x$ with $s \geq 0$, the $\ell_p$ distance satisfies*

$$
\ell_p\big((S_s)_\#\mu, (S_s)_\#\nu\big) \leq c(s)\,\ell_p(\mu, \nu), \qquad c(s) := s^{1/p}.
$$

*Proof.* **Scale–sensitivity via change of variables.**
Let $\gamma \in \mathbb{R}^*$. Using $F_{(S_\gamma)_\#\mu}(x) = F_\mu(x/\gamma)$, we compute

$$
\begin{aligned}
\ell_p\big((S_\gamma)_\#\mu, (S_\gamma)_\#\nu\big) &= \left( \int_\mathbb{R} \left| F_\mu(x/\gamma) - F_\nu(x/\gamma) \right|^p dx \right)^{1/p} \\
&= \left( \int_\mathbb{R} \left| F_\mu(u) - F_\nu(u) \right|^p |\gamma|\, du \right)^{1/p} \qquad \text{(C.V. } u = x/\gamma) \\
&= |\gamma|^{1/p} \left( \int_\mathbb{R} \left| F_\mu(u) - F_\nu(u) \right|^p du \right)^{1/p} \\
&= |\gamma|^{1/p}\, \ell_p(\mu, \nu).
\end{aligned}
$$

**Conclusion (Scale–Lipschitz).**
For $s \geq 0$, the above identity gives

$$
\ell_p\big((S_s)_\#\mu, (S_s)_\#\nu\big) = s^{1/p}\, \ell_p(\mu, \nu) \leq c(s)\,\ell_p(\mu, \nu) \quad \text{with } c(s) := s^{1/p},
$$

which is the desired scale–Lipschitz property. $\qquad\qquad\square$

**p-convexity**

**Proposition 5** (Mixture $p$-convexity for $\ell_p$ (integral form)). *Let $(\Omega, \mathcal{F}, \rho)$ be a probability space, $p \in [1, \infty)$, and let $(\mu_c)_{c \in \Omega}, (\nu_c)_{c \in \Omega} \subset \mathcal{P}_1(\mathbb{R})$ be measurable families with CDFs $(F_{\mu_c}), (F_{\nu_c})$. Then*

$$
\ell_p\Big( \int_\Omega \mu_c\, \rho(dc), \int_\Omega \nu_c\, \rho(dc) \Big) \leq \left( \int_\Omega \ell_p(\mu_c, \nu_c)^p\, \rho(dc) \right)^{1/p}.
$$

*Proof.* **CDF linearity under mixtures.**

For every $x \in \mathbb{R}$,

$$F_{\int \mu_c \, d\rho}(x) \;=\; \int_\Omega F_{\mu_c}(x) \, \rho(dc), \qquad F_{\int \nu_c \, d\rho}(x) \;=\; \int_\Omega F_{\nu_c}(x) \, \rho(dc).$$

Hence

$$F_{\int \mu_c \, d\rho}(x) - F_{\int \nu_c \, d\rho}(x) \;=\; \int_\Omega \big( F_{\mu_c}(x) - F_{\nu_c}(x) \big) \, \rho(dc). \tag{19}$$

**Jensen inside the $x$–integral.**

Since $|\cdot|^p$ is convex and $\rho$ is a probability measure,

$$\left| \int_\Omega \big( F_{\mu_c}(x) - F_{\nu_c}(x) \big) \, \rho(dc) \right|^p \;\leq\; \int_\Omega \big| F_{\mu_c}(x) - F_{\nu_c}(x) \big|^p \, \rho(dc) \qquad \text{(Jensen on } \Omega\text{)}.$$

**Integrate over $x$ and swap the order.**

Therefore

$$\begin{aligned}
\ell_p\!\left( \int \mu_c \, d\rho, \int \nu_c \, d\rho \right)^p &= \int_\mathbb{R} \big| F_{\int \mu_c \, d\rho}(x) - F_{\int \nu_c \, d\rho}(x) \big|^p \, dx \quad \text{(def. of } \ell_p\text{)} \\
&= \int_\mathbb{R} \left| \int_\Omega \big( F_{\mu_c}(x) - F_{\nu_c}(x) \big) \, \rho(dc) \right|^p \, dx \quad \text{(by equation 19)} \\
&\leq \int_\mathbb{R} \int_\Omega \big| F_{\mu_c}(x) - F_{\nu_c}(x) \big|^p \, \rho(dc) \, dx \quad \text{(Jensen on } \Omega\text{)} \\
&= \int_\Omega \int_\mathbb{R} \big| F_{\mu_c}(x) - F_{\nu_c}(x) \big|^p \, dx \, \rho(dc) \quad \text{(Fubini–Tonelli)} \\
&= \int_\Omega \ell_p(\mu_c, \nu_c)^p \, \rho(dc).
\end{aligned}$$

Taking the $p$-th root yields the claim. $\qquad\square$

## B.3 MMD

The Maximum Mean Discrepancy (MMD) is a kernel-based discrepancy that measures how far apart two probability laws are in a reproducing kernel Hilbert space (RKHS) $\mathcal{H}$. Given a symmetric kernel $k\colon X \times X \to \mathbb{R}$ with feature map $\phi(x) = k(x, \cdot)$, each distribution admits a *mean embedding* in $\mathcal{H}$:

$$\mu_P = \int_X \phi(x) \, dP(x), \qquad \mu_Q = \int_X \phi(x) \, dQ(x).$$

The distance between these embeddings defines

$$\mathbf{MMD}_k(P, Q) \;=\; \| \mu_P - \mu_Q \|_\mathcal{H}.$$

Its square can be expanded in terms of expectations of the kernel:

$$\begin{aligned}
\mathbf{MMD}_k^2(P, Q) &= \| \mu_P - \mu_Q \|_\mathcal{H}^2 \tag{20} \\
&= \iint k(x, x') \, dP(x) \, dP(x') \;+\; \iint k(y, y') \, dQ(y) \, dQ(y') \\
&\quad - 2 \iint k(x, y) \, dP(x) \, dQ(y).
\end{aligned}$$

**Definition 2** (Conditionally positive definite (CPD) kernel — integral form)**.** *Let $X$ be a measurable space and let $k : X \times X \to \mathbb{R}$ be symmetric. We say that $k$ is* conditionally positive definite (CPD) *if*

$$\iint_{X \times X} k(x, x') \, d\mu(x) \, d\mu(x') \;\geq\; 0 \qquad \text{for all finite signed measures } \mu \text{ on } X \text{ with } \mu(X) = 0.$$

*If the inequality is strict for every nonzero such $\mu$, then $k$ is* conditionally strictly positive definite *(CSPD).*

### B.3.1 ESTIMATOR

The **MMD** can be approximated from samples in two standard ways, both originating from Gretton et al. (2012). Given two sets of $m$ samples $\{x_i\}_{i=1}^m \sim P$ and $\{y_i\}_{i=1}^m \sim Q$, the *biased* estimator is

$$\widehat{\text{MMD}}_b^2 = \frac{1}{m^2} \sum_{i,j=1}^m k(x_i, x_j) + \frac{1}{m^2} \sum_{i,j=1}^m k(y_i, y_j) - \frac{2}{m^2} \sum_{i,j=1}^m k(x_i, y_j). \tag{21}$$

while the *unbiased* estimator excludes diagonal terms:

$$\widehat{\text{MMD}}_u^2 = \frac{1}{m(m-1)} \sum_{\substack{i,j=1 \\ i \neq j}}^m k(x_i, x_j) + \frac{1}{m(m-1)} \sum_{\substack{i,j=1 \\ i \neq j}}^m k(y_i, y_j) - \frac{2}{m^2} \sum_{i,j=1}^m k(x_i, y_j). \tag{22}$$

Although the unbiased form eliminates a small finite-sample bias, the biased estimator is often preferred in practice. In particular, applications of **MMD** to distributional RL (Nguyen et al., 2020b; Killingberg & Langseth, 2023b) consistently rely on the biased version due to its lower variance and greater numerical stability during training.

### B.3.2 PROPERTIES

**Metric**

**Proposition 6** (Equivalence of $\gamma_k$ and RKHS–MMD for CPD kernels). *Let* $k : X \times X \to \mathbb{R}$ *be conditionally positive definite (CPD) and define*

$$\rho_k(x, y) := k(x, x) + k(y, y) - 2k(x, y).$$

*Fix* $z_0 \in X$ *and set the* distance–induced *(one–point centered) kernel*

$$k^\circ(x, y) := \tfrac{1}{2}\big[\rho_k(x, z_0) + \rho_k(y, z_0) - \rho_k(x, y)\big] = k(x, y) - k(x, z_0) - k(z_0, y) + k(z_0, z_0).$$

*Then* $k^\circ$ *is positive definite and admits an RKHS* $\mathcal{H}_{k^\circ}$. *For any* $P, Q$ *with finite integrals,*

$$
\begin{aligned}
\gamma_k(P, Q)^2 &:= \iint k(x, y)\, d(P - Q)(x)\, d(P - Q)(y) \\
&= \iint k^\circ(x, y)\, d(P - Q)(x)\, d(P - Q)(y) \\
&= \big\| \mu_{k^\circ}(P) - \mu_{k^\circ}(Q) \big\|_{\mathcal{H}_{k^\circ}}^2 \\
&= \text{MMD}_{k^\circ}(P, Q)^2.
\end{aligned}
$$

*Justification.* *This follows from the distance–induced kernel construction and equivalence results in Sejdinovic et al. (2013).*

**Proposition 7** (MMD as a Metric on $\mathcal{P}(X)$). *Let* $k \colon X \times X \to \mathbb{R}$ *be a symmetric kernel. We say that* $\text{MMD}_k$ *defines a metric on* $\mathcal{P}(X)$ *iff* $k$ *is conditionally strictly positive definite (CSPD), i.e., for every nonzero finite signed Borel measure* $\nu$ *with* $\nu(X) = 0$,

$$\iint_{X \times X} k(x, y)\, d\nu(x)\, d\nu(y) > 0.$$

*Then* $\text{MMD}_k$ *satisfies the metric axioms on* $\mathcal{P}(X)$:

1. Nonnegativity: $\text{MMD}_k(P, Q) \geq 0$.

2. Symmetry: $\text{MMD}_k(P, Q) = \text{MMD}_k(Q, P)$.

3. Identity of indiscernibles: $\text{MMD}_k(P, Q) = 0 \Rightarrow P = Q$.

4. Triangle inequality: *for any* $P, Q, R \in \mathcal{P}(X)$, $\text{MMD}_k(P, Q) \leq \text{MMD}_k(P, R) + \text{MMD}_k(R, Q)$.

*Justification.* *This is the standard correspondence between negative-type distances, distance-induced kernels, and RKHS MMD metrics as outlined in Sejdinovic et al. (2013).*

**Scale-Lipschitz**

**Proposition 8** (Scale–Lipschitz property of (squared) MMD with MQ kernel). *Let* $k_h(x,y) = -\sqrt{1 + h^2\|x - y\|^2}$ *with* $h > 0$.

*For probability measures* $\mu, \nu$ *on* $\mathbb{R}^d$ *with finite second moments, define the population* $\mathrm{MMD}^2$ *by*
$$\mathrm{MMD}^2_{k_h}(\mu, \nu) = \mathbb{E}\, k_h(X, X') + \mathbb{E}\, k_h(Y, Y') - 2\,\mathbb{E}\, k_h(X, Y),$$
*for* $X, X' \sim \mu$ *i.i.d. and* $Y, Y' \sim \nu$ *i.i.d.*

*For the scaling map* $S_s : x \mapsto sx$ *with* $s \geq 0$*, we have*
$$\mathrm{MMD}^2_{k_h}\big((S_s)_\#\mu, (S_s)_\#\nu\big) \leq c_2(s)\,\mathrm{MMD}^2_{k_h}(\mu, \nu), \qquad c_2(s) := \max\{\, s,\ s^2\, \}.$$

*Consequently, the (unsquared) MMD satisfies*
$$\mathrm{MMD}_{k_h}\big((S_s)_\#\mu, (S_s)_\#\nu\big) \leq c_1(s)\,\mathrm{MMD}_{k_h}(\mu, \nu), \qquad c_1(s) := \max\{\, \sqrt{s},\ s\, \}.$$

*In particular, for* $s < 1$ *the map* $S_s$ *is a contraction for both* $\mathrm{MMD}^2_{k_h}$ *and* $\mathrm{MMD}_{k_h}$.

*Proof.* Set
$$\phi(r) = \sqrt{1 + h^2 r^2}.$$
With this notation,
$$\mathrm{MMD}^2_{k_h}(\mu, \nu) = 2\,\mathbb{E}\, \phi(\|X - Y\|) - \mathbb{E}\, \phi(\|X - X'\|) - \mathbb{E}\, \phi(\|Y - Y'\|).$$

When $0 \leq s \leq 1$, note that $\phi(0) = 1$ and $\phi$ is convex, as we have
$$\phi'(r) = \frac{h^2 r}{\sqrt{1 + h^2 r^2}}, \qquad \phi''(r) = \frac{h^2}{(1 + h^2 r^2)^{3/2}} \geq 0.$$
By convexity, for any $a, b \in \mathbb{R}$ and $s \in [0, 1]$,
$$\phi((1-s)a + sb) \leq (1-s)\phi(a) + s\phi(b).$$
Taking $a = 0$, $b = r$, and recalling $\phi(0) = 1$, this gives
$$\phi(sr) \leq (1-s) + s\,\phi(r), \qquad 0 \leq s \leq 1.$$

Applying this inequality inside each expectation, the constants cancel in the linear combination since $(2 - 1 - 1)(1 - s) = 0$. Therefore
$$\mathrm{MMD}^2_{k_h}\big((S_s)_\#\mu, (S_s)_\#\nu\big) \leq s\,\mathrm{MMD}^2_{k_h}(\mu, \nu).$$

When $s \geq 1$, consider $f(u) = \sqrt{1 + h^2 u}$ for $u \geq 0$; it is concave as
$$f'(u) = \frac{h^2}{2\sqrt{1 + h^2 u}}, \qquad f''(u) = -\frac{h^4}{4(1 + h^2 u)^{3/2}} \leq 0.$$
By definition, $\phi(r) = f(r^2)$. For any $u \geq 0$ and $\lambda \geq 1$, concavity gives
$$f(u) = f\Big((1 - \tfrac{1}{\lambda}) \cdot 0 + \tfrac{1}{\lambda} \cdot (\lambda u)\Big) \geq (1 - \tfrac{1}{\lambda})\, f(0) + \tfrac{1}{\lambda}\, f(\lambda u),$$
hence
$$f(\lambda u) \leq \lambda f(u) - (\lambda - 1)f(0).$$
Taking $u = r^2$, $\lambda = s^2$, and recalling that $f(0) = 1$, we obtain
$$\phi(sr) = \sqrt{1 + h^2 s^2 r^2} \leq s^2 \phi(r) - (s^2 - 1).$$

Again inserting this inequality into the definition of $\mathrm{MMD}^2$, the constants cancel as before, and we obtain
$$\mathrm{MMD}^2_{k_h}\big((S_s)_\#\mu, (S_s)_\#\nu\big) \leq s^2\, \mathrm{MMD}^2_{k_h}(\mu, \nu).$$

Combining both cases, the multiplicative factor is $s$ for $0 \leq s \leq 1$ and $s^2$ for $s \geq 1$. Hence
$$c_2(s) = \max\{s,\ s^2\}.$$
Taking square roots gives the corresponding bound for the unsquared MMD,
$$c_1(s) = \max\{\sqrt{s},\ s\}.$$

$\square$

**p-convexity**

**Proposition 9** (Mixture $p$–convexity of $\mathrm{MMD}_k$ in an RKHS). *Let $k : X \times X \to \mathbb{R}$ be a symmetric positive–semidefinite reproducing kernel with RKHS $(\mathcal{H}, \langle \cdot, \cdot \rangle)$ and feature map $\phi(x) = k(x, \cdot)$. Let $(\Omega, \mathcal{F}, \rho)$ be a probability space, and let $(\mu_c)_{c \in \Omega}$ and $(\nu_c)_{c \in \Omega}$ be measurable families of probability measures on $X$ for which the mean embeddings $\mu_{\mu_c} := \int_X \phi \, d\mu_c$ and $\mu_{\nu_c} := \int_X \phi \, d\nu_c$ exist in $\mathcal{H}$. Define the mixtures $\bar{\mu} := \int_\Omega \mu_c \, \rho(dc)$ and $\bar{\nu} := \int_\Omega \nu_c \, \rho(dc)$. Assume all mean embeddings and integrals below are well defined. Then for every $p \geq 1$,*

$$\boxed{\mathrm{MMD}_k(\bar{\mu}, \bar{\nu}) \ \leq \ \left( \int_\Omega \mathrm{MMD}_k(\mu_c, \nu_c)^p \, \rho(dc) \right)^{1/p}}.$$

*Proof.* By linearity of mean embeddings,

$$\mu_{\bar{\mu}} = \int_\Omega \mu_{\mu_c} \, \rho(dc), \qquad \mu_{\bar{\nu}} = \int_\Omega \mu_{\nu_c} \, \rho(dc),$$

where $\mu_{\mu_c} = \int_X \phi(x) \, d\mu_c(x)$ and $\mu_{\nu_c} = \int_X \phi(x) \, d\nu_c(x)$ are elements of $\mathcal{H}$. Thus,

$$\mu_{\bar{\mu}} - \mu_{\bar{\nu}} = \int_\Omega v(c) \, \rho(dc), \qquad v(c) := \mu_{\mu_c} - \mu_{\nu_c} \in \mathcal{H}.$$

Hence

$$\mathrm{MMD}_k(\bar{\mu}, \bar{\nu}) = \| \mu_{\bar{\mu}} - \mu_{\bar{\nu}} \|_\mathcal{H} = \left\| \int_\Omega v(c) \, \rho(dc) \right\|_\mathcal{H}$$

$$\leq \int_\Omega \| v(c) \|_\mathcal{H} \, \rho(dc) \qquad \qquad \text{(triangle inequality in } \mathcal{H})$$

$$\leq \left( \int_\Omega \| v(c) \|_\mathcal{H}^p \, \rho(dc) \right)^{1/p} \qquad \qquad (L^1 \leq L^p \text{ on a probability space}).$$

Finally, $\| v(c) \|_\mathcal{H} = \| \mu_{\mu_c} - \mu_{\nu_c} \|_\mathcal{H} = \mathrm{MMD}_k(\mu_c, \nu_c)$, which gives the claim. $\qquad \square$

**Proposition 10** (Mixture $p$–convexity for CPD kernels via the distance–induced RKHS). *Let $k : X \times X \to \mathbb{R}$ be conditionally positive definite (CPD) and let $k^\circ$ be the associated distance–induced (one–point centered) kernel from Proposition 6, so that for all probabilities $P, Q$ with finite integrals,*

$$\gamma_k(P, Q) = \mathrm{MMD}_{k^\circ}(P, Q).$$

*Let $(\Omega, \mathcal{F}, \rho)$ be a probability space, and let $(\mu_c)_{c \in \Omega}$ and $(\nu_c)_{c \in \Omega}$ be measurable families of probability measures on $X$ with finite embeddings for $k^\circ$. Define the mixtures $\bar{\mu} := \int_\Omega \mu_c \, \rho(dc)$ and $\bar{\nu} := \int_\Omega \nu_c \, \rho(dc)$. Then for every $p \geq 1$,*

$$\boxed{\gamma_k(\bar{\mu}, \bar{\nu}) \ \leq \ \left( \int_\Omega \gamma_k(\mu_c, \nu_c)^p \, \rho(dc) \right)^{1/p}}.$$

*Proof.* By Proposition 6, $\gamma_k = \mathrm{MMD}_{k^\circ}$. Applying Lemma 9 to the PSD kernel $k^\circ$ and the families $(\mu_c), (\nu_c)$ yields

$$\mathrm{MMD}_{k^\circ}(\bar{\mu}, \bar{\nu}) \ \leq \ \left( \int_\Omega \mathrm{MMD}_{k^\circ}(\mu_c, \nu_c)^p \, \rho(dc) \right)^{1/p}.$$

Replacing $\mathrm{MMD}_{k^\circ}$ by $\gamma_k$ via Proposition 6 gives the claim. $\qquad \square$

# C  THEORETICAL RESULTS

## C.1  METRIC PROPERTY

**Lemma 1** (Basic metric properties of slicing (from Nadjahi et al. (2020))). *Let $\Delta : \mathcal{P}(\mathbb{R}) \times \mathcal{P}(\mathbb{R}) \to [0, \infty]$ be a divergence and let $p \in [1, \infty)$. For $\mu, \nu \in \mathcal{P}(\mathbb{R}^d)$ define*

$$\mathbf{S}\Delta_p^p(\mu, \nu) \ = \ \int_{S^{d-1}} \Delta^p \big( (P_\theta)_\# \mu, (P_\theta)_\# \nu \big) \, d\sigma(\theta),$$

*where $(P_\theta)_{\#}\mu$ is the pushforward of $\mu$ by $x \mapsto \langle \theta, x \rangle$ and $\sigma$ is the uniform measure on $S^{d-1}$. This reproduces Proposition 1 Nadjahi et al. (2020).*

**Statement.** *If $\Delta$ is a metric on $\mathcal{P}(\mathbb{R})$, then $\mathbf{S}\Delta_p$ is a metric on $\mathcal{P}(\mathbb{R}^d)$. In particular:*

- **Nonnegativity and symmetry.** *If $\Delta$ is nonnegative (resp. symmetric) on $\mathcal{P}(\mathbb{R})$, then $\mathbf{S}\Delta_p$ is nonnegative (resp. symmetric) on $\mathcal{P}(\mathbb{R}^d)$.*

- **Identity of indiscernibles.** *If $\Delta(\alpha, \beta) = 0$ iff $\alpha = \beta$ for $\alpha, \beta \in \mathcal{P}(\mathbb{R})$, then $\mathbf{S}\Delta_p(\mu, \nu) = 0$ iff $\mu = \nu$ for $\mu, \nu \in \mathcal{P}(\mathbb{R}^d)$.*

- **Triangle inequality.** *If $\Delta$ is a metric on $\mathcal{P}(\mathbb{R})$, then $\mathbf{S}\Delta_p$ satisfies the triangle inequality on $\mathcal{P}(\mathbb{R}^d)$.*

*Proof (this is a reproduction from Nadjahi et al. (2020), App. A.1).* We prove that $\mathbf{S}\Delta_p$ satisfies the three defining properties required for a metric on $\mathcal{P}(\mathbb{R}^d)$.

**Nonnegativity and symmetry.**
This is immediate from the definition since the integrand inherits these properties from $\Delta$, and taking a $p$-th root preserves them.

**Identity of indiscernibles.**
We need to show that $\mathbf{S}\Delta_p(\mu, \nu) = 0$ implies $\mu = \nu$. (The converse implication is immediate from the definition, since if $\mu = \nu$ then every slice coincides and the integral vanishes.)

(1) Assume $\mathbf{S}\Delta_p(\mu, \nu) = 0$. Since the integrand is nonnegative, this yields

$$\Delta\big((P_\theta)_{\#}\mu, (P_\theta)_{\#}\nu\big) = 0 \quad \text{for } \sigma\text{-almost every } \theta \in S^{d-1}.$$

By the base property of $\Delta$ in one dimension, we obtain

$$(P_\theta)_{\#}\mu = (P_\theta)_{\#}\nu \quad \text{for } \sigma\text{-almost every } \theta \in S^{d-1}.$$

*Notation.* For a probability measure $\xi$ on $\mathbb{R}^d$, we write $\widehat{\xi}$ for its characteristic function:

$$\widehat{\xi}(z) \;=\; \int_{\mathbb{R}^d} e^{i\langle z, x \rangle} \, d\xi(x), \qquad z \in \mathbb{R}^d.$$

(2) By Lemma 4, the one–dimensional pushforward $(P_\theta)_{\#}\xi$ satisfies

$$\widehat{(P_\theta)_{\#}\xi}(t) = \int_{\mathbb{R}} e^{itu} \, d\big((P_\theta)_{\#}\xi\big)(u) = \int_{\mathbb{R}^d} e^{it\langle \theta, x \rangle} \, d\xi(x) = \widehat{\xi}(t\theta), \quad t \in \mathbb{R}.$$

Hence $(P_\theta)_{\#}\mu = (P_\theta)_{\#}\nu$ implies

$$\widehat{\mu}(t\theta) = \widehat{\nu}(t\theta) \quad \text{for } \sigma\text{-almost every } \theta \in S^{d-1} \text{ and all } t \in \mathbb{R}.$$

*Interpretation.* Projecting onto $\theta$ in the original space corresponds to restricting $\widehat{\mu}$ to the line $\{t\theta : t \in \mathbb{R}\}$ in frequency space. Thus the two characteristic functions agree along almost all such lines.

(3) Therefore $\widehat{\mu} = \widehat{\nu}$ on $\mathbb{R}^d$, and by the injectivity of characteristic functions (distinct measures cannot share the same characteristic function; see e.g. (Billingsley, 1995, Thm. 26.2)) we conclude $\mu = \nu$.

**Triangle inequality.**
(iii) Assume that $\Delta$ is a metric on $\mathcal{P}(\mathbb{R})$. Let $\mu, \nu, \xi \in \mathcal{P}(\mathbb{R}^d)$. For every $\theta \in S^{d-1}$, the base triangle inequality gives

$$\Delta\big((P_\theta)_{\#}\mu, (P_\theta)_{\#}\nu\big) \;\leq\; \Delta\big((P_\theta)_{\#}\mu, (P_\theta)_{\#}\xi\big) + \Delta\big((P_\theta)_{\#}\xi, (P_\theta)_{\#}\nu\big).$$

Taking the $p$-th power and integrating over the sphere yields

$$\int_{S^{d-1}} \Delta^p\big((P_\theta)_{\#}\mu, (P_\theta)_{\#}\nu\big) \, d\sigma(\theta) \;\leq\; \int_{S^{d-1}} \big[\Delta\big((P_\theta)_{\#}\mu, (P_\theta)_{\#}\xi\big) + \Delta\big((P_\theta)_{\#}\xi, (P_\theta)_{\#}\nu\big)\big]^p \, d\sigma(\theta).$$

*Minkowski's inequality.* For $p \geq 1$ and measurable $f, g$ on a measure space $(X, \mu)$,

$$\left( \int_X |f(x) + g(x)|^p \, d\mu(x) \right)^{1/p} \leq \left( \int_X |f(x)|^p \, d\mu(x) \right)^{1/p} + \left( \int_X |g(x)|^p \, d\mu(x) \right)^{1/p}.$$

Using this inequality with $X = S^{d-1}$, $\mu = \sigma$, and

$$f(\theta) = \Delta\big((P_\theta)_{\#}\mu, (P_\theta)_{\#}\xi\big), \qquad g(\theta) = \Delta\big((P_\theta)_{\#}\xi, (P_\theta)_{\#}\nu\big),$$

we obtain

$$\mathbf{S}\Delta_p(\mu, \nu) \leq \mathbf{S}\Delta_p(\mu, \xi) + \mathbf{S}\Delta_p(\xi, \nu).$$

$\square$

**Lemma 2** (Max–sliced metric properties)**.** *Let* $\Delta : \mathcal{P}(\mathbb{R}) \times \mathcal{P}(\mathbb{R}) \to [0, \infty]$ *be a metric on* $\mathcal{P}(\mathbb{R})$. *For* $\mu, \nu \in \mathcal{P}(\mathbb{R}^d)$ *define*

$$\mathbf{MS}\Delta(\mu, \nu) := \sup_{\theta \in \mathbb{S}^{d-1}} \Delta\big((P_\theta)_{\#}\mu, \, (P_\theta)_{\#}\nu\big), \qquad P_\theta(x) = \langle \theta, x \rangle.$$

*Then* $\mathbf{MS}\Delta$ *is a metric on* $\mathcal{P}(\mathbb{R}^d)$*: it is nonnegative and symmetric, satisfies the identity of indiscernibles, and obeys the triangle inequality.*

*Proof.* We prove that $\mathbf{MS}\Delta$ satisfies the three defining properties required for a metric on $\mathcal{P}(\mathbb{R}^d)$.

**Nonnegativity and symmetry.** Each slice is nonnegative and symmetric because $\Delta$ is; taking a supremum preserves both properties.

**Identity of indiscernibles.** If $\mu = \nu$ then every slice is equal, so $\mathbf{MS}\Delta(\mu, \nu) = 0$. Conversely, if $\mathbf{MS}\Delta(\mu, \nu) = 0$, then

$$(P_\theta)_{\#}\mu = (P_\theta)_{\#}\nu \quad \text{for all } \theta \in \mathbb{S}^{d-1}.$$

The argument given in Proposition 1 for the sliced case then applies verbatim, showing that $\mu = \nu$.

**Triangle inequality.** For any $\theta \in \mathbb{S}^{d-1}$ and any $\mu, \nu, \xi \in \mathcal{P}(\mathbb{R}^d)$, the base metric property yields

$$\Delta\big((P_\theta)_{\#}\mu, (P_\theta)_{\#}\nu\big) \leq \Delta\big((P_\theta)_{\#}\mu, (P_\theta)_{\#}\xi\big) + \Delta\big((P_\theta)_{\#}\xi, (P_\theta)_{\#}\nu\big).$$

Taking the supremum over $\theta$ on both sides gives

$$\mathbf{MS}\Delta(\mu, \nu) \leq \mathbf{MS}\Delta(\mu, \xi) + \mathbf{MS}\Delta(\xi, \nu).$$

All three metric axioms hold; hence $\mathbf{MS}\Delta$ is a metric on $\mathcal{P}(\mathbb{R}^d)$. $\square$

**Lemma 3** (Supremum lift preserves metricity for SPDs and MaxSPDs—follows closely from Nguyen-Tang et al. (2021), Proposition 1 (Appendix A.1))**.** *Let* $\mathcal{D}$ *be a metric on* $\mathcal{P}(\mathbb{R}^d)$. *(In our use,* $\mathcal{D}$ *will be either the SPD* $\mathbf{S}\Delta^{\rho,p}$ *or the MaxSPD* $\mathbf{MS}\Delta$.) *Define, for* $\mu, \nu : \mathcal{S} \times \mathcal{A} \to \mathcal{P}(\mathbb{R}^d)$,

$$\overline{\mathcal{D}}(\mu, \nu) := \sup_{(s,a) \in \mathcal{S} \times \mathcal{A}} \mathcal{D}\big(\mu(s,a), \nu(s,a)\big).$$

*Then* $\overline{\mathcal{D}}$ *is a metric on* $\mathcal{P}(\mathbb{R}^d)^{\mathcal{S} \times \mathcal{A}}$.

*Proof.* **Nonnegativity and symmetry.** Since $\mathcal{D}$ is nonnegative and symmetric pointwise, the supremum of such quantities preserves these properties. Hence $\overline{\mathcal{D}}(\mu, \nu) \geq 0$ and $\overline{\mathcal{D}}(\mu, \nu) = \overline{\mathcal{D}}(\nu, \mu)$.

**Identity of indiscernibles.** If $\mu = \nu$, then every term vanishes and $\overline{\mathcal{D}}(\mu, \nu) = 0$. Conversely, if $\overline{\mathcal{D}}(\mu, \nu) = 0$, then $\mathcal{D}(\mu(s,a), \nu(s,a)) = 0$ for each $(s,a)$, which by metricity of $\mathcal{D}$ implies $\mu(s,a) = \nu(s,a)$ everywhere, hence $\mu = \nu$.

**Triangle inequality.** Let $\mu, \nu, \eta : \mathcal{S} \times \mathcal{A} \to \mathcal{P}(\mathbb{R}^d)$. Then

$$\overline{\mathcal{D}}(\mu, \nu) = \sup_{(s,a)} \mathcal{D}\big(\mu(s,a),\, \nu(s,a)\big) \tag{23}$$

$$\overset{(a)}{\leq} \sup_{(s,a)} \Big\{ \mathcal{D}\big(\mu(s,a),\, \eta(s,a)\big) + \mathcal{D}\big(\eta(s,a),\, \nu(s,a)\big) \Big\} \tag{24}$$

$$\overset{(b)}{\leq} \sup_{(s,a)} \mathcal{D}\big(\mu(s,a),\, \eta(s,a)\big) \; + \; \sup_{(s,a)} \mathcal{D}\big(\eta(s,a),\, \nu(s,a)\big) \tag{25}$$

$$= \overline{\mathcal{D}}(\mu, \eta) + \overline{\mathcal{D}}(\eta, \nu). \tag{26}$$

Here (a) is the pointwise triangle inequality for $\mathcal{D}$, and (b) uses $\sup(A+B) \leq \sup A + \sup B$.

Thus $\overline{\mathcal{D}}$ satisfies all four metric axioms. Specializing $\mathcal{D}$ to $\mathbf{S}\Delta^{\rho,p}$ or $\mathbf{MS}\Delta$ yields that $\overline{\mathbf{S}\Delta}^{\rho,p}$ and $\overline{\mathbf{MS}\Delta}$ are metrics on $\mathcal{P}(\mathbb{R}^d)^{\mathcal{S} \times \mathcal{A}}$. $\qquad\square$

**Theorem 1** (Global metricity of (max-)sliced lifts)**.** *Let $\Delta$ be a metric on $\mathcal{P}(\mathbb{R})$ and let $p \in [1, \infty)$. Let $\sigma$ denote the uniform probability measure on the unit sphere $S^{d-1} \subset \mathbb{R}^d$. Define the uniform sliced divergence*

$$\mathbf{S}\Delta_p(\mu, \nu) := \left( \int_{S^{d-1}} \Delta^p\big((P_\theta)_\#\mu, (P_\theta)_\#\nu\big) \, d\sigma(\theta) \right)^{1/p},$$

*and the max–sliced divergence*

$$\mathbf{MS}\Delta(\mu, \nu) := \sup_{\theta \in S^{d-1}} \Delta\big((P_\theta)_\#\mu, (P_\theta)_\#\nu\big), \qquad P_\theta(x) = \langle \theta, x \rangle.$$

*Then:*

1. *$\mathbf{S}\Delta_p$ is a metric on $\mathcal{P}(\mathbb{R}^d)$.*

2. *$\mathbf{MS}\Delta$ is a metric on $\mathcal{P}(\mathbb{R}^d)$.*

3. *For return–distribution functions $\eta_i : \mathcal{S} \times \mathcal{A} \to \mathcal{P}(\mathbb{R}^d)$, the supremum lifts*

$$\overline{\mathbf{S}\Delta}_p(\eta_1, \eta_2) := \sup_{(s,a)} \mathbf{S}\Delta_p\big(\eta_1(s,a), \eta_2(s,a)\big),$$

   *and*

$$\overline{\mathbf{MS}\Delta}(\eta_1, \eta_2) := \sup_{(s,a)} \mathbf{MS}\Delta\big(\eta_1(s,a), \eta_2(s,a)\big),$$

   *are metrics on $\mathcal{P}(\mathbb{R}^d)^{\mathcal{S} \times \mathcal{A}}$.*

Proof. *(i) is Lemma 1; (ii) is Lemma 2; (iii) follows from Lemma 3 by taking $\mathcal{D} = \mathbf{S}\Delta_p$ or $\mathcal{D} = \mathbf{MS}\Delta$.*

## C.2 CONTRACTION PROPERTY

**Lemma 4** (Push-forward law identity). *Let $Z$ be a random variable with distribution $\mu$, and let $f$ be any measurable function. Then*

$$f_{\#}\mu \;=\; \mathrm{Law}\big(f(Z)\big).$$

*Proof.* For any Borel set $A$,

$$\Pr\big(f(Z) \in A\big) = \Pr\big(Z \in f^{-1}(A)\big) = \mu\big(f^{-1}(A)\big) = f_{\#}\mu(A).$$

Since this holds for all $A$, we conclude $f_{\#}\mu = \mathrm{Law}(f(Z))$. $\qquad\square$

**Lemma 5** (Affine Bellman update = affine pushforward). *Fix $(s,a)$. Let $C$ collect all environment/policy randomness, and let $(S', A') = g(s, a; C)$. Let $\eta$ map each $(x, u)$ to a law $\eta(x, u)$ on $\mathbb{R}^d$, and let $X' \sim \eta(S', A')$ (conditionally on $C$). Given an offset $b_{s,a} : \mathrm{supp}(C) \to \mathbb{R}^d$ and a measurable matrix map $L_{s,a} : \mathrm{supp}(C) \to \mathbb{R}^{d \times d}$, define*

$$\Phi_{s,a}(x; C) \;=\; b_{s,a}(C) + L_{s,a}(C)\,x.$$

*Then*

$$(T^{\pi}\eta)(s, a) \;=\; \mathrm{Law}\big(\Phi_{s,a}(X'; C)\big).$$

*Proof.* Fix a Borel set $A \subset \mathbb{R}^d$. Using the definition of pushforward laws,

$$\Pr\big(\Phi_{s,a}(X'; C) \in A\big) = \mathbb{E}_C\Big[\, \Pr\big(b_{s,a}(C) + L_{s,a}(C)X' \in A \,\big|\, C\big)\Big]$$

$$= \mathbb{E}_C\Big[\, \big(x \mapsto b_{s,a}(C) + L_{s,a}(C)x\big)_{\#}\,\eta(S', A')(A)\,\Big].$$

By definition of the distributional Bellman operator with affine update $z \mapsto b_{s,a}(C) + L_{s,a}(C)z$ and next index $(S', A')$, the right-hand side equals $(T^{\pi}\eta)(s, a)(A)$. Since this holds for all Borel $A$, the laws coincide. $\qquad\square$

### C.2.1 UNIVARIATE CASE

**Lemma 6** (Univariate affine push-forward contraction). *Let $\Delta$ be a metric on $\mathcal{P}(\mathbb{R})$. Assume for all $\mu, \nu \in \mathcal{P}(\mathbb{R})$:*

(T) *Translation non-expansion: for every $t \in \mathbb{R}$,*

$$\Delta\big((T_t)_{\#}\mu, (T_t)_{\#}\nu\big) \leq \Delta(\mu, \nu), \quad T_t(x) = x + t.$$

(S) *Scale–Lipschitz: there exists a nondecreasing $c : [0, \infty) \to [0, \infty)$ such that for every $s \geq 0$,*

$$\Delta\big((x \mapsto sx)_{\#}\mu, (x \mapsto sx)_{\#}\nu\big) \leq c(s)\,\Delta(\mu, \nu).$$

*Let $F(x) = t + \gamma x$ with arbitrary $t \in \mathbb{R}$ and the same $\gamma \in [0, 1)$. Then, for all $\mu, \nu \in \mathcal{P}(\mathbb{R})$,*

$$\Delta\big(F_{\#}\mu, F_{\#}\nu\big) \leq c(\gamma)\,\Delta(\mu, \nu)$$

*In particular, if $c(\gamma) < 1$, the push-forward $F_{\#}$ is a contraction on $(\mathcal{P}(\mathbb{R}), \Delta)$.*

*Proof.* Let $U \sim \mu$ and $V \sim \nu$. By Lemma 4,

$$\Delta\big(F_{\#}\mu, F_{\#}\nu\big) = \Delta\big(\mathrm{Law}(t + \gamma U), \mathrm{Law}(t + \gamma V)\big).$$

By (T),

$$\Delta\big(\mathrm{Law}(t + \gamma U), \mathrm{Law}(t + \gamma V)\big) \leq \Delta\big(\mathrm{Law}(\gamma U), \mathrm{Law}(\gamma V)\big).$$

By (S) with $s = \gamma$,

$$\Delta\big(\mathrm{Law}(\gamma U), \mathrm{Law}(\gamma V)\big) \leq c(\gamma)\,\Delta\big(\mathrm{Law}(U), \mathrm{Law}(V)\big) = c(\gamma)\,\Delta(\mu, \nu).$$

$\qquad\square$

**Lemma 7** (Mixture $p$-convexity $\Rightarrow$ marginal bound). *Let $\Delta$ be a metric on $\mathcal{P}(\mathbb{R}^d)$ and fix $p \in [1, \infty)$. Assume $\Delta$ satisfies the mixture $p$-convexity property:*

$$\Delta\left(\int_\Omega \mu_c \, \rho(dc), \int_\Omega \nu_c \, \rho(dc)\right) \leq \left(\int_\Omega \Delta(\mu_c, \nu_c)^p \, \rho(dc)\right)^{1/p}, \qquad (27)$$

*for all probability spaces $(\Omega, \mathcal{F}, \rho)$ and measurable families $(\mu_c)_{c\in\Omega}, (\nu_c)_{c\in\Omega}$.*

*Let $C$ be a random variable with law $\rho$ and let $Z_1, Z_2$ be $\mathbb{R}^d$-valued random variables. If*

$$\sup_{c\in\Omega} \Delta\big(\mathrm{Law}(Z_1 \mid C = c), \, \mathrm{Law}(Z_2 \mid C = c)\big) \leq \delta,$$

*then*

$$\Delta\big(\mathrm{Law}(Z_1), \, \mathrm{Law}(Z_2)\big) \leq \delta.$$

*Proof.* Set $\mu_c := \mathrm{Law}(Z_1 \mid C = c)$ and $\nu_c := \mathrm{Law}(Z_2 \mid C = c)$. By the law of total probability,

$$\mathrm{Law}(Z_1) = \int_\Omega \mu_c \, \rho(dc), \qquad \mathrm{Law}(Z_2) = \int_\Omega \nu_c \, \rho(dc).$$

Define $f(c) := \Delta(\mu_c, \nu_c) \geq 0$. The hypothesis gives the pointwise bound $f(c) \leq \delta$ for all $c \in \Omega$. Applying equation 27 and then monotonicity of the integral,

$$\Delta\big(\mathrm{Law}(Z_1), \mathrm{Law}(Z_2)\big) \leq \left(\int_\Omega f(c)^p \, \rho(dc)\right)^{1/p} \leq \left(\int_\Omega \delta^p \, \rho(dc)\right)^{1/p} = \delta.$$

$\square$

**Theorem 2** (Supremum-$\Delta$ contraction of the univariate distributional Bellman operator). *This proposition slightly generalizes Theorem 4.25 of Bellemare et al. (2023a).*

*Let $\Delta$ be a metric on $\mathcal{P}(\mathbb{R})$ and define*

$$\bar{\Delta}(\eta_1, \eta_2) := \sup_{(s,a)} \Delta\big(\eta_1(s,a), \eta_2(s,a)\big), \qquad \eta_i : \mathcal{S} \times \mathcal{A} \to \mathcal{P}(\mathbb{R}).$$

*Assume $\Delta$ satisfies:*

**(T)** *Translation nonexpansion: $\Delta\big((T_t)_\# \mu, (T_t)_\# \nu\big) \leq \Delta(\mu, \nu)$ for all $t \in \mathbb{R}$.*

**(S)** *Scale–Lipschitz: there exists a nondecreasing $c : [0, \infty) \to [0, \infty)$ such that for every $s \geq 0$,*

$$\Delta\big((x \mapsto sx)_\# \mu, (x \mapsto sx)_\# \nu\big) \leq c(s)\, \Delta(\mu, \nu).$$

**($M_p$)** *Mixture $p$-convexity: for some $p \in [1, \infty)$ and all probability spaces $(\Omega, \mathcal{F}, \rho)$ and measurable families $(\mu_c), (\nu_c) \subset \mathcal{P}(\mathbb{R})$,*

$$\Delta\Big(\int_\Omega \mu_c \, \rho(dc), \int_\Omega \nu_c \, \rho(dc)\Big) \leq \Big(\int_\Omega \Delta(\mu_c, \nu_c)^p \, \rho(dc)\Big)^{1/p}.$$

*For each $(s,a)$, let $C$ be a random element, set $(S', A') = g(s, a; C)$, and let $b_{s,a} : \mathrm{supp}(C) \to \mathbb{R}$ be measurable. Define*

$$(T^\pi \eta)(s,a) := \mathrm{Law}\big(b_{s,a}(C) + \gamma X'\big), \qquad X' \sim \eta(S', A') \text{ conditionally on } C.$$

*Then, for all $\eta_1, \eta_2$,*

$$\boxed{\bar{\Delta}\big(T^\pi \eta_1, \, T^\pi \eta_2\big) \ \leq \ c(\gamma)\, \bar{\Delta}\big(\eta_1, \eta_2\big).}$$

*In particular, if $c(\gamma) < 1$, the operator $T^\pi$ is a contraction on $(\mathcal{S} \times \mathcal{A} \to \mathcal{P}(\mathbb{R}), \bar{\Delta})$.*

*Proof.* By definition,
$$\bar{\Delta}\big(T^\pi \eta_1, T^\pi \eta_2\big) = \sup_{(s,a)} \Delta\big((T^\pi \eta_1)(s,a),\, (T^\pi \eta_2)(s,a)\big).$$

Fix $(s,a)$. Let $Z_i := b_{s,a}(C) + \gamma X_i'$ where, conditionally on $C$, $X_i' \sim \eta_i(S', A')$ and $(S', A') = g(s,a;C)$. By the push-forward law identity (Lemma 4),
$$(T^\pi \eta_i)(s,a) = \mathrm{Law}(Z_i), \qquad \mathrm{Law}(X_i' \mid C) = \eta_i(S', A').$$

Condition on $C$ and define $\Phi_{s,a}(\cdot; C) : x \mapsto b_{s,a}(C) + \gamma x$. By the univariate affine push-forward contraction (Lemma 6, using (T) and (S)),

$$\Delta\big(\mathrm{Law}(Z_1 \mid C),\, \mathrm{Law}(Z_2 \mid C)\big) \leq c(\gamma)\, \Delta\big(\mathrm{Law}(X_1' \mid C),\, \mathrm{Law}(X_2' \mid C)\big)$$
$$= c(\gamma)\, \Delta\big(\eta_1(S', A'),\, \eta_2(S', A')\big).$$

Apply mixture $p$-convexity (assumption ($M_p$)) to the conditional laws and then Lemma 7 (with the pointwise bound $\Delta(\mathrm{Law}(Z_1 \mid C), \mathrm{Law}(Z_2 \mid C)) \leq c(\gamma)\, \Delta(\eta_1(S', A'), \eta_2(S', A'))$):

$$\Delta\big(\mathrm{Law}(Z_1),\, \mathrm{Law}(Z_2)\big) \leq \Big(\mathbb{E}\big[\,\Delta\big(\mathrm{Law}(Z_1 \mid C), \mathrm{Law}(Z_2 \mid C)\big)^p\big]\Big)^{1/p}$$
$$\leq c(\gamma)\, \Big(\mathbb{E}\big[\,\Delta\big(\eta_1(S', A'), \eta_2(S', A')\big)^p\big]\Big)^{1/p}$$
$$\leq c(\gamma)\, \bar{\Delta}(\eta_1, \eta_2).$$

Therefore,
$$\Delta\big((T^\pi \eta_1)(s,a),\, (T^\pi \eta_2)(s,a)\big) \leq c(\gamma)\, \bar{\Delta}(\eta_1, \eta_2).$$

Taking the supremum over $(s,a)$ yields the stated bound. $\qquad\square$

### C.2.2  UNIFORM SLICING

**Lemma 8** (Sliced affine push-forward contraction — scaled orthogonal case). *Let $\Delta$ be a divergence on $\mathcal{P}(\mathbb{R})$. Assume that for all $\alpha, \beta \in \mathcal{P}(\mathbb{R})$ the following hold:*

**(T)** *Translation nonexpansion: for every $t \in \mathbb{R}$,*
$$\Delta\big((x \mapsto x + t)_{\#}\alpha, \, (x \mapsto x + t)_{\#}\beta\big) = \Delta(\alpha, \beta).$$

**(S)** *Scale–Lipschitz: there exists a nondecreasing $c : [0, \infty) \to [0, \infty)$ such that for every $s \geq 0$,*
$$\Delta\big((x \mapsto sx)_{\#}\mu, \, (x \mapsto sx)_{\#}\nu\big) \leq c(s) \, \Delta(\mu, \nu).$$

*For $\sigma$ a rotation-invariant probability measure on $\mathbb{S}^{d-1}$ and $q \in [1, \infty)$, define the sliced lift*
$$\mathbf{S}\Delta_q(\mu, \nu) := \Big( \int_{\mathbb{S}^{d-1}} \Delta\big((P_\theta)_{\#}\mu, (P_\theta)_{\#}\nu\big)^q \, d\sigma(\theta) \Big)^{1/q}, \quad P_\theta(x) = \langle \theta, x \rangle.$$

*Let $F(x) = Ax + b$ with $A = \gamma O$ where $O$ is orthogonal and $\gamma \in [0, 1)$. Then, for all $\mu, \nu \in \mathcal{P}(\mathbb{R}^d)$,*
$$\boxed{\mathbf{S}\Delta_q\big(F_{\#}\mu, F_{\#}\nu\big) \, \leq \, c(\gamma) \, \mathbf{S}\Delta_q(\mu, \nu).}$$

*In particular, if $c(\gamma) < 1$, the push-forward $F_{\#}$ is a contraction on $(\mathcal{P}(\mathbb{R}^d), \mathbf{S}\Delta^q)$.*

*Proof.* Fix $\theta \in \mathbb{S}^{d-1}$ and let $\phi_\theta := O^\top \theta$ (note $\|\phi_\theta\| = 1$). For any $X \sim \mu$ and $Y \sim \nu$,
$$\langle \theta, AX + b \rangle = \langle \theta, b \rangle + \gamma \langle O^\top \theta, X \rangle = \langle \theta, b \rangle + \gamma \langle \phi_\theta, X \rangle,$$
and similarly for $Y$. By **(T)**,
$$\Delta(\mathrm{Law}(\langle \theta, AX + b \rangle), \, \mathrm{Law}(\langle \theta, AY + b \rangle)) \, \leq \, \Delta(\mathrm{Law}(\gamma \langle \phi_\theta, X \rangle), \, \mathrm{Law}(\gamma \langle \phi_\theta, Y \rangle)).$$
By **(S)** with $s = \gamma$,
$$\Delta(\mathrm{Law}(\gamma \langle \phi_\theta, X \rangle), \, \mathrm{Law}(\gamma \langle \phi_\theta, Y \rangle)) \leq c(\gamma) \, \Delta(\mathrm{Law}(\langle \phi_\theta, X \rangle), \, \mathrm{Law}(\langle \phi_\theta, Y \rangle)). \qquad (\star)$$
Raise $(\star)$ to the $q$-th power and integrate over $\theta \sim \sigma$:
$$\int \Delta\big((P_\theta)_{\#}F_{\#}\mu, (P_\theta)_{\#}F_{\#}\nu\big)^q \, d\sigma(\theta) \, \leq \, c(\gamma)^q \int \Delta\big((P_{\phi_\theta})_{\#}\mu, (P_{\phi_\theta})_{\#}\nu\big)^q \, d\sigma(\theta).$$
Since $\sigma$ is rotation-invariant and $\phi_\theta = O^\top \theta$, the change of variables $\phi = O^\top \theta$ preserves $\sigma$:
$$\int \Delta\big((P_{\phi_\theta})_{\#}\mu, (P_{\phi_\theta})_{\#}\nu\big)^q \, d\sigma(\theta) = \int \Delta\big((P_\phi)_{\#}\mu, (P_\phi)_{\#}\nu\big)^q \, d\sigma(\phi).$$
Taking the $q$-th root yields $\mathbf{S}\Delta_q(F_{\#}\mu, F_{\#}\nu) \leq c(\gamma) \, \mathbf{S}\Delta_q(\mu, \nu)$. $\qquad\square$

**Lemma 9** (Mixture $p$-convexity lifts to the sliced divergence). *Let $\Delta$ be a divergence on $\mathcal{P}(\mathbb{R})$ satisfying mixture $p$-convexity: for every probability space $(\Omega, \mathcal{F}, \rho)$ and measurable families $(\mu_c)_{c \in \Omega}, (\nu_c)_{c \in \Omega} \subset \mathcal{P}(\mathbb{R})$,*
$$\Delta\bigg( \int_\Omega \mu_c \, \rho(dc), \int_\Omega \nu_c \, \rho(dc) \bigg) \leq \bigg( \int_\Omega \Delta(\mu_c, \nu_c)^p \, \rho(dc) \bigg)^{1/p}, \quad p \in [1, \infty).$$

*Fix any probability measure $\sigma$ on $\mathbb{S}^{d-1}$. Define the sliced lift for $\mu, \nu \in \mathcal{P}(\mathbb{R}^d)$ by*
$$\mathbf{S}\Delta_p(\mu, \nu) := \bigg( \int_{\mathbb{S}^{d-1}} \Delta\big((P_\theta)_{\#}\mu, (P_\theta)_{\#}\nu\big)^p \, \sigma(d\theta) \bigg)^{1/p}, \qquad P_\theta(x) = \langle \theta, x \rangle.$$

*Then $\mathbf{S}\Delta_p$ is mixture $p$-convex on $\mathcal{P}(\mathbb{R}^d)$, i.e., for any measurable families $(\mu_c)_{c \in \Omega}, (\nu_c)_{c \in \Omega} \subset \mathcal{P}(\mathbb{R}^d)$,*
$$\boxed{\mathbf{S}\Delta_p\bigg( \int_\Omega \mu_c \, \rho(dc), \int_\Omega \nu_c \, \rho(dc) \bigg) \, \leq \, \bigg( \int_\Omega \mathbf{S}\Delta_p(\mu_c, \nu_c)^p \, \rho(dc) \bigg)^{1/p}.}$$

*Proof.* Fix $\theta \in \mathbb{S}^{d-1}$ and set $\mu_c^\theta := (P_\theta)_\# \mu_c$, $\nu_c^\theta := (P_\theta)_\# \nu_c \in \mathcal{P}(\mathbb{R})$. By linearity of pushforward w.r.t. mixtures,

$$(P_\theta)_\# \left( \int_\Omega \mu_c \, \rho(dc) \right) = \int_\Omega \mu_c^\theta \, \rho(dc), \qquad (P_\theta)_\# \left( \int_\Omega \nu_c \, \rho(dc) \right) = \int_\Omega \nu_c^\theta \, \rho(dc).$$

Applying mixture $p$-convexity of $\Delta$ in 1-D at this fixed $\theta$,

$$\Delta\left( \int_\Omega \mu_c^\theta \, \rho(dc), \int_\Omega \nu_c^\theta \, \rho(dc) \right) \leq \left( \int_\Omega \Delta(\mu_c^\theta, \nu_c^\theta)^p \, \rho(dc) \right)^{1/p}.$$

Raise to the $p$th power and integrate over $\theta \sim \sigma$; Tonelli/Fubini yields

$$\int_{\mathbb{S}^{d-1}} \Delta\big( (P_\theta)_\# \textstyle\int \mu_c \, d\rho, \ (P_\theta)_\# \int \nu_c \, d\rho \big)^p \ \sigma(d\theta)$$

$$\leq \int_\Omega \left( \int_{\mathbb{S}^{d-1}} \Delta\big( (P_\theta)_\# \mu_c, \ (P_\theta)_\# \nu_c \big)^p \, \sigma(d\theta) \right) \rho(dc).$$

By the very definition of the sliced divergence,

$$\int_{\mathbb{S}^{d-1}} \Delta\big( (P_\theta)_\# \textstyle\int \mu_c \, d\rho, \ (P_\theta)_\# \int \nu_c \, d\rho \big)^p \ \sigma(d\theta) = \mathbf{S}\Delta_p \big( \textstyle\int \mu_c \, d\rho, \ \int \nu_c \, d\rho \big)^p$$

$$\leq \int_\Omega \left( \int_{\mathbb{S}^{d-1}} \Delta\big( (P_\theta)_\# \mu_c, \ (P_\theta)_\# \nu_c \big)^p \, \sigma(d\theta) \right) \rho(dc)$$

$$= \int_\Omega \mathbf{S}\Delta_p(\mu_c, \nu_c)^p \, \rho(dc).$$

Taking the $p$th root gives

$$\mathbf{S}\Delta_p \left( \int_\Omega \mu_c \, \rho(dc), \int_\Omega \nu_c \, \rho(dc) \right) \leq \left( \int_\Omega \mathbf{S}\Delta_p(\mu_c, \nu_c)^p \, \rho(dc) \right)^{1/p}.$$

$\square$

**Theorem 3** (Supremum–sliced contraction of the multivariate distributional Bellman operator (scaled isometry))**.** *Let $\Delta$ be a divergence on $\mathcal{P}(\mathbb{R})$ and let $\sigma$ be a rotation–invariant probability measure on $\mathbb{S}^{d-1}$. The sliced probability divergence $\mathbf{S}\Delta_p$ is defined using this fixed slicing measure $\sigma$ (cf. Lemma 8). Define*

$$\overline{\mathbf{S}\Delta}_p(\eta_1, \eta_2) := \sup_{(s,a)} \mathbf{S}\Delta_p \big( \eta_1(s,a), \eta_2(s,a) \big), \qquad \eta_i : \mathcal{S} \times \mathcal{A} \to \mathcal{P}(\mathbb{R}^d).$$

*Assume $\Delta$ satisfies:*

    **(T)** *Translation nonexpansion:* $\Delta\big( (x \mapsto x+t)_\# \alpha, \ (x \mapsto x+t)_\# \beta \big) \leq \Delta(\alpha, \beta)$ *for all $t \in \mathbb{R}$.*

    **(S)** *Scale–Lipschitz at $\gamma$: there exists $c : [0, \infty) \to [0, \infty)$ such that*

$$\Delta\big( (x \mapsto sx)_\# \alpha, \ (x \mapsto sx)_\# \beta \big) \leq c(s) \, \Delta(\alpha, \beta) \quad \text{for all } s \geq 0,$$

    *with some $\gamma \in (0, 1)$ for which $c(\gamma) < 1$.*

  **($M_p$)** *Mixture $p$-convexity: for every probability space $(\Omega, \mathcal{F}, \rho)$ and measurable families $(\alpha_c), (\beta_c) \subset \mathcal{P}(\mathbb{R})$,*

$$\Delta\left( \int \mu_c \, \rho(dc), \int \nu_c \, \rho(dc) \right) \leq \left( \int \Delta(\mu_c, \nu_c)^p \, \rho(dc) \right)^{1/p}.$$

***Bellman update (scaled isometry).*** *Fix a state–action pair $(s, a)$. All randomness induced by the dynamics and the policy is gathered in a single random element $C$. Once $C$ is realized, it determines the successor index through a measurable mapping $g$:*

$$(S', A') := g(s, a; C).$$

*At $(s, a)$ we allow an affine transformation composed of a translation and a rotation scaled by the discount. The translation is simply a vector that may depend on $C$; we write*

$$b_{s,a}(C) \in \mathbb{R}^d.$$

*The rotation may also depend on $C$: take any $O_{s,a}(C) \in \mathsf{O}(d)$. The linear part of the update is the scaled isometry*

$$A_{s,a}(C) := \gamma \, O_{s,a}(C) \qquad (\gamma \in (0,1)).$$

*Conditioned on $C$, the "next" sample is drawn from the law at the successor index:*

$$X' \,|\, C \sim \eta(S', A').$$

*The Bellman update at $(s, a)$ is then defined as the push-forward of $X'$ by this affine map; equivalently, it is the law of the random vector obtained by translating and rotating–scaling $X'$:*

$$(T^\pi \eta)(s,a) := \mathrm{Law}\big(b_{s,a}(C) + A_{s,a}(C) \, X'\big).$$

*Then for all $\eta_1, \eta_2$,*

$$\boxed{\overline{\mathbf{S}\Delta}_p\big(T^\pi \eta_1, \, T^\pi \eta_2\big) \;\leq\; c(\gamma) \, \overline{\mathbf{S}\Delta}_p\big(\eta_1, \, \eta_2\big).}$$

*In particular, if $c(\gamma) < 1$, the operator $T^\pi$ is a contraction on $\big(\mathcal{S} \times \mathcal{A} \to \mathcal{P}(\mathbb{R}^d), \, \overline{\mathbf{S}\Delta}_p\big)$.*

*Proof.* Fix $(s, a)$ and condition on $C$. Define

$$\Phi_{s,a}(x; C) := b_{s,a}(C) + A_{s,a}(C)x \quad \text{with} \quad A_{s,a}(C) = \gamma O_{s,a}(C).$$

By Lemma 5, with $L_{s,a} = A_{s,a}(C)$, the update satisfies

$$(T^\pi \eta)(s,a) = \mathrm{Law}\big(\Phi_{s,a}(X'; C)\big).$$

For $X_i' \sim \eta_i(S', A')$ (conditionally on $C$), set

$$Z_i := \Phi_{s,a}(X_i'; C).$$

**Affine push-forward at fixed $C$.** By Lemma 8, which itself relies on **(T)** and **(S)**, pushing forward *any* pair of multivariate laws by a map $x \mapsto b + \gamma O x$ (translation plus scaled isometry) contracts the sliced divergence by at most the factor $c(\gamma)$. Applying this to $\mathrm{Law}(X_i' \mid C)$ yields. Since $(S', A') = g(s, a; C)$ is fixed once $C$ is given, we have $\mathrm{Law}(X_i' \mid C) = \eta_i\big(g(s, a; C)\big) = \eta_i(S', A')$. Thus

$$\mathbf{S}\Delta_p\big(\mathrm{Law}(Z_1 \mid C), \, \mathrm{Law}(Z_2 \mid C)\big) \leq c(\gamma) \, \mathbf{S}\Delta_p\big(\mathrm{Law}(X_1' \mid C), \, \mathrm{Law}(X_2' \mid C)\big) \qquad (28)$$

$$= c(\gamma) \, \mathbf{S}\Delta_p\big(\eta_1\big(S', A'\big), \, \eta_2\big(S', A'\big)\big). \qquad (29)$$

**Averaging over $C$.** Lemma 9 asserts that **($\mathbf{M}_p$)** lifts from $\Delta$ to its sliced version. Combining the mixture $p$-convexity inequality with the bound valid for each fixed $C$ in equation 28 gives

$$\mathbf{S}\Delta_p\big(\mathrm{Law}(Z_1), \, \mathrm{Law}(Z_2)\big) \leq \left( \int \mathbf{S}\Delta_p\big(\mathrm{Law}(Z_1 \mid C), \mathrm{Law}(Z_2 \mid C)\big)^p \, \rho(dC) \right)^{1/p} \qquad (30)$$

$$\leq \left( \int \big(c(\gamma) \, \mathbf{S}\Delta_p\big(\eta_1(S', A'), \eta_2(S', A')\big)\big)^p \, \rho(dC) \right)^{1/p} \qquad (31)$$

$$= c(\gamma) \left( \int \mathbf{S}\Delta_p\big(\eta_1(S', A'), \eta_2(S', A')\big)^p \, \rho(dC) \right)^{1/p}. \qquad (32)$$

**Supremum bound.** For any given realization of $C$ and by definition of the supremum metric,

$$\mathbf{S}\Delta_p\big(\eta_1(S', A'), \eta_2(S', A')\big) \;\leq\; \sup_{(s,a)} \mathbf{S}\Delta_p\big(\eta_1(s,a), \eta_2(s,a)\big) = \overline{\mathbf{S}\Delta}_p(\eta_1, \eta_2).$$

Combining this pointwise bound with the integral inequality obtained above,

$$\mathbf{S}\Delta_p\big((T^\pi\eta_1)(s,a),\,(T^\pi\eta_2)(s,a)\big) = \mathbf{S}\Delta_p\big(\mathrm{Law}(Z_1),\,\mathrm{Law}(Z_2)\big) \tag{33}$$

$$\leq c(\gamma)\left(\int \mathbf{S}\Delta_p\big(\eta_1(S',A'),\eta_2(S',A')\big)^p\,\rho(dC)\right)^{1/p} \tag{34}$$

$$\leq c(\gamma)\left(\int \overline{\mathbf{S}\Delta}_p(\eta_1,\eta_2)^p\,\rho(dC)\right)^{1/p} \tag{35}$$

$$= c(\gamma)\,\overline{\mathbf{S}\Delta}_p(\eta_1,\eta_2). \tag{36}$$

Taking the supremum over $(s,a)$ yields

$$\overline{\mathbf{S}\Delta}_p(T^\pi\eta_1, T^\pi\eta_2) \;\leq\; c(\gamma)\,\overline{\mathbf{S}\Delta}_p(\eta_1,\eta_2).$$

$\square$

### C.2.3 MAX SLICING

**Lemma 10** (Max–sliced affine push-forward contraction — anisotropic linear case). *Let $\Delta$ be a divergence on $\mathcal{P}(\mathbb{R})$. Assume that for all $\mu, \nu \in \mathcal{P}(\mathbb{R})$:*

> **(T)** *Translation non-expansion: for every $t \in \mathbb{R}$,*
> $$\Delta\big((x \mapsto x + t)_{\#}\mu, \, (x \mapsto x + t)_{\#}\nu\big) \ \leq \ \Delta(\mu, \nu).$$

> **(S)** *Scale–Lipschitz: there exists a nondecreasing $c : [0, \infty) \to [0, \infty)$ such that for every $s \geq 0$,*
> $$\Delta\big((x \mapsto sx)_{\#}\mu, \, (x \mapsto sx)_{\#}\nu\big) \leq c(s)\, \Delta(\mu, \nu).$$

*Define the* max–sliced *lift of $\Delta$ by*
$$\mathbf{MS}\Delta(\mu, \nu) := \sup_{\theta \in \mathbb{S}^{d-1}} \Delta\big((P_\theta)_{\#}\mu, \, (P_\theta)_{\#}\nu\big), \qquad P_\theta(x) = \langle \theta, x \rangle.$$

*Let $F(x) = Ax + b$ with an arbitrary matrix $A \in \mathbb{R}^{d \times d}$ and $b \in \mathbb{R}^d$, and denote $L := \|A\|_{\mathrm{op}} = \sup_{\|v\|=1} \|Av\|$. Then, for all $\mu, \nu \in \mathcal{P}(\mathbb{R}^d)$,*

$$\boxed{\mathbf{MS}\Delta\big(F_{\#}\mu, F_{\#}\nu\big) \ \leq \ c(L)\, \mathbf{MS}\Delta(\mu, \nu).}$$

*Proof.* Fix $\theta \in \mathbb{S}^{d-1}$ and set $w_\theta := A^\top \theta$.

**Case 1:** $w_\theta = 0$. Then $(P_\theta \circ F)(x) = \langle \theta, b \rangle$ is constant, hence
$$\Delta\big((P_\theta)_{\#}F_{\#}\mu, \, (P_\theta)_{\#}F_{\#}\nu\big) = 0 \tag{37}$$
$$\leq c(0)\, \Delta\big((P_\phi)_{\#}\mu, \, (P_\phi)_{\#}\nu\big) \qquad \text{for any unit } \phi, \tag{38}$$

so the desired bound holds trivially.

**Case 2:** $\|w_\theta\| > 0$. Write $r_\theta := \|w_\theta\|$ and $\phi_\theta := w_\theta/r_\theta \in \mathbb{S}^{d-1}$. For any $X \sim \mu$ and $Y \sim \nu$,
$$(P_\theta \circ F)(X) = \langle \theta, AX + b \rangle = \langle \theta, b \rangle + r_\theta \langle \phi_\theta, X \rangle,$$

and similarly for $Y$. By **(T)** and **(S)** we obtain
$$\Delta\big((P_\theta)_{\#}F_{\#}\mu, \, (P_\theta)_{\#}F_{\#}\nu\big) = \Delta\big(\mathrm{Law}(r_\theta \langle \phi_\theta, X \rangle), \, \mathrm{Law}(r_\theta \langle \phi_\theta, Y \rangle)\big) \tag{39}$$
$$\leq c(r_\theta)\, \Delta\big((P_{\phi_\theta})_{\#}\mu, \, (P_{\phi_\theta})_{\#}\nu\big). \tag{40}$$

**Taking the supremum.** Now take the supremum over $\theta \in \mathbb{S}^{d-1}$:
$$\sup_\theta \Delta\big((P_\theta)_{\#}F_{\#}\mu, \, (P_\theta)_{\#}F_{\#}\nu\big) \leq \sup_\theta c(r_\theta) \sup_\phi \Delta\big((P_\phi)_{\#}\mu, \, (P_\phi)_{\#}\nu\big). \tag{41}$$

Since $r_\theta = \|A^\top \theta\| \leq \|A^\top\|_{\mathrm{op}} = \|A\|_{\mathrm{op}} = L$ and $c$ is nondecreasing,
$$\sup_\theta \Delta\big((P_\theta)_{\#}F_{\#}\mu, \, (P_\theta)_{\#}F_{\#}\nu\big) \leq c(L)\, \mathbf{MS}\Delta(\mu, \nu). \tag{42}$$

The left-hand side is exactly $\mathbf{MS}\Delta(F_{\#}\mu, F_{\#}\nu)$, which proves the claim. $\qquad \square$

**Lemma 11** (Max–sliced mixture $p$-convexity). *This result is the max–sliced analogue of Lemma 9.*

*Let $\Delta$ be a divergence on $\mathcal{P}(\mathbb{R})$ that is mixture $p$-convex for some $p \in [1, \infty)$: for every probability space $(\Omega, \mathcal{F}, \rho)$ and measurable families $(\mu_c), (\nu_c) \subset \mathcal{P}(\mathbb{R})$,*

$$\Delta\left(\int_\Omega \mu_c\, \rho(dc), \int_\Omega \nu_c\, \rho(dc)\right) \ \leq \ \left(\int_\Omega \Delta(\mu_c, \nu_c)^p\, \rho(dc)\right)^{1/p}.$$

*Define the max–sliced lift on $\mathcal{P}(\mathbb{R}^d)$ by*
$$\mathbf{MS}\Delta(\mu, \nu) := \sup_{\theta \in \mathbb{S}^{d-1}} \Delta\big((P_\theta)_{\#}\mu, \, (P_\theta)_{\#}\nu\big), \qquad P_\theta(x) = \langle \theta, x \rangle.$$

*Then* $\mathbf{MS}\Delta$ *is also mixture p-convex:*

$$\mathbf{MS}\Delta\left(\int_\Omega \mu_c\,\rho(dc),\ \int_\Omega \nu_c\,\rho(dc)\right) \leq \left(\int_\Omega \mathbf{MS}\Delta(\mu_c,\nu_c)^p\,\rho(dc)\right)^{1/p}.$$

*Proof.* Fix $\theta \in \mathbb{S}^{d-1}$ and set

$$\mu_c^\theta := (P_\theta)_{\#}\mu_c, \qquad \nu_c^\theta := (P_\theta)_{\#}\nu_c \ \in \mathcal{P}(\mathbb{R}).$$

Pushforward commutes with mixtures:

$$(P_\theta)_{\#}\left(\int \mu_c\,d\rho\right) = \int \mu_c^\theta\,d\rho, \qquad (P_\theta)_{\#}\left(\int \nu_c\,d\rho\right) = \int \nu_c^\theta\,d\rho.$$

By mixture $p$-convexity of $\Delta$ in one dimension,

$$\Delta\big((P_\theta)_{\#}\textstyle\int \mu_c\,d\rho,\ (P_\theta)_{\#}\int \nu_c\,d\rho\big) \leq \left(\int \Delta(\mu_c^\theta,\nu_c^\theta)^p\,d\rho\right)^{1/p}. \tag{43}$$

Taking the supremum over $\theta$ on the left-hand side of equation 43 gives

$$\sup_\theta \Delta\big((P_\theta)_{\#}\textstyle\int \mu_c\,d\rho,\ (P_\theta)_{\#}\int \nu_c\,d\rho\big) \leq \sup_\theta \left(\int \Delta(\mu_c^\theta,\nu_c^\theta)^p\,d\rho\right)^{1/p}. \tag{44}$$

Define $f(\theta,c) := \Delta(\mu_c^\theta,\nu_c^\theta)$ and $h(c) := \sup_\phi f(\phi,c) = \mathbf{MS}\Delta(\mu_c,\nu_c)$. Since $f(\theta,c) \leq h(c)$ pointwise in $c$, we obtain for every $\theta$,

$$\left(\int f(\theta,c)^p\,d\rho(c)\right)^{1/p} \leq \left(\int h(c)^p\,d\rho(c)\right)^{1/p}.$$

Taking $\sup_\theta$ yields

$$\sup_\theta \left(\int \Delta(\mu_c^\theta,\nu_c^\theta)^p\,d\rho\right)^{1/p} \leq \left(\int \mathbf{MS}\Delta(\mu_c,\nu_c)^p\,d\rho\right)^{1/p}. \tag{45}$$

Combining equation 44 and equation 45 shows

$$\mathbf{MS}\Delta\left(\int \mu_c\,d\rho,\ \int \nu_c\,d\rho\right) \leq \left(\int \mathbf{MS}\Delta(\mu_c,\nu_c)^p\,\rho(dc)\right)^{1/p},$$

as claimed. $\qquad\square$

**Theorem 4** (Supremum–max–sliced contraction of the multivariate distributional Bellman operator (anisotropic linear map))**.** *Let* $\Delta$ *be a divergence on* $\mathcal{P}(\mathbb{R})$ *and define the max–sliced lift on* $\mathcal{P}(\mathbb{R}^d)$ *by*

$$\mathbf{MS}\Delta(\mu,\nu) := \sup_{\theta \in \mathbb{S}^{d-1}} \Delta\big((P_\theta)_{\#}\mu,\ (P_\theta)_{\#}\nu\big), \qquad P_\theta(x) = \langle \theta, x\rangle.$$

*Assume* $\Delta$ *satisfies:*

(**T**) *Translation nonexpansion:* $\Delta\big((x \mapsto x+t)_{\#}\mu,\ (x \mapsto x+t)_{\#}\nu\big) \leq \Delta(\mu,\nu)$ *for all* $t \in \mathbb{R}$.

(**S**) *Scale–Lipschitz: there exists a nondecreasing* $c : [0,\infty) \to [0,\infty)$ *such that, for all* $s \geq 0$,

$$\Delta\big((x \mapsto sx)_{\#}\mu,\ (x \mapsto sx)_{\#}\nu\big) \leq c(s)\,\Delta(\mu,\nu).$$

(**M**$_p$) *Mixture p-convexity: for every probability space* $(\Omega, \mathcal{F}, \rho_0)$ *and measurable families* $(\mu_c), (\nu_c) \subset \mathcal{P}(\mathbb{R})$,

$$\Delta\left(\int \mu_c\,\rho_0(dc),\ \int \nu_c\,\rho_0(dc)\right) \leq \left(\int \Delta(\mu_c,\nu_c)^p\,\rho_0(dc)\right)^{1/p}, \qquad p \in [1,\infty).$$

**Bellman update (anisotropic linear map).** *Fix* $(s, a)$. *Gather all environment/policy randomness into a single random element $C$, which determines the successor index through a measurable mapping $g$:*

$$(S', A') := g(s, a; C).$$

*At $(s, a)$, apply an affine transformation with a $C$-dependent translation and an arbitrary $C$-dependent linear map:*

$$b_{s,a}(C) \in \mathbb{R}^d, \qquad A_{s,a}(C) \in \mathbb{R}^{d \times d}.$$

*Conditioned on $C$, the next sample is drawn from the law at the successor index,*

$$X' \mid C \sim \eta(S', A'),$$

*and the Bellman update is the push-forward of $X'$ by this affine map:*

$$(T^\pi \eta)(s, a) := \mathrm{Law}\big(b_{s,a}(C) + A_{s,a}(C)\, X'\big).$$

*Define, for each $C$,*

$$L(C) := \|A_{s,a}(C)\|_{\mathrm{op}},$$

*and the global envelope*

$$\bar{L} := \sup_{(s,a)} \sup_C L(C).$$

*Also define the supremum metric*

$$\overline{\mathbf{MS\Delta}}(\eta_1, \eta_2) := \sup_{(s,a)} \mathbf{MS\Delta}\big(\eta_1(s, a),\, \eta_2(s, a)\big).$$

*Then, for all $\eta_1, \eta_2$,*

$$\boxed{\overline{\mathbf{MS\Delta}}\big(T^\pi \eta_1,\ T^\pi \eta_2\big) \ \le\ c(\bar{L})\, \overline{\mathbf{MS\Delta}}\big(\eta_1,\ \eta_2\big).}$$

*Proof.* Fix $(s, a)$ and condition on $C$. Set

$$\Phi_{s,a}(x; C) := b_{s,a}(C) + A_{s,a}(C)\, x, \qquad Z_i := \Phi_{s,a}(X_i'; C),$$

with $X_i' \mid C \sim \eta_i(S', A')$. By Lemma 5,

$$(T^\pi \eta_i)(s, a) = \mathrm{Law}\big(\Phi_{s,a}(X_i'; C)\big) = \mathrm{Law}(Z_i).$$

**Affine push-forward at fixed $C$.** Applying Lemma 10, which relies on **(T)** and **(S)**, to the conditional laws $\mathrm{Law}(X_i' \mid C)$ gives

$$\mathbf{MS\Delta}\big(\mathrm{Law}(Z_1 \mid C),\, \mathrm{Law}(Z_2 \mid C)\big) \le c\big(L(C)\big)\, \mathbf{MS\Delta}\big(\mathrm{Law}(X_1' \mid C),\, \mathrm{Law}(X_2' \mid C)\big) \qquad (46)$$

$$= c\big(L(C)\big)\, \mathbf{MS\Delta}\big(\eta_1(S', A'),\, \eta_2(S', A')\big). \qquad (47)$$

**Averaging over $C$.** Lemma 11, which relies on **(M$_p$)**, together with equation 46 yields

$$\mathbf{MS\Delta}\big(\mathrm{Law}(Z_1),\, \mathrm{Law}(Z_2)\big) \le \left( \int \mathbf{MS\Delta}\big(\mathrm{Law}(Z_1 \mid C),\, \mathrm{Law}(Z_2 \mid C)\big)^p\, \rho(dC) \right)^{1/p} \qquad (48)$$

$$\le \left( \int \big(c\big(L(C)\big)\, \mathbf{MS\Delta}\big(\eta_1(S', A'), \eta_2(S', A')\big)\big)^p\, \rho(dC) \right)^{1/p} \qquad (49)$$

$$\le c(\bar{L}) \left( \int \mathbf{MS\Delta}\big(\eta_1(S', A'), \eta_2(S', A')\big)^p\, \rho(dC) \right)^{1/p}, \qquad (50)$$

since $c$ is nondecreasing and $L(C) \le \bar{L}$ for all $C$.

**Supremum bound.** For any realization of $C$, by definition of the supremum metric,

$$\mathbf{MS\Delta}\big(\eta_1(S', A'), \eta_2(S', A')\big) \ \le\ \sup_{(u,v)} \mathbf{MS\Delta}\big(\eta_1(u, v),\, \eta_2(u, v)\big) = \overline{\mathbf{MS\Delta}}(\eta_1, \eta_2).$$

Combining this with the previous inequality,

$$\mathbf{MS\Delta}\big((T^\pi \eta_1)(s,a),\, (T^\pi \eta_2)(s,a)\big) = \mathbf{MS\Delta}\big(\mathrm{Law}(Z_1),\, \mathrm{Law}(Z_2)\big) \tag{51}$$

$$\leq c(\bar{L}) \left( \int \overline{\mathbf{MS\Delta}}(\eta_1, \eta_2)^p \, \rho(dC) \right)^{1/p} \tag{52}$$

$$= c(\bar{L}) \, \overline{\mathbf{MS\Delta}}(\eta_1, \eta_2). \tag{53}$$

Taking the supremum over $(s,a)$ completes the proof:

$$\overline{\mathbf{MS\Delta}}(T^\pi \eta_1, T^\pi \eta_2) \;\leq\; c(\bar{L}) \, \overline{\mathbf{MS\Delta}}(\eta_1, \eta_2).$$

$\square$

**Lemma 12** (Fixed-point law of the distributional Bellman operator (general linear discount))**.** *Define the infinite–horizon return under policy $\pi$ recursively by*

$$Z(s,a) \;\overset{d}{=}\; \Phi_{s,a}\big(Z(S', A'); C\big),$$

*where $C$ collects the one–step randomness, $(S', A') = g(s,a; C)$ is the successor pair, and*

$$\Phi_{s,a}(x; C) := r(s,a; C) + \Gamma(s,a; C)\, x, \qquad r(s,a; C) \in \mathbb{R}^d, \;\; \Gamma(s,a; C) \in \mathbb{R}^{d\times d}.$$

*Equivalently, along a trajectory $(S_t, A_t)$ with one–step randomness $(C_t)_{t\geq 0}$, set*

$$r_t := r(S_t, A_t; C_t), \qquad \Gamma_t := \Gamma(S_t, A_t; C_t), \qquad \Pi_{0:t-1} := \Gamma_0 \Gamma_1 \cdots \Gamma_{t-1} \;\; (\Pi_{0:-1} := I_d),$$

*and, whenever the series converges,*

$$Z(s,a) \;=\; \sum_{t=0}^{\infty} \Pi_{0:t-1}\, r_t.$$

*Set*

$$\eta^\pi(s,a) := \mathrm{Law}\big(Z(s,a)\big) \in \mathcal{P}(\mathbb{R}^d).$$

$$\boxed{T_\pi \eta^\pi \;=\; \eta^\pi.}$$

*Proof.* By definition,

$$Z(s,a) \;\overset{d}{=}\; \Phi_{s,a}\big(Z(S', A'); C\big), \qquad (S', A') = g(s,a; C).$$

Conditioning on $C$ gives

$$Z(S', A') \,\big|\, C \;\sim\; \eta^\pi(S', A').$$

By the push–forward law (Lemma 4),

$$\mathrm{Law}\big(Z(s,a)\big) = \mathrm{Law}\big(\Phi_{s,a}(X'; C)\big), \qquad X' \,\big|\, C \sim \eta^\pi(S', A').$$

By definition of the distributional Bellman operator, $(T_\pi \eta^\pi)(s,a) \;=\; \mathrm{Law}\big(\Phi_{s,a}(X'; C)\big)$, hence $(T_\pi \eta^\pi)(s,a) = \eta^\pi(s,a)$ for all $(s,a)$. $\square$

**Theorem 5** (Convergence of sliced / max–sliced evaluation iterates)**.** *Under the conditions of either Theorem 3 or Theorem 4, let $\kappa$ denote the corresponding contraction constant (e.g. $\kappa = c(\gamma)$ in the scaled–isometry sliced case, or $\kappa = c(\bar{L})$ in the anisotropic max–sliced case), and assume $\kappa < 1$.*

*For any initial return–distribution function $\eta_0$, define the iteration*

$$\eta_{n+1} = T_\pi \eta_n,$$

*where $T_\pi$ is the chosen evaluation operator (sliced $T_\pi^{\mathbf{S}}$ or max–sliced $T_\pi^{\mathbf{MS}}$). Then, by Banach's fixed-point theorem, the iterates converge to the unique fixed point $\eta^\pi$ (cf. Lemma 12):*

$$\overline{\mathbf{S\Delta}}^{\rho,p}\big(\eta_n,\, \eta^\pi\big) \;\leq\; \kappa^n\, \overline{\mathbf{S\Delta}}^{\rho,p}\big(\eta_0,\, \eta^\pi\big) \;\xrightarrow[n\to\infty]{}\; 0 \quad \text{(sliced case)},$$

*and*

$$\overline{\mathbf{MS\Delta}}\big(\eta_n,\, \eta^\pi\big) \;\leq\; \kappa^n\, \overline{\mathbf{MS\Delta}}\big(\eta_0,\, \eta^\pi\big) \;\xrightarrow[n\to\infty]{}\; 0 \quad \text{(max–sliced case)}.$$

*In particular, $\eta_n \to \eta^\pi$ in the corresponding supremum metric.*

## C.3 SAMPLE COMPLEXITY

### C.3.1 UNIFORM SLICING

**Theorem 6** (Sample complexity of sliced divergences). *This is a rewrite of Theorem 5 from Nadjahi et al. (2020).*

*Fix $p \in [1, \infty)$. Let $\Delta$ be a divergence on $\mathcal{P}(\mathbb{R})$ and assume there exists a function $\alpha(p, n) \geq 0$ such that for every $\mu \in \mathcal{P}(\mathbb{R})$ with empirical $\hat{\mu}_n$,*

$$\mathbb{E}\big[\Delta(\hat{\mu}_n, \mu)^p\big] \leq \alpha(p, n).$$

*For $\mu, \nu \in \mathcal{P}(\mathbb{R}^d)$, define*

$$\mathbf{S}\Delta_p(\mu, \nu) := \left( \int_{S^{d-1}} \Delta^p\big((P_\theta)_\# \mu, (P_\theta)_\# \nu\big) \, d\sigma(\theta) \right)^{1/p},$$

*where $P_\theta(x) = \langle \theta, x \rangle$ and $\sigma$ is the uniform probability measure on $S^{d-1}$. Then:*

(i) *For any $\mu \in \mathcal{P}(\mathbb{R}^d)$ with empirical $\hat{\mu}_n$,*

$$\mathbb{E}\big|\mathbf{S}\Delta_p^p(\hat{\mu}_n, \mu)\big| \leq \alpha(p, n).$$

(ii) *If $\Delta$ verifies nonnegativity, symmetry, and the triangle inequality on $\mathcal{P}(\mathbb{R})$ (hence $\mathbf{S}\Delta_p$ verifies them on $\mathcal{P}(\mathbb{R}^d)$ by Proposition 1), then for any $\mu, \nu \in \mathcal{P}(\mathbb{R}^d)$ with empirical measures $\hat{\mu}_n, \hat{\nu}_n$,*

$$\mathbb{E}\big|\mathbf{S}\Delta_p(\mu, \nu) - \mathbf{S}\Delta_p(\hat{\mu}_n, \hat{\nu}_n)\big| \leq 2\,\alpha(p, n)^{1/p}.$$

*Proof.* **(i) One-sample bound for $\mathbf{S}\Delta_p^p$.**

$$\mathbb{E}\big|\mathbf{S}\Delta_p^p(\hat{\mu}_n, \mu)\big| = \mathbb{E}\left| \int_{S^{d-1}} \Delta^p\big((P_\theta)_\# \hat{\mu}_n, (P_\theta)_\# \mu\big) \, d\sigma(\theta) \right|$$

$$\leq \mathbb{E} \int_{S^{d-1}} \big|\Delta^p\big((P_\theta)_\# \hat{\mu}_n, (P_\theta)_\# \mu\big)\big| \, d\sigma(\theta) \quad \text{(triangle inequality for the integral)}$$

$$= \int_{S^{d-1}} \mathbb{E}\big|\Delta^p\big((P_\theta)_\# \hat{\mu}_n, (P_\theta)_\# \mu\big)\big| \, d\sigma(\theta) \quad \text{(Tonelli)}$$

$$= \int_{S^{d-1}} \mathbb{E}\,\Delta^p\big((P_\theta)_\# \hat{\mu}_n, (P_\theta)_\# \mu\big) \, d\sigma(\theta) \quad \text{(non-negativity)}$$

$$\leq \int_{S^{d-1}} \alpha(p, n) \, d\sigma(\theta) = \alpha(p, n).$$

**(ii) Two-sample bound for $\mathbf{S}\Delta_p$.** By Proposition 1 (triangle–inequality item), the triangle inequality for $\Delta$ on $\mathcal{P}(\mathbb{R})$ implies that $\mathbf{S}\Delta_p$ satisfies the triangle inequality on $\mathcal{P}(\mathbb{R}^d)$. Hence

$$\big|\mathbf{S}\Delta_p(\mu, \nu) - \mathbf{S}\Delta_p(\hat{\mu}_n, \hat{\nu}_n)\big| \leq \big|\mathbf{S}\Delta_p(\hat{\mu}_n, \mu)\big| + \big|\mathbf{S}\Delta_p(\hat{\nu}_n, \nu)\big| \quad \text{(triangle inequality)}$$

$$= \mathbf{S}\Delta_p(\hat{\mu}_n, \mu) + \mathbf{S}\Delta_p(\hat{\nu}_n, \nu) \quad \text{(non-negativity)}.$$

Taking expectations with respect to the empirical draws $(\hat{\mu}_n, \hat{\nu}_n)$,

$$\mathbb{E}\big|\mathbf{S}\Delta_p(\mu, \nu) - \mathbf{S}\Delta_p(\hat{\mu}_n, \hat{\nu}_n)\big| \leq \mathbb{E}\big|\mathbf{S}\Delta_p(\hat{\mu}_n, \mu)\big| + \mathbb{E}\big|\mathbf{S}\Delta_p(\hat{\nu}_n, \nu)\big|.$$

Since $x \mapsto x^{1/p}$ is concave for $p \geq 1$, Jensen's inequality gives

$$\mathbb{E}\big|\mathbf{S}\Delta_p(\hat{\mu}_n, \mu)\big| \leq \big\{\mathbb{E}\big|\mathbf{S}\Delta_p(\hat{\mu}_n, \mu)\big|^p\big\}^{1/p} = \big\{\mathbb{E}\,\mathbf{S}\Delta_p^p(\hat{\mu}_n, \mu)\big\}^{1/p},$$

$$\mathbb{E}\big|\mathbf{S}\Delta_p(\hat{\nu}_n, \nu)\big| \leq \big\{\mathbb{E}\big|\mathbf{S}\Delta_p(\hat{\nu}_n, \nu)\big|^p\big\}^{1/p} = \big\{\mathbb{E}\,\mathbf{S}\Delta_p^p(\hat{\nu}_n, \nu)\big\}^{1/p}.$$

Applying the bound from part (i) to both terms,

$$\mathbb{E}\big|\mathbf{S}\Delta_p(\mu, \nu) - \mathbf{S}\Delta_p(\hat{\mu}_n, \hat{\nu}_n)\big| \leq \alpha(p, n)^{1/p} + \alpha(p, n)^{1/p} = 2\,\alpha(p, n)^{1/p}.$$

$\square$

### C.3.2 MAX SLICING

**Lemma 13** (Half–spaces and CDFs of projections). *As noted in the proof of Theorem 4 of Nguyen et al. (2020a), the CDF of a projection can be written as the probability of a half–space.*

*Let $P \in \mathcal{P}(\mathbb{R}^d)$ and $X_1, \ldots, X_n \overset{iid}{\sim} P$, with empirical measure $P_n = \frac{1}{n} \sum_{i=1}^n \delta_{X_i}$. For $\theta \in \mathbb{S}^{d-1}$ and $t \in \mathbb{R}$, define the half–space*

$$H_{\theta,t} := \{x \in \mathbb{R}^d : \langle \theta, x \rangle \le t\}.$$

*We also write $P_\theta(x) = \langle \theta, x \rangle$ for the one–dimensional projection map. Then, for all $t \in \mathbb{R}$, the CDF of the projection $(P_\theta)_\# P$ is*

$$F_\theta(t) = (P_\theta)_\# P((-\infty, t]) = P(H_{\theta,t}),$$

*while the empirical CDF of the projection $(P_\theta)_\# P_n$ is*

$$F_{n,\theta}(t) = (P_\theta)_\# P_n((-\infty, t]) = P_n(H_{\theta,t}) = \frac{1}{n} \sum_{i=1}^n \mathbf{1}\{\langle \theta, X_i \rangle \le t\}.$$

*Proof.* By definition of the pushforward, for any Borel $A \subseteq \mathbb{R}$,

$$(P_\theta)_\# P(A) = P\big(\{x \in \mathbb{R}^d : P_\theta(x) \in A\}\big).$$

Taking $A = (-\infty, t]$ yields

$$F_\theta(t) = (P_\theta)_\# P((-\infty, t]) = P\big(\{x : \langle \theta, x \rangle \le t\}\big) = P(H_{\theta,t}).$$

The same argument with $P$ replaced by $P_n$ gives

$$F_{n,\theta}(t) = (P_\theta)_\# P_n((-\infty, t]) = P_n(H_{\theta,t}).$$

Finally, since $P_n$ is the empirical measure,

$$P_n(H_{\theta,t}) = \frac{1}{n} \sum_{i=1}^n \mathbf{1}\{\langle \theta, X_i \rangle \le t\}.$$

$\square$

**Lemma 14** (VC inequality for half–spaces in $\mathbb{R}^d$). *Let $P \in \mathcal{P}(\mathbb{R}^d)$, let $X_1, \ldots, X_n \overset{iid}{\sim} P$ with empirical measure $P_n = \frac{1}{n} \sum_{i=1}^n \delta_{X_i}$, and let*

$$\mathcal{H} = \big\{H_{\theta,t} = \{x \in \mathbb{R}^d : \langle \theta, x \rangle \le t\} : \theta \in \mathbb{S}^{d-1}, t \in \mathbb{R}\big\}.$$

*Define*

$$Z := \sup_{H \in \mathcal{H}} \big|P_n(H) - P(H)\big| = \sup_{\theta \in \mathbb{S}^{d-1}, t \in \mathbb{R}} \big|P_n(H_{\theta,t}) - P(H_{\theta,t})\big|.$$

*Then, for any $\delta \in (0, 1)$,*

$$\Pr\big(Z \le c_{n,\delta}\big) \ge 1 - \delta, \qquad c_{n,\delta} := \sqrt{\frac{32}{n}\Big((d+1)\log(n+1) + \log \frac{8}{\delta}\Big)}.$$

*This is the explicit VC bound used in the proof of Theorem 4 of Nguyen et al. (2020a).*

**Theorem 7** (Max–sliced bound from a 1D CDF control, in expectation). *Let $P \in \mathcal{P}(\mathbb{R}^d)$ and $X_1, \ldots, X_n \overset{iid}{\sim} P$ with empirical measure $P_n = \frac{1}{n} \sum_{i=1}^n \delta_{X_i}$. Assume $\mathrm{diam}(\mathrm{supp}\, P) \le D$ (so for every $\theta$, the range of $x \mapsto \langle \theta, x \rangle$ over $\mathrm{supp}\, P$ has length $\le D$). Let $\Delta$ be a divergence on $\mathcal{P}(\mathbb{R})$ such that for any one–dimensional laws $\mu, \nu$ supported on an interval of length $\le D$ there exist*

$$\alpha \in (0, 1], \qquad \beta \ge 0, \qquad L > 0$$

*with the* CDF–dominance *inequality*

$$\Delta(\mu, \nu) \le L\, D^\beta\, \|F_\mu - F_\nu\|_\infty^\alpha. \tag{A}$$

*Define*

$$\mathbf{MS}\Delta(\mu, \nu) := \sup_{\theta \in \mathbb{S}^{d-1}} \Delta\big((P_\theta)_\# \mu, \, (P_\theta)_\# \nu\big), \qquad P_\theta(x) = \langle \theta, x \rangle.$$

*Then*

$$\mathbb{E}\,\mathbf{MS}\Delta(P_n, P) = \mathcal{O}\Big(D^\beta \big(\tfrac{d \log n}{n}\big)^{\alpha/2}\Big).$$

*More precisely, there exists a constant $C_\Delta$ depending only on $L$ and $\alpha$ such that*

$$\mathbb{E}\,\mathbf{MS}\Delta(P_n, P) \;\le\; L\,D^\beta \bigg(\sqrt{\tfrac{32(d+1)\log(n+1)}{n}} + 4\sqrt{\tfrac{32\pi}{n}}\bigg)^\alpha \;\le\; C_\Delta\,D^\beta \bigg(\sqrt{\tfrac{d \log(n+1)}{n}}\bigg)^\alpha.$$

*Proof.* Let

$$Z \;:=\; \sup_{\theta \in \mathbb{S}^{d-1},\, t \in \mathbb{R}} \big|F_{n,\theta}(t) - F_\theta(t)\big|,$$

where Lemma 13 identifies $F_{n,\theta}(t) = P_n(H_{\theta,t})$ and $F_\theta(t) = P(H_{\theta,t})$. By (A), for each $\theta$,

$$\Delta\big((P_\theta)_\# P_n, (P_\theta)_\# P\big) \;\le\; L\,D^\beta \,\|F_{n,\theta} - F_\theta\|_\infty^\alpha,$$

hence, after taking $\sup_\theta$,

$$\mathbf{MS}\Delta(P_n, P) \;\le\; L\,D^\beta\,Z^\alpha.$$

Taking expectations and using Jensen (concavity of $x \mapsto x^\alpha$ for $\alpha \in (0,1]$),

$$\mathbb{E}\,\mathbf{MS}\Delta(P_n, P) \;\le\; L\,D^\beta\,\mathbb{E}[Z^\alpha] \;\le\; L\,D^\beta\,(\mathbb{E}Z)^\alpha.$$

By Lemma 14, for any $\delta \in (0,1)$, $\Pr(Z \le c_{n,\delta}) \ge 1 - \delta$ with $c_{n,\delta}$ as stated. Put $b_n := \sqrt{32(d+1)\log(n+1)/n}$ and take $\delta = 8e^{-ns^2/32}$ so that $c_{n,\delta} \le b_n + s$ and $\Pr(Z > b_n + s) \le 8e^{-ns^2/32}$ for all $s \ge 0$. Integrating the tail,

$$\mathbb{E}Z \;=\; \int_0^\infty \Pr(Z > t)\,dt \;\le\; b_n + \int_0^\infty 8e^{-ns^2/32}\,ds \;=\; b_n + 4\sqrt{\tfrac{32\pi}{n}}.$$

Insert this into the previous display and absorb numerical constants into $C_\Delta$ to obtain the claim. $\square$

**Corollary 7.1** (Max–sliced $\mathbf{W}_1$). If $\Delta = \mathbf{W}_1$ (one-dimensional Wasserstein–1), then

$$\mathbb{E}\,\mathbf{MSW}_1(P_n, P) = \mathcal{O}\Big(D\,\sqrt{\tfrac{d \log n}{n}}\Big).$$

*Proof.* By Vallender's identity (Vallender, 1974), for probability laws $\alpha, \beta$ on $\mathbb{R}$ with CDFs $F_\alpha, F_\beta$,

$$\mathbf{W}_1(\alpha, \beta) \;=\; \int_{\mathbb{R}} \big|F_\alpha(x) - F_\beta(x)\big|\,dx.$$

If the support of $\alpha$ and $\beta$ lies within an interval of length $D$, then

$$\int_{\mathbb{R}} \big|F_\alpha(x) - F_\beta(x)\big|\,dx \;\le\; D\,\|F_\alpha - F_\beta\|_\infty.$$

Hence

$$\mathbf{W}_1(\alpha, \beta) \;\le\; D\,\|F_\alpha - F_\beta\|_\infty,$$

which verifies condition (A) with $(\alpha, \beta, L) = (1, 1, 1)$. Applying Theorem 7 concludes the proof. $\square$

**Corollary 7.2** (Max–sliced $\mathbf{W}_p$ for $p > 1$). Fix $p > 1$ and $\Delta = \mathbf{W}_p$. Then

$$\mathbb{E}\,\mathbf{MSW}_p(P_n, P) = \mathcal{O}\Big(D\,\big(\tfrac{d \log n}{n}\big)^{1/(2p)}\Big).$$

*Proof.* By the 1D quantile representation,

$$\mathbf{W}_p^p(\alpha, \beta) = \int_0^1 \left| F_\alpha^{-1}(u) - F_\beta^{-1}(u) \right|^p du.$$

If $\alpha, \beta$ are supported on an interval of length $D$, then every quantile difference $F_\alpha^{-1}(u) - F_\beta^{-1}(u)$ lies in $[-D, D]$. Hence, for $x = F_\alpha^{-1}(u) - F_\beta^{-1}(u)$,

$$|x|^p = |x|^{p-1} |x| \ \leq\ D^{p-1} |x|.$$

Applying this bound inside the integral gives

$$\mathbf{W}_p^p(\alpha, \beta) \ \leq\ D^{p-1} \int_0^1 \left| F_\alpha^{-1}(u) - F_\beta^{-1}(u) \right| du.$$

The integral on the right is exactly the 1D Wasserstein–1 distance,

$$\int_0^1 \left| F_\alpha^{-1}(u) - F_\beta^{-1}(u) \right| du \ =\ \mathbf{W}_1(\alpha, \beta).$$

Hence

$$\mathbf{W}_p^p(\alpha, \beta) \ \leq\ D^{p-1} \mathbf{W}_1(\alpha, \beta).$$

By Vallender's identity (Vallender, 1974) and the support bound of length $D$, we already established in Corollary 7.1 that

$$\mathbf{W}_1(\alpha, \beta) \ \leq\ D \, \|F_\alpha - F_\beta\|_\infty.$$

Combining the two inequalities yields

$$\mathbf{W}_p^p(\alpha, \beta) \ \leq\ D^p \, \|F_\alpha - F_\beta\|_\infty.$$

Taking the $p$-th root finally gives

$$\mathbf{W}_p(\alpha, \beta) \ \leq\ D \, \|F_\alpha - F_\beta\|_\infty^{1/p}.$$

Thus condition (A) holds with $(\alpha, \beta, L) = (1/p, 1, 1)$, and Theorem 7 applies. $\square$

**Corollary 7.3** (Max–sliced Cramér)**.** Let $\Delta(\alpha, \beta) = \|F_\alpha - F_\beta\|_{L^2(\mathbb{R})}$. Then

$$\boxed{\mathbb{E}\,\mathbf{MSC}_2(\hat{\mu}_n, \mu) = \mathcal{O}\!\left( \sqrt{D} \, \sqrt{\tfrac{d \log n}{n}} \right).}$$

*Proof.* On an interval of length $D$, one has $\| \cdot \|_{L^2} \leq D^{1/2} \| \cdot \|_\infty$, so (A) holds with $(\alpha, \beta, L) = (1, 1/2, 1)$. Applying Theorem 7 yields the result. $\square$

### C.4 INSTANTIATIONS

#### C.4.1 WASSERSTEIN

Wasserstein is a metric on $\mathcal{P}_p(\mathbb{R})$ (Proposition 2 in Givens & Shortt (1984)). It satisfies **(T)** as it is translation invariant, and **(S)** with $c(s) = s$ due to the exact scaling law

$$\mathbf{W}_p\big((S_s)_{\#}\mu, \ (S_s)_{\#}\nu\big) \ =\ s\,\mathbf{W}_p(\mu, \nu),$$

as established in Proposition 1. It also satisfies **(M$_p$)** by mixture $p$–convexity (Proposition 2).

*Contraction factors.* By Theorem 3 with $\Delta = \mathbf{W}_p$ (so $c(s) = s$), the *sliced Wasserstein* update with $A = \gamma O$ contracts with factor $\gamma < 1$:

$$\mathbf{SW}_p\big(T^\pi \eta_1, \ T^\pi \eta_2\big) \ \leq\ \gamma \, \mathbf{SW}_p(\eta_1, \eta_2).$$

By Theorem 4, the *max–sliced Wasserstein* update with a general linear map $A$ contracts with factor $\bar{L} = \sup \|A\|_{\mathrm{op}}$, strictly so whenever $\bar{L} < 1$:

$$\mathbf{MSW}_p\big(T^\pi \eta_1, \ T^\pi \eta_2\big) \ \leq\ \bar{L} \, \mathbf{MSW}_p(\eta_1, \eta_2).$$

*Sample complexity (uniform slicing).* Let $p \in [1, \infty)$ and assume $\mu \in \mathcal{P}_q(\mathbb{R}^d)$ with $q > 2p$ (finite $q$-th moment). Let $\hat{\mu}_n$ be the empirical measure from $n$ samples. Carrying the same steps as in Corollary 2 of Nadjahi et al. (2020) but in the one-sample setting, and plugging the 1D base bound from Theorem 1 of Fournier & Guillin (2015), we obtain the dimension–free rate

$$\mathbb{E}[\mathbf{SW}_p(\hat{\mu}_n, \mu)] \;=\; \mathcal{O}\!\left(n^{-1/(2p)}\right).$$

Thus, uniform slicing avoids the curse of dimensionality.

*Sample complexity (max–sliced).* By Theorem 7 and Corollaries 7.1–7.2, for $\mathrm{diam}(\mathrm{supp}\,\mu) \leq D$,

$$\mathbb{E}\,\mathbf{MS}W_1(\hat{\mu}_n, \mu) = O\!\left(D\,\sqrt{\tfrac{d \log n}{n}}\right), \qquad \mathbb{E}\,\mathbf{MS}W_p(\hat{\mu}_n, \mu) = O\!\left(D\left(\tfrac{d \log n}{n}\right)^{1/(2p)}\right) \quad (p > 1).$$

### C.4.2   Cramér

Cramér (the $L^2$ distance between CDFs) enjoys all the structural assumptions we require. By Proposition 3, it is a metric. It satisfies (T) by Proposition 2 in Bellemare et al. (2017b) and Proposition 3.2 in Odin & Charpentier (2020), and (S) with $c(s) = s^{1/2}$ via Proposition 4. It also satisfies $(\mathbf{M}_p)$ (Proposition 5).

*Contraction factors.* By Theorem 3 with $\Delta = \mathbf{C}_2$ (so $c(s) = s^{1/2}$), the *sliced Cramér* update with $A = \gamma O$ contracts with factor $\gamma^{1/2}$:

$$\mathbf{SC}_2\!\left(T^\pi \eta_1, T^\pi \eta_2\right) \;\leq\; \gamma^{1/2}\,\mathbf{SC}_2(\eta_1, \eta_2).$$

By Theorem 4, the *max–sliced Cramér* update with a general linear map $A$ contracts with factor $c(\bar{L}) = \bar{L}^{1/2}$, strictly so whenever $\bar{L} < 1$:

$$\mathbf{MSC}_2\!\left(T^\pi \eta_1, T^\pi \eta_2\right) \;\leq\; \bar{L}^{1/2}\,\mathbf{MSC}_2(\eta_1, \eta_2).$$

*Sample complexity (uniform slicing).* For the one–dimensional Cramér distance (the $L^2$–CDF discrepancy), it is standard that

$$\mathbb{E}\,\|F_n - F\|_{L^2(F)} \;=\; \mathcal{O}\!\left(n^{-1/2}\right).$$

Plugging this base rate into Theorem 6 yields the dimension–free bound

$$\mathbb{E}[\mathbf{SC}_2(\hat{\mu}_n, \mu)] \;=\; \mathcal{O}\!\left(n^{-1/2}\right).$$

Thus, uniform slicing avoids the curse of dimensionality.

*Sample complexity (max–sliced).* By Theorem 7 and Corollary 7.3, for $\mathrm{diam}(\mathrm{supp}\,\mu) \leq D$,

$$\mathbb{E}[\mathbf{MSC}_2(\hat{\mu}_n, \mu)] \;=\; \mathcal{O}\!\left(\sqrt{D}\,\sqrt{\tfrac{d \log n}{n}}\right).$$

### C.4.3   MMD

The Maximum Mean Discrepancy (MMD) with a conditionally strictly positive definite kernel (?Sejdinovic et al., 2013) is a valid metric on probability laws. With the multiquadric (MQ) kernel $k_h(x, y) = -\sqrt{1 + h^2 \|x - y\|^2}$, it enjoys all the structural assumptions we require. By Proposition 6 and Proposition 7, it is a metric. It satisfies (T) since MMD is translation invariant for all shift–invariant kernels. It satisfies (S) with $c(s) = \max\{\sqrt{s}, s\}$ for the MQ kernel (Proposition 8), reflecting its scale–sensitivity. Finally, it satisfies $(\mathbf{M}_p)$ by mixture convexity of RKHS embeddings (Proposition 9).

*Contraction factors.* By Theorem 3 with $\Delta = \mathbf{MMD}_{k_h}$ and the scale bound

$$c(s) = \max\{\sqrt{s},\, s\},$$

the *sliced MMD* update with $A = \gamma O$ satisfies

$$\mathbf{SMMD}_{k_h}\!\left(T^\pi \eta_1, T^\pi \eta_2\right) \;\leq\; c(\gamma)\,\mathbf{SMMD}_{k_h}(\eta_1, \eta_2).$$

In particular, for scalar discounts $\gamma \in (0,1)$ we have $c(\gamma) = \sqrt{\gamma}$.

By Theorem 4, the *max–sliced MMD* update with a general linear map $A$ satisfies

$$\mathbf{MSMMD}_{k_h}\big(T^\pi \eta_1, \, T^\pi \eta_2\big) \;\leq\; c(\bar{L}) \, \mathbf{MSMMD}_{k_h}\big(\eta_1, \, \eta_2\big), \qquad c(\bar{L}) = \max\{\sqrt{\bar{L}}, \, \bar{L}\}.$$

In particular, under $\bar{L} < 1$ this reduces to $c(\bar{L}) = \sqrt{\bar{L}}$.

*Sample complexity (uniform slicing).* In one dimension, the unbiased empirical MMD (equivalently, the energy distance) is a U–statistic (Gretton et al., 2012; Sejdinovic et al., 2013), so classical U–statistic theory yields the standard rate

$$\mathbb{E}[\mathbf{MMD}_{k_h}(\hat{\mu}_n, \mu)] \;=\; \mathcal{O}\Big(n^{-1/2}\Big).$$

Plugging this into Theorem 6 yields the dimension–free bound

$$\mathbb{E}[\mathbf{SMMD}_{k_h}(\hat{\mu}_n, \mu)] \;=\; \mathcal{O}\Big(n^{-1/2}\Big).$$

Thus, uniform slicing avoids the curse of dimensionality.

*Sample complexity (max–sliced).* We were not able to establish a sharp sample complexity bound for the max–sliced MMD. Deriving such a result remains an open problem for future work.

## D PSEUDO-CODES

---

**Algorithm 2:** Estimation of MS$\Delta$ from empirical samples

---

**Input:** Empirical samples $X = \{x_i\}_{i=1}^N \subset \mathbb{R}^d$, $Y = \{y_i\}_{i=1}^N \subset \mathbb{R}^d$

**Input:** Base 1D divergence $\Delta$; gradient steps $T$; step size $\eta$

**Initialize a unit direction:** $w \sim \mathcal{N}(0, I_d)$; $\quad \theta \leftarrow w/\|w\|$ `// random unit direction`

**Project–optimize over directions: for** $t = 1, \ldots, T$ **do**

$\quad\quad u_i \leftarrow \langle \theta, x_i \rangle, \quad v_i \leftarrow \langle \theta, y_i \rangle \quad$ for $i = 1, \ldots, N \quad\quad$ `// project to 1D along` $\theta$

$\quad\quad J(\theta) \leftarrow \Delta\big(\{u_i\}_{i=1}^N, \{v_i\}_{i=1}^N\big) \quad\quad\quad$ `// objective to maximize over` $\theta$

$\quad\quad g \leftarrow \nabla_\theta J(\theta) \quad\quad\quad\quad\quad\quad$ `// gradient w.r.t. direction`

$\quad\quad w \leftarrow w + \eta\, g \quad\quad\quad\quad$ `// ascent step in unconstrained space`

$\quad\quad \theta \leftarrow w/\|w\| \quad\quad\quad\quad$ `// re-normalize onto the unit sphere`

$\bar{\theta} \leftarrow \text{stop\_grad}(\theta) \quad\quad\quad\quad$ `// stop gradient on the final direction`

**Output:** $\widehat{\text{MS}\Delta}(X, Y) \leftarrow \Delta\big(\{\langle \bar{\theta}, x_i \rangle\}_{i=1}^N, \{\langle \bar{\theta}, y_i \rangle\}_{i=1}^N\big)$

---

# E EXPERIMENTAL SETUP

## E.1 MULTI-OBJECTIVE ENVIRONMENTS

In MO-Gymnasium (Felten et al., 2023), the reward space is vector-valued. The standard Gymnasium (Towers et al., 2024) scalar reward is recovered through a linear scalarization with fixed weights:

- **MO-Humanoid**
  - Reward space: $(r_{\text{forward}}, r_{\text{control}})$
  - Scalarization (Humanoid-v5):
  $$r = 1.25 \times r_{\text{forward}} + 0.1 \times r_{\text{control}}.$$

- **MO-Hopper**
  - Reward space: $(r_{\text{forward}}, r_{\text{height}}, r_{\text{control}})$
  - Scalarization (Hopper-v5):
  $$r = 1.0 \times r_{\text{forward}} + 0.0 \times r_{\text{height}} + 10^{-3} \times r_{\text{control}}.$$

- **MO-Ant**
  - Reward space: $(r_{x\text{-vel}}, r_{y\text{-vel}}, r_{\text{control}})$
  - Scalarization (Ant-v5, cost merged):
  $$r = 1.0 \times r_{x\text{-vel}} + 0.0 \times r_{y\text{-vel}}.$$

- **MO-HalfCheetah**
  - Reward space: $(r_{\text{forward}}, r_{\text{control}})$
  - Scalarization (HalfCheetah-v5):
  $$r = 1.0 \times r_{\text{forward}} + 0.1 \times r_{\text{control}}.$$

- **MO-Walker2D**
  - Reward space: $(r_{\text{forward}}, r_{\text{control}})$
  - Scalarization (Walker2d-v5):
  $$r = 1.0 \times r_{\text{forward}} + 10^{-3} \times r_{\text{control}}.$$

- **MO-Reacher**
  - Reward space: $(r_1, r_2, r_3, r_4)$ with
  $$r_i = 1 - 4 \times \|\text{finger\_tip} - \text{target}_i\|^2, \quad i \in \{1, 2, 3, 4\}.$$
  - Scalarization (Reacher-v4):
  $$r = r_1 + r_2 + r_3 + r_4.$$

## E.2 MULTI-HORIZON RL

| $N$ (heads) | $k$ | $\gamma_{\max}$ | Integral rule |
|---|---|---|---|
| 32 | 0.01 | 0.997 | lower Riemann |

Table 2: Hyperparameters for hyperbolic discounting experiments in MuJoCo. We use $N$ parallel heads trained with Bellman discounts $\gamma_i^k$, where $\{\gamma_i\}$ form a power-law grid up to $\gamma_{\max}$. The heads are combined into a hyperbolic $Q$ via a left Riemann sum approximation. We refer to Fedus et al. (2019) for the meaning of these hyperparameters.

## E.3 ARCHITECTURES AND HYPERPARAMETERS

**Critic**

**Actor**

**General hyperparameters**

# F  LLM USAGE

We used an LLM-based assistant to support the preparation of this paper. In particular, it was employed to (i) rephrase draft paragraphs for clarity and suggest alternative framings of related work, (ii) format proofs, explore directions, and verify intermediate steps, (iii) assist in debugging code, (iv) suggest LaTeX equation formatting, and (v) help identify relevant theoretical results in preceding works. All core research contributions, including the development of theoretical results, algorithms, and experiments, were carried out by the authors.

