# OpenReview forum: "Sliced Distributional Reinforcement Learning"
_ICLR.cc/2026/Conference — ICLR 2026 Conference Withdrawn Submission_

### Official Review · Reviewer_oGpL · 2025-10-22

**Soundness:** 2
**Presentation:** 2
**Contribution:** 3
**Rating:** 4
**Confidence:** 4

**Summary:**

The paper presents a novel and principled framework for tackling the challenging problem of Multivariate Distributional Reinforcement Learning (MDRL). The core innovation is the application of Sliced Probability Divergences (SPD)  to compare deal with high-dimensional return distributions efficiently. The authors provide strong theoretical foundations, including the contraction guarantees for general matrix-discounted Bellman operators, and back their claims with empirical results on MuJoCo benchmarks.

**Strengths:**

The paper’s primary strength lies in its theoretical results. The authors demonstrate convergence properties and provide finite sample error bounds. In particular, the introduction of uniform slicing is shown to effectively mitigate the curse of dimensionality.

**Weaknesses:**

The paper lacks a formal definition of some notations, which could hinder readers' understanding. Specifically, the notation $\mathbb{S}$ is used without a clear definition , and the term $A_{s,a}$ in Theorem 4 is not defined. Additionally, in Theorem 4, if the authors consider the scenario where the range of $C$ (this term depends on the reward?) is unbounded. It would be beneficial to clarify these notations and explicitly discuss the unbounded reward case.

While the paper introduces a technically sound framework, the motivation behind using distributional reinforcement learning (RL) for multi-reward settings could be presented more convincingly. The practical significance of the approach is not fully articulated.

I would be willing to raise my score if the authors address my confusion regarding these points.

**Questions:**

1. Does the proposed framework account for the influence of correlated reward functions? If so, how is this incorporated into the framework?

2. The authors mention that the implementation of Maximum Mean Discrepancy (MMD) focuses on a single kernel. However, as far as I know, in distributional RL, as in [1], multiple bandwidths are often used, as a single bandwidth can degrade performance. Additionally, kernel methods are known to be sensitive to hyperparameter choices. Could the authors comment on these?

3. If the authors consider sliced Wasserstein distance or Cramer distance. I know there are rarely related work deal with multi-objective using these two metrics. if the authors have any comments?

4. The architecture of $Z_{\phi}$ is introduced without much detail. Considering that it is a generative model, did the authors consider using particular generative models, such as diffusion models or GANs? Further elaboration on this would be useful.

5. The authors employ a scalarization rule that computes a weighted average of the multiple rewards. Is this rule specific to the MO-MuJoCo environments, or could it be generalized? I note that in the standard MuJoCo setup, the reward is a linear combination of multiple reward components.

6. The paper introduces a distributional version of the multi-horizon Bellman equation, which seems to be a natural extension of the expected multi-horizon Bellman equation. However, I’m curious about how the framework handles the case where two rewards are highly correlated (either positively or negatively). Does the proposed equation still hold in such cases? Does it have convergence properties, particularly regarding fixed points?



   [1] Distributional reinforcement learning via moment matching.

**Details Of Ethics Concerns:**

No ethical concerns are apparent in the paper.

---

> ### Author Response · Authors · 2025-12-03
>
> We thank the reviewer for their detailed and constructive comments. We respond to the main points below.
>
> ### 1. Motivation for multivariate distributional RL
>
> Our motivation is to study multivariate distributional reinforcement learning in settings where returns are vector valued and discounting may differ across components. Sliced probability divergences provide a general tool for comparing such distributions, and support contraction guarantees for both scalar and matrix discounted Bellman operators. This complements existing work, which focuses primarily on empirical performance or scalar discounting.
>
> ### 2. Correlated reward functions
>
> Correlated reward components are handled naturally, as the framework models the full joint distribution of returns. The theoretical results, including contraction and fixed point properties, do not rely on independence assumptions and hold for arbitrary correlation structures.
>
> ### 3. Kernel choice in MMD
>
> We focused on the multiquadric kernel because it has been shown to perform well in multivariate distributional reinforcement learning and satisfies the regularity assumptions required by our analysis [1]. Using multiple bandwidths for RBF kernels is common in practice, but RBF kernels do not induce a contractive MMD in general, which prevents them from serving as a basis for the contraction results developed in this work. The multiquadric kernel is also empirically robust to bandwidth selection. Extending the theory to broader kernel families will be explored in a future submission.
>
> ### 4. Alternative metrics
>
> To the best of our knowledge, the only well known work that explicitly studies multivariate distributional reinforcement learning is based on MMD. We are not aware of prior work that investigates multivariate distributional RL through sliced divergences, so this line of work appears to be new. For a general treatment of sliced probability divergences, including their statistical and topological properties, we refer to [2].
>
> ### 5. Architecture
>
> The critic is implemented as a generative model that maps a state, action, and noise to a return sample, which is exactly the same push forward construction used in the generator of a GAN. Diffusion models rely on a different training paradigm and are typically more costly to implement. In our setting, we require a lightweight, reparametrization based generator because the framework is sample based.
>
> ### 6. Scalarization
>
> The scalarization rule used in the experiments is a standard linear aggregation of reward components, and is not specific to MO MuJoCo. It is widely used in multi objective reinforcement learning.
>
> ## References
>
> [1] Nguyen-Tang, T., Gupta, S., and Venkatesh, S.
> Distributional reinforcement learning via moment matching.
> Proceedings of the AAAI Conference on Artificial Intelligence, 35(10):9144–9152, 2021.
>
> [2] Nadjahi, K., Durmus, A., Chizat, L., Kolouri, S., Shahrampour, S., and Simsekli, U.
> Statistical and topological properties of sliced probability divergences.
> Advances in Neural Information Processing Systems, 33:20802–20812, 2020.

---

### Official Review · Reviewer_ZfEH · 2025-10-26

**Soundness:** 2
**Presentation:** 1
**Contribution:** 1
**Rating:** 2
**Confidence:** 4

**Summary:**

This paper focuses on the multivariate distributional RL setting. The authors proposed a new algorithm based on the sliced probability divergence, which can extend multiple common univariate distributional RL algorithms into a multivariate version within one framework. Theoretical results are provided with a focus on the Bellman contraction with a discussion of related properties. Experiments are conducted on several Mujoco environments with some simple baselines.

**Strengths:**

1. The proposed method seems novel and technically sound, which is established by multiple previous works on sliced probability divergence.

2. The theoretical framework seems to be unified, which includes plenty of common divergences as its instantiations. The analysis, from a general metric property to contraction, is sound.

**Weaknesses:**

There are multiple apparent weaknesses in this work.

1. **Limited and less clear motivation**. To the best of my knowledge, the multivariate distributional RL has already been explored by many prior works in the literature, such as MMD variant in Zhang et al. (2021); Sinkhorn divergence in Sun et al. (2024), and Wiltzer et al (2024), where the proposed algorithms have already achieved satisfactory performance in large-scale environments under relatively clear foundations. Since the main motivation of this paper is in this setting, it would be important to clearly articulate the advantages over existing algorithms and the differences from them, which thus helps the readers to posit the contribution of this paper. It seems that the current motivation is mainly explained in Line 124, which is relatively weak and less unified. This too specific direction may not gain broad interest in the community.

2. **Explanation of theoretical results is insufficient**. Although the proof seems comprehensive, which involves many aspects, I have a difficult time understanding the interpretation of the given theoretical results. For instance, the plain sliced method uses random projection, which is more likely to be inefficient. This has not been explained by any theoretical results. In particular, it lacks an explanation of Theorems 6 and 7. Can we find any advantages of the proposed method over other baselines from the perspective of sample complexity? What does $S_s$ in Line 296 mean? It lacks explanation. What is the insight to understand the non-dimension-dependent penalty? That seems counterintuitive, as the dependence must have some role to play. If not, why? Can the contraction plain sup-sliced in Theorem 3 be extended to the general $\Gamma_t$? What is $A_{s, a}(C)$ in Theorem 4, which may not be defined?


3. **Experiments are very limited**. Most of the published papers that develop distributional RL algorithms conduct experiments on Atari games, but this paper only provides limited results in several MuJoCo environments. It is well-known that the demonstration in the actor-critic framework tends to be less clear [1] compared with the pure value-based RL in environments with a discrete action space, such as Atari. Prior works such as Zhang et al. (2021) and Sun et al. (2024) have already provided results on several Atari games, making this paper’s experiment very limited. In addition, there lack of considered baselines. 5 random sees are insufficient. It is hard to interpret the entangled curves in Figure 1.


4. **Unfinished and unclear writing**.  (1) I found this submission is unfinished with an empty Appendix E.3, which largely affects my impression of this work. (2) Connections with other areas are discussed in a less sufficient way, which easily confuses. For instance, the authors try to connect the proposed method with GVF, but GVF is general in the sense that it generalizes the reward and allows a discounting to be a function of s_t and a_t. However, the proposed algorithms mainly consider a discounting function of time t without focusing on the prediction function beyond the rewards. Therefore, this connection is relatively weak and likely to confuse readers. (3) How do we understand control with scalarization in Line 237? Typically, we just use the Bellman optimal operator with a max operation in the control. (4) It is less clear what the computation cost would be when we use the max-slice mentioned in Line 203. Readers expect sufficient explanation in the main content instead of looking into details in the appendix. (5) What is the joint vector of multivariate returns in Line 210? Can it be explained mathematically? (6) What does the ‘true’ multi-objective control mean?

[1] Addressing Function Approximation Error in Actor-Critic Methods (ICML 2018)

**Questions:**

1. Max-sliced algorithms enjoy better theoretical properties at the cost of heavier computation. Does the experiment reflect this conclusion? If the practical improvement is limited, practitioners, in particular, are likely to quickly lose interest in this variant.

2. What would be the practice choice for $\Gamma_t$ in the proposed multivariate distributional RL setting? I understand the generality of such a formulation, but I suspect whether the setting is practical. For example, are there many papers that are in this setting with some choices of $\Gamma_t$ beyond the papers provided in Line 105?

3. The single choice of kernel in MMD seems questionable. Does the contraction of the proposed method hold for general kernels in MMD? If not, is there a counter-example available? If so, why?

---

> ### Author Response · Authors · 2025-12-03
>
> ## Rebuttal to Reviewer ZfEH
>
> We thank the reviewer for their detailed assessment and for recognizing the novelty and technical soundness of the proposed framework. We respond to the main concerns below.
>
> ### 1. Motivation and relation to prior work
>
> The primary motivation of this paper is to provide a general framework for multivariate distributional reinforcement learning with contraction guarantees for both scalar and general matrix discounted Bellman operators, using sliced probability divergences. Existing methods such as MMD based approaches or Sinkhorn divergences have reported strong empirical results, but do not provide contraction guarantees in the anisotropic or matrix discounted setting. In particular, the MMD based approach in [1] does not establish contraction guarantees for multivariate value distributions, so the theoretical foundation of that method remains incomplete. Our contribution is therefore complementary, and focuses on the theoretical aspects of multivariate distributional updates in general discounting scenarios.
>
>
>
> ### 2. Interpretation of theoretical results
>
> Regarding the comment on random projections and dimensional dependence, sliced probability divergences inherit their properties from the one dimensional base divergence, because all comparisons are carried out on projected one dimensional marginals. Under this construction, contraction bounds depend on the regularity of the base divergence and do not introduce explicit dependence on the ambient dimension. This is a standard result in the literature on sliced optimal transport and is not an artifact of our analysis. Random projections serve to approximate the integral over directions.
>
> ### 3. Experimental scope and baselines
>
> We acknowledge that the experimental evaluation is limited in scope. Our aim in this work is to demonstrate that the proposed framework can be instantiated with standard actor critic architectures and trained stably, not to provide a comprehensive empirical study. We chose multi objective MuJoCo tasks because they naturally produce multivariate returns in continuous control, and because large scale evaluations in multivariate Atari settings would require substantial additional computational effort. MMD with a multiquadric kernel was selected as a strong baseline [2].
>
> ### 4. Computational cost of max slicing
>
> Max slicing introduces additional computation because it optimizes projection directions to obtain contraction guarantees in the general matrix discounted setting. Contraction guarantees and empirical performance are distinct, and stronger guarantees do not imply better performance. Whether the additional cost yields practical benefits is application dependent.
>
> ### 5. Questions
>
> **Max slicing versus uniform slicing**
> Our experiments do not show consistent empirical advantages for max slicing. As noted above, contraction guarantees and empirical performance are distinct properties. We expect improvements to depend on domain and algorithmic refinements.
>
> **Practicality of general discounting**
> Problems with non scalar discounting arise, for example, in multi horizon settings and in value functions where different reward components evolve on different scales. These provide natural use cases for the proposed framework.
>
>
> **Kernel choice in MMD**
> We focused on the multiquadric kernel because it has been shown to be effective in multivariate distributional reinforcement learning and supports the regularity assumptions required by our analysis. Extending the theory to broader kernel families will be explored in a future submission.
>
> ## References
>
> [1] Zhang, P., Chen, X., Zhao, L., Xiong, W., Qin, T., and Liu, T.
> Distributional reinforcement learning for multi dimensional reward functions.
> Advances in Neural Information Processing Systems, 34:1519–1529, 2021.
>
> [2] Killingberg, L., and Langseth, H.
> The multiquadric kernel for moment matching distributional reinforcement learning.
> Journal of Machine Learning Research, 2023.

---

### Official Review · Reviewer_ue4b · 2025-10-28

**Soundness:** 3
**Presentation:** 3
**Contribution:** 2
**Rating:** 6
**Confidence:** 4

**Summary:**

This paper proposed Sliced Distributional Reinforcement Learning (SDRL), an algorithm framework for distributional RL which uses (max or L^p) sliced probability divergence as measure of error. They showed that the distributional Bellman operator is a contraction under L^p slicing for shared scalar discounts, and under max slicing for general matrix discounts. They also analyzed the sample complexity of estimating uniform and max slicing using Monte-Carlo methods. They provided numerical experiments to demonstrate the efficiency of the proposed SDRL.

**Strengths:**

The paper is well-written, clearly demonstrating the problem of multivariate distributional RL and how to effectively learn the joint distribution of return vectors by minimizing (max or L^p) sliced probability divergence. It provides clear contraction analysis of their algorithms and presents some experimental evidence.

**Weaknesses:**

1.Compared to Wiltzer et al. (2024a), this paper does not consider the finite-dimensional representation of probability distributions and only focuses on the simplified non-parametric setting, while this aligns with the experimental setup. Given that the work is not theoretically oriented, this weakness is acceptable .

2.As a paper focusing on distributional RL, the experiments in the main text do not demonstrate the effectiveness of the SDRL algorithm in fitting the joint distribution of returns. In my view, it is necessary to include experiments that showcase this aspect.

## Minor Points:

Use \eqref instead of \ref for references of equations.

In line 36 and 46, the notation n is used without explain.

In line 77-78, the sentence ‘Given a policy π(a|s), the agent seeks to maximize the expected discounted return’ seems confusing because Q is a vector and the policy \pi is given here.

In Equation (3) and (15), T^\pi should depend on time t.

In line 159 and 186, the notation L is used without explain.

In line 219, L represents projection count, but in line 360, L is a CDF-dominance constant, please avoid using the same notation.

In line 294-296, do not use s, which also represents state in this paper.

In line 342-343, the notation A_{s, a} is used without explain.

Theorem 6 uses \hat{\mu}_n and \mu, but Theorem 7 uses P_n and P, please use consistent notations.

**Questions:**

See weakness 2

---

> ### Author Response · Authors · 2025-12-03
>
> ## Rebuttal to Reviewer ue4b
>
> We thank the reviewer for their thoughtful assessment and for recognizing the clarity of the problem statement, the contraction analysis, and the experimental evidence. We respond to the weaknesses and questions below.
>
> ### 1. Scope of the theoretical contribution
>
> The paper is primarily theoretical. Our aim is to establish contraction guarantees for multivariate distributional reinforcement learning using sliced probability divergences, both under shared scalar discounts and general matrix discounted updates. The experimental section is intended to verify that such a framework can be instantiated in practice and trained stably, not to exhaust all possible algorithmic instantiations or parametrizations.
>
> ### 2. Finite dimensional representations
>
> > “This paper does not consider the finite dimensional representation of probability distributions and only focuses on the simplified non parametric setting.”
>
> We agree. Our focus is on establishing contraction and sample complexity guarantees for sliced divergences at the level of distributions, without committing to a specific finite representation. We view the choice of representation as an orthogonal design decision that can vary between applications and architectures. We note that many finite dimensional parametrizations commonly used in one dimensional distributional reinforcement learning, such as quantile or fixed support representations, can become challenging to scale when the return dimension increases, since the size of the parametrization often grows with the number of returns' components. From this perspective, working in a non parametric sample based regime is not a simplification, but a different modeling choice that aims to maintain flexibility in high dimensional settings. Extending the framework with scalable finite dimensional parameterizations is of interest.
>
>
> ### 3. Demonstrating the fit of the joint distribution
>
> > “The experiments do not demonstrate the effectiveness of the SDRL algorithm in fitting the joint distribution of returns. It is necessary to include experiments that showcase this aspect.”
>
> We agree. The experiments focus on policy performance and do not directly assess how well the learned models fit the joint return distribution. Evaluating distributional fit more explicitly is a valuable direction for future work, and we plan to explore this in a subsequent submission.

---

### Official Review · Reviewer_57qh · 2025-11-01

**Soundness:** 3
**Presentation:** 2
**Contribution:** 2
**Rating:** 2
**Confidence:** 3

**Summary:**

This paper introduces Sliced Distributional Reinforcement Learning (SDRL) — a new framework for distributional RL in multivariate settings based on sliced probability divergences (SPDs). Existing multivariate distributional RL methods show non-contractive and have high-dimensional optimal transport. To bridge this gap, SDRL uses sliced versions of 1D divergences, achieving both theoretical soundness and computational scalability. Furthermore, the paper extends this framework to a Maximum-Sliced variant (MSDRL), which optimizes over projection directions rather than averaging them, enabling provable contraction even under general matrix-discounted Bellman operators.

**Strengths:**

+ Defines Sliced Probability Divergences (SPDs) by projecting high-dimensional distributions into 1D directions and averaging 1D divergences, then introduce the SDRL.
+ Extends SDRL to maximum slicing, optimizing over directions instead of sampling and introduce MSDRL.
+ For both the SDRL and MSDRL frameworks, the paper provides theoretical guarantees for the contraction of the Bellman operator, proves that the proposed divergences preserve metric properties, and presents a detailed sample complexity analysis.
+ Reduces high-dimensional divergence computation to multiple 1D projections and improve the sample efficiency for multivariate Distributional Reinforcement Learning.

**Weaknesses:**

+ While SDRL generalizes across different divergence measures, its performance depends heavily on the choice of base metric. In addition, using random projection directions introduces sampling variance into the loss, causing slight fluctuations in the estimated divergences across batches.

+ The optimization over projection directions in MSDRL (via gradient ascent) increases computational cost, which may reduce its practical appeal compared to simpler variants.

+ Although the slicing technique is statistically consistent, it remains an approximation in practice since it is infeasible to integrate over infinitely many directions. Consequently, there is no guarantee of fully preserving high-dimensional structure after slicing. Each one-dimensional projection captures only linear combinations of variables along specific directions, potentially missing nonlinear or higher-order dependencies between dimensions.

+ The paper provides extensive theoretical analysis, but the experimental results do not demonstrate equally strong improvements. The empirical gains are modest — SDRL and MSDRL often match but rarely surpass MMD-based baselines. Moreover, the max-sliced variants, while theoretically superior, do not exhibit clear empirical advantages.

+ All experiments are conducted on standard MuJoCo control tasks, lacking evaluation on large-scale or complex high-dimensional domains. The work also does not address true multi-objective control, where learning must generalize across multiple scalarization rules.

+ The notation in the paper could be clearer; several symbols and terms are referenced before being properly defined, which may hinder readability.

**Questions:**

1.	The contraction proofs rely on several strong regularity assumptions about the base divergence, such as translation invariance and scaling Lipschitz continuity. However, these assumptions may not hold in complex multivariate distributional reinforcement learning settings. It remains unclear how the theoretical guarantees would extend—or whether they would still hold—when these conditions are violated.
2.	Are the theoretical results holds for all divergences or kernels (especially in high-dimensional MMD)?

---

> ### Author Response · Authors · 2025-12-03
> **Part 1**
>
> We thank the reviewer for the careful reading of the paper and for the positive assessment of the soundness and clarity of the theoretical framework. Our main goal is to introduce a general framework for multivariate distributional reinforcement learning with contraction guarantees under scalar and matrix discounting, rather than to propose a new state of the art algorithm. We respond to the specific points below.
>
> ### 1. Role of the base divergence
>
> > “Performance depends heavily on the choice of base metric.”
>
> We agree. This dependence is intrinsic to any divergence based distributional method. Our contribution is not that all base divergences perform similarly, but that a wide family of one dimensional base metrics that satisfy simple properties can be lifted to the multivariate setting and retain contraction via sliced or max sliced constructions. This is the focus of Theorems 3 and 4 and the instantiations in Table 1.
>
> The paper therefore separates two aspects
> 1. theoretical guarantees that hold for a broad class of base divergences
> 2. empirical performance for particular instantiations, which may differ between divergences
>
> ### 2. Variance from random projections
>
> > “Using random projection directions introduces sampling variance into the loss.”
>
> We agree that random slicing introduces variance. This is a known feature of sliced divergences that comes with two well understood mitigation strategies
> - using more random directions, which is computationally cheap in the sliced setting
> - using optimized directions, that is the max sliced variant, which trades some extra computation for reduced variance
>
> Our goal is to show that both variants fit naturally into a distributional Bellman framework and retain contraction properties under suitable conditions.
>
> ### 3. Cost of max slicing
>
> > “Optimization over projection directions in MSDRL increases computational cost.”
>
> We agree. Max slicing requires iterative optimization of a direction on the unit sphere, which adds overhead relative to uniform slicing. The reason to introduce MSDRL is that it extends contraction guarantees to general matrix discounted and anisotropic updates, which are not covered by uniform slicing.
>
> In other words, MSDRL is motivated by theory, not by an expectation of uniform empirical gains. Whether max slicing is the right practical choice will depend on the application and resource budget.
>
> ### 4. Approximation nature of slicing and preservation of structure
>
> > “There is no guarantee of fully preserving high dimensional structure after slicing.”
>
> Sliced divergences compare high dimensional distributions through their one dimensional projections. This is exactly the setting studied in the Cramér Wold and sliced Wasserstein literature that underpins our work. The theoretical guarantees we prove concern metricity and contraction of the lifted divergences, not perfect reconstruction of arbitrary high dimensional structure from a finite number of projections. Our contribution is to show that, under the stated assumptions on the base divergence, the sliced and max sliced multivariate Bellman operators are contractive, with constants that do not introduce explicit dimension dependent penalties.
>
> ### 5. Empirical gains and comparison with MMD
>
> > “Empirical gains are modest and SDRL and MSDRL often match but rarely surpass MMD based baselines.”
>
> We agree, and we would like to clarify our intent. The empirical part of the paper is not meant as a competition against MMD, but as a feasibility study. It shows that SDRL and MSDRL can be implemented with standard architectures and hyperparameters, that they train stably, and that they are competitive with a strong baseline.
>
> Our MMD baseline uses the multiquadric kernel, which is highlighted in prior work [1] as a strong choice distributional reinforcement learning. Using this kernel provides a demanding baseline and was deliberate.

---

> > ### Author Response · Authors · 2025-12-03
> > **Part 2**
> >
> > ### 6. Choice of environments and scale
> >
> > > “All experiments are conducted on standard MuJoCo tasks, lacking evaluation on large scale or high dimensional domains.”
> >
> > We agree that larger scale experiments such as Atari would be interesting. In our work we chose multi objective MuJoCo tasks from MO Gymnasium because
> > - they are canonical benchmarks for multivariate rewards in continuous control
> > - they allow instantiating both multi objective and multihorizon settings with a controlled computational budget
> > - large scale Atari experiments in a multivariate distributional setting would require substantial additional resources that were not available to us
> >
> > The aim of the experiments is to validate that the proposed framework is usable and competitive in a standard continuous control setting, not to exhaust all possible benchmarks.
> >
> > ### 7. Multi objective control versus fixed scalarization
> >
> > > “The work does not address true multi objective control, where learning must generalize across multiple scalarization rules.”
> >
> > We agree and consider this outside the scope of the paper. Our focus is distributional policy evaluation for a fixed scalarization rule in multivariate reward settings. This matches the setup of several prior works that use multi objective environments as an instance of multivariate temporal difference learning. Extending SDRL or MSDRL to preference conditioned policies is an interesting direction for future work.
> >
> > ### 8. Assumptions on the base divergence
> >
> > > “The contraction proofs rely on strong regularity assumptions, it is unclear how guarantees extend when conditions are violated.”
> >
> > The assumptions in Theorem 2, namely translation nonexpansion, scale Lipschitzness, and mixture convexity, are properties of the one dimensional base divergence. They do not depend on the dimension or complexity of the multivariate environment.
> >
> > The slicing step takes such a base divergence and aggregates its one dimensional evaluations across directions. Hence, if the base divergence has these properties in one dimension, the lifted divergences used in SDRL and MSDRL inherit the contraction guarantees we state. When a particular divergence does not satisfy these assumptions, our theorems simply do not apply, which we view as a limitation of the divergence, not of the sliced framework itself.
> >
> > ### 9. Applicability to MMD and other kernels
> >
> > > “Do the theoretical results hold for all divergences or kernels, especially in high dimensional MMD.”
> >
> > The results do not hold for all possible divergences or kernels. They hold for any base divergence that satisfies the structural conditions described in Theorem 2. For MMD, this translates into conditions on the kernel.
> >
> > In the paper we instantiate these results for the multiquadric kernel, which is known to be contractively well behaved in the relevant sense and empirically strong in multivariate distributional reinforcement learning. Other kernels can be treated in the same framework when they satisfy the same regularity assumptions.
> >
> > ## References
> >
> > [1] Killingberg, L. and Langseth, H.
> > The multiquadric kernel for moment matching distributional reinforcement learning
> > Journal of Machine Learning Research, 2023.

---

### Note · Authors · 2025-12-03

**Comment:**

We thank the reviewers for their time and constructive feedback. Although we will withdraw the paper, we would like to briefly outline how we plan to improve this line of work.

First, the submission did not clearly communicate that the contribution is primarily theoretical. A revised presentation will place stronger emphasis on the theoretical nature of the work, namely establishing contraction guarantees for multivariate distributional Bellman operators under sliced probability divergences, and clarifying how this differs from existing approaches. We will also make it clearer that contraction guarantees should not be interpreted as implying improved empirical performance, as the two properties are conceptually distinct.

Second, we will expand the empirical section in future work. In particular, we aim to include experiments on discrete action environments such as Atari, and investigations of how well the learned distributional critic captures the true return distribution through appropriate experiments and visualisation, rather than relying solely on downstream control performance.

Third, the theoretical analysis of sliced MMD can be extended. We can generalize the contraction arguments beyond the multiquadric kernel to other kernels.

We appreciate the reviewers’ comments and hope that a future submission will better convey the scope and potential of this research direction.

**Withdrawal Confirmation:**

I have read and agree with the venue's withdrawal policy on behalf of myself and my co-authors.